# Newly identified climatically and environmentally significant high-latitude dust sources

Outi Meinander[1], Pavla Dagsson-Waldhauserová[2,3], Pavel Amosov[4], Elena Aseyeva[5], Cliff Atkins[6], Alexander Baklanov[7], Clarissa Baldo[8], Sarah Barr[9], Barbara Barzycka[10], Liane G. Benning[11,23], Bojan Cvetkovic[12], Polina Enchilik[5], Denis Frolov[5], Santiago Gassó[13], Konrad Kandler[14], Nikolay Kasimov[5], Jan Kavan[15,b], James King[16], Tatyana Koroleva[5], Viktoria Krupskaya[5,6], Markku Kulmala[18], Monika Kusiak[19], Hanna K Lappalainen[18], Michał Laska[11], Jerome Lasne[20], Marek Lewandowski[19], Bartłomiej Luks[19], James B McQuaid[10], Beatrice Moroni[21], Benjamin J Murray[10], Ottmar Möhler[22], Adam Nawrot[19], Slobodan Nickovic[13], Norman T. O'Neill[23], Goran Pejanovic[13], Olga B. Popovicheva[5], Keyvan Ranjbar[23,a], Manolis N. Romanias[20], Olga Samonova[5], Alberto Sanchez-Marroquin[10], Kerstin Schepanski[24], Ivan Semenkov[5], Anna Sharapova[11], Elena Shevnina[1], Zongbo Shi[9], Mikhail Sofiev[1], Frédéric Thevenet[20], Throstur Thorsteinsson[25], Mikhail A. Timofeev[5], Nsikanabasi Silas Umo[22], Andreas Uppstu[1], Darya Urupina[20], György Varga[26], Tomasz Werner[19], Olafur Arnalds[2], and Ana Vukovic Vimic[27]

[1]Finnish Meteorological Institute, Helsinki, 00101, Finland
[2]Agricultural University of Iceland, Reykjavik, 112, Iceland
[3]Czech University of Life Sciences Prague, Prague, 16521, Czech Republic
[4]INEP Kola Science Center RAS, Apatity, Russia
[5]Lomonosov Moscow State University, Moscow, 119991, Russia
[6]Institute of Geology of Ore Deposits, Petrography, Moscow, 119017, Russia
[7]Te Herenga Waka—Victoria University of Wellington, Wellington, 6012, New Zealand
[87]World Meteorological Organization, WMO, Geneva, 1211, Switzerland
[9]University of Birmingham, Birmingham, B15 2TT, United Kingdom
[10]University of Leeds, Leeds, LS2 9JT, United Kingdom
[11]University of Silesia in Katowice, Sosnowiec, 41-200, Poland
[12]German Research Centre for Geosciences, Helmholtz Centre Potsdam, 14473, Germany
[13]Republic Hydrometereological Service of Serbia, 11030, Belgrade, Serbia
[14]University of Maryland, College Park MD, 20742, United States of America
[15]Technical University of Darmstadt, Darmstadt, 64287, Germany
[16]Masaryk University, Brno, 61137, Czech Republic
[17]University of Montreal, Montreal, H3T 1J4, Canada
[18]Institute for Atmospheric and Earth System Research, University of Helsinki, Helsinki, 00101, Finland
[19]Institute of Geophysics, Polish Academy of Sciences, Warsaw, 01-452, Poland
[20] Institut Mines-Télécom Nord Europe, Université de Lille, Center for Energy and Environment, Lille, 59000, France
[21]University of Perugia, Perugia, 06123, Italy
[22] Institute of Meteorology and Climate Research, Karlsruhe Institute of Technology, Karlsruhe, 76227, Germany.
[23]Université de Sherbrooke, Sherbrooke, J1K, Canada
[24]Free University of Berlin, Berlin, 12165, Germany
[25]University of Iceland, Reykjavik, 102, Iceland

[26]Research Centre for Astronomy and Earth Sciences, Budapest, 1112, Hungary
[27]University of Belgrade, Faculty of Agriculture, Belgrade, 11080, Serbia
[a]now at: Flight Research Laboratory, National Research Council Canada, Ottawa, ON, Canada
[b]now at: University of Wroclaw, Wroclaw, 50-137, Poland
*Correspondence to*: Outi Meinander (outi.meinander@fmi.fi)
**Abstract.** Dust particles from high latitudes have a potentially large local, regional, and global significance to climate and the
environment as short-lived climate forcers, air pollutants, and nutrient sources. Identifying the locations of local dust sources
and their emission, transport, and deposition processes is important for understanding the multiple impacts of High Latitude
Dust (HLD) on the Earth's systems. Here, we identify, describe, and quantify the Source Intensity (SI) values, which show the
potential of soil surfaces for dust emission scaled to values 0 to 1 concerning globally best productive sources, using the Global
Sand and Dust Storms Source Base Map (G-SDS-SBM). This includes sixty-four HLD sources in our collection for the
Northern (Alaska, Canada, Denmark, Greenland, Iceland, Svalbard, Sweden, and Russia) and Southern (Antarctica and
Patagonia) high latitudes. Activity from most of these HLD dust sources shows seasonal character. It is estimated that high-
latitude land areas with higher (SI$\geq$0.5), very high (SI$\geq$0.7), and the highest potential (SI$\geq$0.9) for dust emission cover >1
670 000 km$^2$, >560 000 km$^2$, and >240 000 km$^2$, respectively. In the Arctic HLD region ($\geq$ 60°N), land area with SI$\geq$0.5 is
5.5% (1 035 059 km$^2$), area with SI$\geq$0.7 is 2.3% (440 804 km$^2$), and with SI$\geq$0.9 is 1.1% (208 701 km$^2$). Minimum SI values
in the north HLD region are about three orders of magnitude smaller, indicating that the dust sources of this region greatly
depend on weather conditions. Our spatial dust source distribution analysis modeling results showed evidence supporting a
northern High Latitude Dust (HLD) belt, defined as the area north of 50°N, with a 'transitional HLD-source area' extending
at latitudes 50–58°N in Eurasia and 50–55°N in Canada, and a 'cold HLD-source area' including areas north of 60°N in Eurasia
and north of 58°N in Canada, with currently 'no dust source' area between the HLD and LLD dust belt, except for British
Columbia. Using the global atmospheric transport model SILAM, we estimated that 1.0% of the global dust emission
originated from the high-latitude regions. About 57% of the dust deposition in snow- and ice-covered Arctic regions was from
HLD sources. In the south HLD region, soil surface conditions are favorable for dust emission during the whole year. Climate
change can decrease snow cover duration, retrieval of glaciers, and increase drought, heatwave intensity, and frequency,
leading to the increasing frequency of topsoil conditions favorable for dust emission, which increases the probability of dust
storms. Our study provides a step forward to improve the representation of HLD in models and to monitor, quantify, and assess
the environmental and climate significance of HLD going forward.

## 1 Introduction

Mineral dust is an essential and relevant climate and environmental variable with multiple socioeconomic effects on, e.g., weather and air quality, marine life, climate, and health (Creamean et al., 2013; Terradellas et al., 2015; Shepherd et al., 2016; Querol et al., 2019; Nemuc et al., 2020). Mineral dust is transported from local sources of high-latitude dust (HLD, ≥50°N and ≥40°S, Bullard et al., 2016), low-latitude dust (LLD, mostly 0-35°N), and the so-called 'global dust belt' (GDB, Prospero et al., 2002), defined to extend into the Northern Hemisphere from the west coast of North Africa, over the Middle East (West Asia), Central and East Asia, and south-west North America (Ginoux et al., 2012), with only minor sources in Southern Hemisphere (Prospero et al., 2002; Ginoux et al., 2012; Bullard et al., 2016; Terradellas et al., 2017). Dust is often associated with hot, subtropical deserts, but the importance of dust sources in the cold, high latitudes has recently increased (Arnalds et al., 2016; Bullard et al., 2016; Groot Zwaafting et al., 2016, 2017; Kavan et al., 2018, 2020a,b; Boy et al., 2019; Gassó and Torres, 2019; IPCC, 2019; Tobo et al., 2019; Bachelder et al., 2020; Cosentino et al., 2020; Ranjbar et al., 2021; Sanchez-Marroqin et al., 2020). Dust produced in high latitudes and cold climates (Iceland, Greenland, Svalbard, Alaska, Canada, Antarctica, New Zealand, and Patagonia) can have regional and global significance (Bullard et al., 2016). Local HLD dust emissions are increasingly being recognized as driving the local climate, biological productivity, and air quality (Groot Zwaafting et al., 2016, 2017; Moroni et al., 2018; Crocchianti et al., 2021; Varga et al., 2021). HLD can induce significant direct (blocking sunlight) and indirect (clouds and cryosphere) radiative forcing (Kylling et al., 2018) on solar radiation fluxes and snow optical characteristics, strongly impacting Arctic amplification, including glacier melt (Boy et al., 2019).

HLD aerosols consist of a variety of different dust particle types with various particle sizes and shapes distributions, as well as physical, chemical, and optical properties that differ from the crustal dust of the Sahara or American deserts (Shepherd et al., 2016; Arnalds et al., 2016; Bachelder et al., 2020; Baldo et al., 2020; Crucius, 2021). Therefore, impacts on climate, environment, and human health can differ from those of LLD. For example, Icelandic dust is of volcanic desert origin, often dark, and has higher proportions of heavy metals than crustal dust (Arnalds et al., 2016). The IPCC special report (IPCC, 2019) recognizes dark dust aerosols as a short-lived climate forcer (SLCF) and light-absorbing aerosols connected to cryospheric changes. Light-absorbing HLD particles can induce direct effects on solar radiation fluxes as SLCF and snow optical characteristics impacting cryosphere melt via radiative feedback (Peltoniemi et al. 2015; Boy et al., 2019; Dagsson-Waldhauserová and Meinander, 2019, 2020; IPCC, 2019; Kylling et al., 2018). HLD significantly affects the formation and properties of clouds (Abbatt et al., 2019; Sanchez-Marroquin et al., 2020; Murray et al., 2021).

Dust is connected to climate change: Historical dust (paleo dust) is not only a contributor to climate change but a record of previous dust and climate conditions (Lamy et al., 2014; Lewandowski et al., 2020). Dust can significantly contribute to air pollution mortalities (Terradellas et al., 2015; Nemuc et al., 2020). Deposition at high latitudes can provide nutrients to the marine system; mineral and organic matter on glaciers, including natural and anthropogenic dust, can form cryoconite granules.

Cryoconite, dust, and ice algae can reduce surface albedo and accelerate the melting of glaciers (Lutz et al., 2016; McCutcheon et al., 2021). Monitoring dust in remote, high-latitude areas has crucial value for climate change assessment and understanding the impacts of global warming on natural systems and socioeconomic sectors. Bullard et al. (2016) summarized natural HLD sources as covering over 500 000 km$^2$ and producing particulate matter of ca. 100 Mt dust per year.

Dust emissions respond to changes in wind speed, soil moisture, and other parameters affected by climate change; changes in land cover and surface properties by human activities can affect dust emissions (Kylling et al., 2018). The fundamental processes controlling aeolian dust emissions in high latitudes are essentially the same as in temperate regions. However, there are other processes specific to or enhanced in cold regions. Low temperatures, humidity, strong winds, permafrost, and niveo-aeolian processes, which can affect the efficiency of dust emission and distribution of sediments, were listed in Bullard et al. (2016).

The modeling of emissions, transport, and deposition complemented with available observations, can provide essential information related to dust's impact on the climate and environment in the high latitudes (IPCC, 2019). The locations and characteristics of local dust sources are two of the major observations documented for inputting information into numerical models to predict or simulate the HLD process from its emission to downwind deposition. In some cases, model results can indicate possible but not yet identified dust sources in the HL regions. A general lack of observational and long-range transport modeling studies results in poor HLD monitoring and predicting. Models have predictive capacity and, without the observations, can constitute a source of information and indicate where more direct observations are needed. The first long-range transport modeling studies show that main transport pathways from HLD sources clearly affect the High Arctic (>80°N) and European mainland (Baddock et al., 2017; Beckett et al., 2017; Đorđević et al., 2019; Groot Zwaafting et al., 2016, 2017; Moroni et al., 2018). The World Meteorological Organization Sand and Dust Storm Warning Advisory and Assessment System (WMO SDS-WAS) monitors and predicts dust storms from the world's major deserts (https://www.wmo.int/sdswas), where HLD sources have recently been included in the SDS-WAS dust forecasts. Europe's largest desert is at a high latitude in Iceland (Arnalds et al., 2016), with dust transport observed over the North Atlantic to European countries (Ovadnevaite et al., 2009; Prospero et al., 2012; Beckett et al., 2017; Đorđević et al., 2019).

HLD is a short-lived climate forcer, air pollutant, and nutrient source, showing the need to identify the geographical extent and dust activity of the HLD sources (Arnalds et al., 2014, 2016; Dagsson-Waldhauserová et al., 2014, 2015; Terradellas et al., 2015; USGCRP, 2018; IPCC, 2019). Bullard et al. (2016) designed the first HLD map based on visibility and dust observations, combined with field and satellite observations of high-latitude dust storms, resulting in 129 locations described in 39 papers. Here, we compile and describe sixty-four HLD sources in the northern and southern high latitudes. This work's main aim is to:

       (i)       identify new and previously unpublished HLD sources ,

(ii)      estimate the high-latitude land area with potential dust activity and calculate the source intensity (SI) for the

identified sources

(iii)     provide model results on HLD emission, long-range transport and deposition at various scales of time and

space

(iv)     specify key climatic and environmental impacts of HLD and related research questions, which could improve

our understanding of HLD sources, with the help of literature surveys on clouds and climate feedback,

atmospheric chemistry, marine environment, cryosphere, and cryosphere-atmosphere feedbacks.


We focus on high latitudes with natural dust sources and include some anthropogenic dust sources, such as road dust, when
unpaved roads serve as a significant dust source. Direct emissions of volcanic eruptions and road dust formed via abrasion and
wear of pavement or traction control materials are excluded. Identifying dust sources is the first step to understand the HLD
life-cycle (dust emission, transport, and deposition). After that, impacts and feedback mechanisms can be identified and
quantified as physical, chemical, and optical properties of dust from these source areas. Their properties during emission,
transport, and deposition are needed to be characterized to allow a holistic understanding.

## 2 Materials and methods

### 2.1 Identification and characteristics of dust sources

Three topical workshops in Russia, Finland, and Iceland (Meinander et al., 2019a,b) on HLD were organized in 2019 to
identify, describe, and assess new high-latitude dust sources ($\geq 50°$N and $\geq 40°$S, according to Bullard et al. 2016, andincluding
the Arctic as a subregion at $\geq 60°$N). The HLD source map and observations on dust properties provided here are based on:

(i)      new field and satellite observations not described in published academic papers

(ii)     newly identified HLD source locations reported in recent literature but not included in previous collections

(iii)    updated observations on previously documented sources.

Each location was assessed to classify each source: Category 1 refers to an active dust source with high ecological significance,
category 2 to a semi-active source with moderate ecological significance, and category 3 to new sources with unknown activity
and importance. Moreover, SI values for each HLD location in the Northern and Southern (Antarctica and Patagonia) high
latitudes were quantified, and the potential land surface area for dust emissions in the north, Arctic, and south HLD regions
was calculated (Section 2.2).

### 2.2 High-latitude dust sources from UNCCD G-SDS-SBM

The Global Sand and Dust Storms Source Base Map (G-SDS-SBM), developed by the United Nations Convention to Combat
Desertification (UNCCD) in collaboration with the United Nations Environment Programme (UNEP) and World
Meteorological Organization (https://maps.unccd.int/sds/; Vukovic, 2019, 2021) represents gridded values of SDS source

intensity (SI, values 0 to 1) on a resolution of 30 arcsec. The Source Base Map was developed by including the information on soil texture, bare land fraction, and NASA satellite Moderate-resolution Imaging Spectroradiometer Enhanced Vegetation Index, MODIS EVI, as well as the data on land cover, topsoil moisture, and temperature. Values of SI represent topsoil's potential to emit soil particles under windy conditions, assigning the highest values of source intensity to the most productive surfaces. SI values are derived under the assumption they are exposed to the same velocity of surface wind. Input data, which change depending on the weather (and possibly human activities) for bare land fraction, moisture, and temperature data, are defined for four months (January, April, July, October—each month representing one season) by using extreme values. This was observed from 2014 to 2018, providing favorable conditions for surfaces to act as sources. Thus, sources that may appear during heatwaves and drier conditions (or drought), when the surface in high latitudes is unfrozen, snow-free, and more susceptible to wind erosion, are included in this map. Such weather extremes under climate change are becoming more frequent and are projected to increase (IPCC, 2013), justifying the source mapping approach using the information on extreme topsoil conditions. Using the maps produced for the four seasons, maximum and minimum values are determined for each grid point to explore the potential of high-latitude land surfaces to act as dust sources, their seasonality, and to compare values of source intensity with marked locations of HLD sources.

**2.3 Methods used to identify and study the dust sources**

Various methods identified the HLD sources (Table 1), including direct observations and measurements; satellite data; emission, long-range transport and deposition modeling; media, social media, and literature sources (e.g., web pages, conference abstracts). More details and literature references can be found in each source section. Dust emission, long-range transport, and deposition modeling calculations were made to study if the HLD sources have local, regional, or global significance. Two well-established dust atmospheric models—SILAM and DREAM—were used to simulate the atmospheric dust process over high latitudes. Both models have been thoroughly evaluated for other deserts where the accuracy of their results has been verified.

**Table 1. Methods used to identify and study the dust sources**

| Method | Sources |
|---|---|
| Direct observation: photographs and visual observations | Marambio, Antarctic Peninsula, Schirmacher Oasis, East Antarctica McMurdo Sound/Ross Sea |
| Satellite images: Meteosat-11 images | Denmark, Sweden, Iceland |
| Instrumentation: SEM | Svalbard |

| | |
|---|---|
| Instrumentation: LOAC | James Ross Island |
| Instrumentation: SL-501 surface and snow albedo | Marambio, Antarctic Peninsula |
| Instrumentation: Magnetic susceptibility upon heating, magnetic hysteresis parameters | Svalbard |
| Instrumentation: ICP-MS, AES-ICP, XRD, XRF | Russia (sources no. 2–5 of Fig.1) |
| Instrumentation: high performed liquid chromatography, potentiometry | Russia (sources no. 7–8 of Fig.1) |
| Passive deposition samplers | James Ross Island |
| Snow samples | Svalbard (Hornsund, Pyramiden), Antarctica |
| Social media: Twitter account (@SanGasso) and hashtag (#highlatitudedust) | South America (Patagonia), Alaska, Greenland, Iceland |
| Literature sources | Denmark, Sweden |
| SILAM model | Arctic |
| DREAM model | Arctic, Antarctic |



Estimates of the emission and deposition of global and Arctic dust were computed separately to assess Arctic dust's global
impact using the SILAM model (Sofiev et al., 2015)—a global to meso-scale atmospheric dispersion and chemistry model—
applied for air quality and atmospheric composition modelling. The dust emission estimate is driven by the European Centre
for Medium Range Weather Forecast ECMWF IFS meteorological model at a resolution of 0.1 x 0.1 degrees. The computations
were performed using ECMWF ERA5 meteorological reanalysis data for 2017 at a resolution of 0.5 x 0.5 degrees. The dust
emission model was validated against AERONET (AErosol RObotic NETwork, www.aeronet.com) aerosol optical density
(AOD) data and provided unbiased results for the main dust emission areas. For Arctic areas, where dust is not contributing to
the AOD as much, the simulated AOD from all aerosols is unbiased concerning the measurements. While the simulation's
relatively coarse resolution cannot capture the smaller point-like dust sources, it is still expected to give a good approximation
of the overall patterns and magnitudes of dust emission and deposition. The SILAM results are presented in sections 3.3 (Fig.
4) and 3.4 (Fig. 12 and Fig. 15).

DREAM is a fully dynamic numerical prediction model for atmospheric dust dispersion originating from soil. The dust component of this system (Pejanovic et al., 2011; Nickovic et al., 2016) is online and driven by the atmospheric model NMME (Janjic et al., 2001). Dust concentration in the model is described with eight particle bins, with radii ranging from 0.18 to 9 μm. DREAM-ICELAND is the model version to predict dust transport emitted from Iceland's largest European dust sources (Cvetkovic et al., 2021, submitted). The size distribution of particles in the model is specified according to in situ measurements in the Icelandic hot spots. The model horizontal resolution of ~3.5 km is sufficiently fine to resolve the Icelandic dust sources' rather heterogeneous and small-scale character. As the first operational numerical HLD model in the international community, DREAM-ICELAND is used daily, having predicted Icelandic dust since April 2018. DREAM results are included in sections 3.4 (Fig. 8 and 11) and 3.6 (Fig. 16), and as a supplementary animation.

## 3 Results and discussion

### 3.1 Locations of the HLD sources

Sixty-four HLD sources at northern and southern high latitudes (Fig. 1) were identified. In the north HLD region are 49 locations (47 locations ≥50°N and two >47°N) in Alaska, Canada, Denmark, Greenland, Iceland, Svalbard, Sweden, and Russia, of 35 are in the Arctic HLD subregion (≥60°N). In the south HLD region (≥40°S), 15 sources were identified in Antarctica and Patagonia, South America. The sources included the Arctic and Antarctic, boreal, remote, rural, mountain, marine and coastal, river sediments, mining, unpaved roads, soils (Podzols, Retisols, Gleysols, Phaeozems, and Stagnosols; USS Working Group WRB, 2015), and glacial dust. The observational periods for these locations varied from days or weeks to multiple years and included data from ground-based measurements, remote sensing data, and modeling results. Results on the calculated source intensity and areas of high-latitude surface land with higher (SI≥0.5), very high (SI≥0.7), and the highest potential (SI≥0.9) for dust emission are shown in Section 3.2. Observations and characteristics of the identified dust sources in our collection (Fig. 1) are presented in Section 3.4 and the Supplement Tables S1-S7 (including the contemporary classification for each source into categories 1–3, based on the currently available observations, in S1; satellite observations on new HLD sources in Iceland in S2; observations on new HLD sources in Greenland and Canada in S3; SI values for each source in S4 and S5, including latitude and longitude; and results from Russian HLD sources in S6-S7).

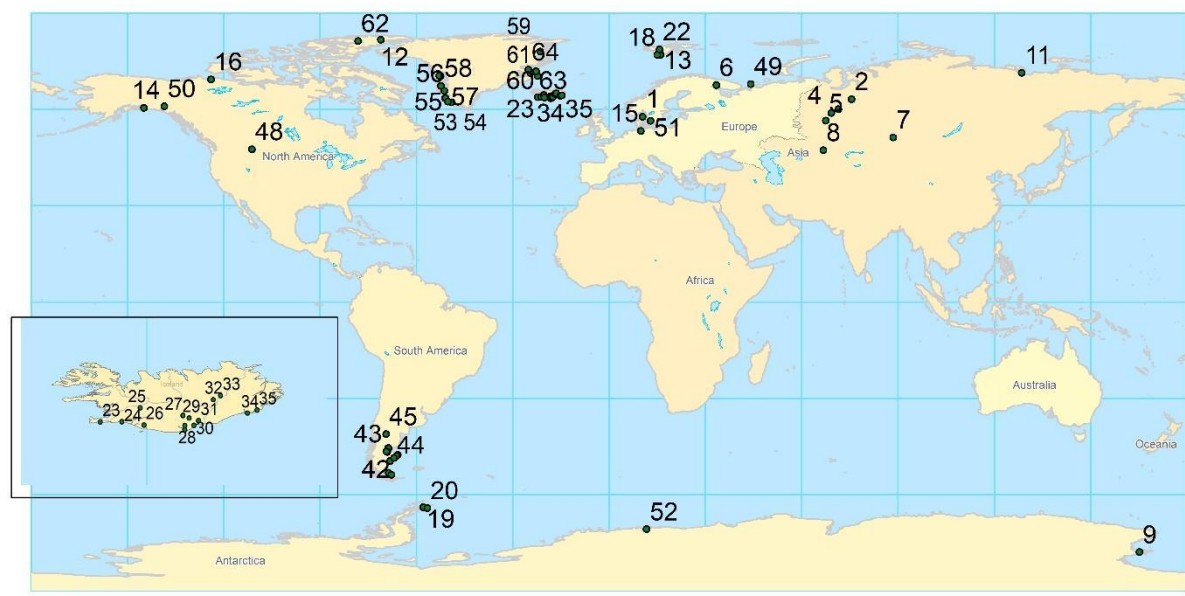

**Figure 1. Map of the locations of the northern (north of 50°N) and southern (south of 40°S) high-latitude dust (HLD) sources identified and included in this study. The numbers are the identified 64 dust sources, as shown in Figure 1.**

### 3.2 Source intensity from UNCCD G-SDS-SBM

Figure 2 presents the G-SDS-SBM source intensity values (maximum and minimum) for the north HLD region. The north HLD region includes the area north of latitude 50°N and the Arctic region (as a subregion of the HLD region) north of 60°N. HLD dust sources show extreme seasonal characteristics, with some exceptions. The sources appear and disappear (or change SI values) seasonally or appear (or increase source intensity values) only during favorable extreme weather conditions. Figure 3 shows G-SDS-SBM source intensity values for the south HLD region (south of 40°S) without values for Antarctica since G-SDS-SBM does not include areas south of 60°S. Supplementary Tables S4 and S5 give the values of SI for specific locations marked in Figure 1. Further analysis consists of assessing the areal coverage of sources, with different thresholds for SI values in absolute values (km$^2$) and the percentage they occupy concerning the total land surface area in the defined HLD regions.


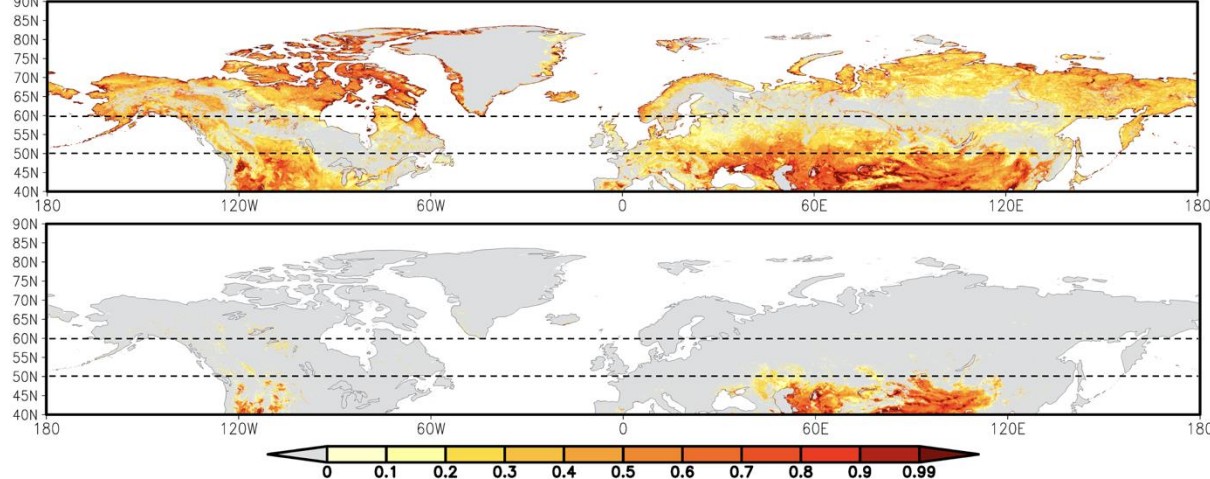

**Figure 2. UNCCD Global Sand and Dust Storms Source Base Map (G-SDS-SBM) for annual maximum (upper panel) and**
**minimum (lower panel) source intensity for the north HLD region and Arctic sub-region (north of 50°N and 60°N, respectively,**
**marked with dashed lines).**

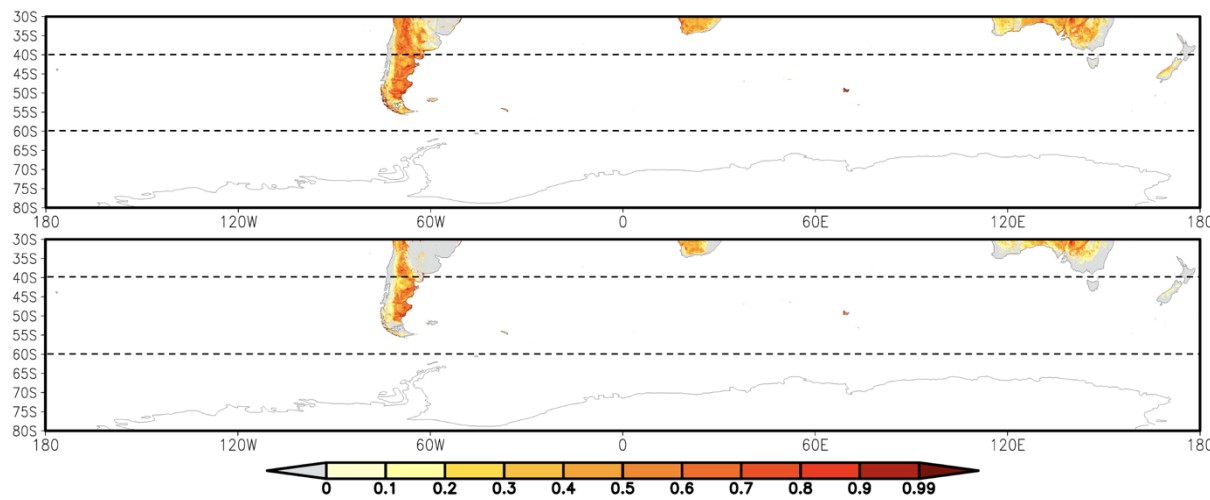

**Figure 3. UNCCD Global Sand and Dust Storms Source Base Map (G-SDS-SBM) for annual maximum (upper panel) and**
**minimum (lower panel) source intensity for the south HLD region (south of 40°S) without Antarctica (south of 60°S), marked with**
**dashed lines.**

The total surface area of dust sources with a higher potential for dust emission (SI$\geq$ 0.5) over the north HLD region (north of
50°N) is 3.9% of the total land surface (1 364 799 km$^2$). The area with a very high potential for dust emission (SI$\geq$0.7) is 1.5%
(509 965 km$^2$). The area with the highest dust emission potential (SI$\geq$0.9) is 0.7% of the total land area (233 336 km$^2$) (Table

2). In the Arctic region (north of 60°N)—the subregion of the north HLD area—dust sources with a higher potential for dust emission (SI≥0.5) are 5.5% of the total land surface (1 035 059 km$^2$). The area with a very high potential for dust emission (SI≥0.7) is 2.3% (440 804 km$^2$). The area with the highest dust emission potential (SI≥0.9) is 1.1% (208 701 km$^2$). The surface of dust-productive areas with minimum seasonal SI values in the north HLD region is about three orders of magnitude smaller than the maximum, meaning the north HLD dust sources highly depend on weather conditions. Maximum surfaces contain dust-productive regions that are defined under the most favorable weather conditions for soil exposure to wind erosion (including extreme weather). All sources defined here are not necessarily active every year nor in the same period, meaning these surfaces can seasonally or occasionally (under severe weather) appear as dust sources.

For the south HLD region (40°S–60°S, area without Antarctica), the land surface is only 2% of the total surface area (Table 3). The surface area of dust sources with SI≥0.5 is 22.6% of the total land surface (309 520 km$^2$). The area with SI≥0.7 is 4.5% (61 527 km$^2$). The area with the highest dust emission potential (SI≥0.9) is 0.6% (8 630 km$^2$). The surface areas for minimum SI values above these thresholds are two to three times smaller than the surfaces for maximum SI values compared to the difference in the north HLD region. This means that soil surface conditions in the south HLD region are favorable for dust emission over the whole year. Especially in locations of HLD markers, SI maximum and minimum values do not change over most locations or decrease by 0.1 or 0.2, except for one location (no. 38), which has SI values changing from 0.9 to 0 at the location of an HLD marker.

**Table 2. Relevant surfaces for the north HLD and Arctic regions: surface of total area of the region, surface of land area within the region (in km$^2$ and % of total surface), total surface (in km$^2$ and % of land surface) of areas with SI values above thresholds (0.5 for surfaces with at least "higher" dust emission potential, 0.7 for surfaces with at least "high" dust emission potential, and 0.9 for surfaces with "highest" dust emission potential) in maximum (max) and minimum (min) seasonal values; values are derived from UNCCD G-SDS-SBM.**

| NORTH HLD REGION (NORTH OF 50°N) | | | | |
|---|---|---|---|---|
| total area (km$^2$) | | land area (km$^2$) | | land area (%) |
| 64392015 | | 34695710 | | 54 |
| | max | | min | |
| | surface area (km$^2$) | surface area (%) | surface area (km$^2$) | surface area (%) |
| SI ≥ 0.5 | 1364799 | 3.9 | 1916 | 0.006 |
| SI ≥ 0.6 | 803372 | 2.3 | 1053 | 0.003 |
| SI ≥ 0.7 | 509965 | 1.5 | 718 | 0.002 |
| SI ≥ 0.8 | 342913 | 1.0 | 562 | 0.002 |
| SI ≥ 0.9 | 233336 | 0.7 | 451 | 0.001 |

**ARCTIC REGION (NORTH OF 60°N)**

| total area (km$^2$) | | land area (km$^2$) | | land area (%) | |
|---|---|---|---|---|---|
| 36876709 | | 18853826 | | 51 | |
| | **max** | | | **min** | |
| | surface area (km$^2$) | surface area (%) | surface area (km$^2$) | surface area (%) | |
| SI ≥ 0.5 | 1035059 | 5.5 | 515 | 0.003 | |
| SI ≥ 0.6 | 665082 | 3.5 | 350 | 0.002 | |
| SI ≥ 0.7 | 440804 | 2.3 | 297 | 0.002 | |
| SI ≥ 0.8 | 303521 | 1.6 | 264 | 0.001 | |
| SI ≥ 0.9 | 208701 | 1.1 | 217 | 0.001 | |

**Table 3. Relevant surfaces for the south HLD region: surface of total area of the region, surface of land area within the region (in km$^2$ and % of total surface), total surface (in km$^2$ and % of land surface) of areas with SI values above thresholds (0.5 for surfaces with at least "higher" dust emission potential, 0.7 for surfaces with at least "high" dust emission potential, and 0.9 for surfaces with "highest" dust emission potential) in maximum (max) and minimum (min) seasonal values; values are derived from UNCCD G-SDS-SBM.**

**SOUTH HLD REGION (SOUTH OF 40°S)**

| total area (km$^2$) | | land area (km$^2$) | | land area (%) | |
|---|---|---|---|---|---|
| 61435208 | | 1367987 | | 2 | |
| | **max** | | | **min** | |
| | surface area (km$^2$) | surface area (%) | surface area (km$^2$) | surface area (%) | |
| SI ≥ 0.5 | 309520 | 22.6 | 186266 | 13.616 | |
| SI ≥ 0.6 | 151480 | 11.1 | 81522 | 5.959 | |
| SI ≥ 0.7 | 61527 | 4.5 | 29256 | 2.139 | |
| SI ≥ 0.8 | 25416 | 1.9 | 10842 | 0.793 | |
| SI ≥ 0.9 | 8630 | 0.6 | 2747 | 0.201 | |

### 3.3 Emission and deposition of global and Arctic dust

The SILAM model estimated the total emission of annual dust and its deposition (data for 2017) onto snow-covered land, frozen sea, and total sea surfaces (frozen and non-frozen) (Fig. 4). The computations were also performed for Arctic dust and total global dust, with results for overall dust (diameter less than 30 μm) and fine dust (diameter less than 2.5 μm) separately (Fig. 15 of Section 3.5). Based on the model, the total emission of Arctic dust equals approximately 1.0% of the globe's total

dust emission. The deposition of Arctic dust onto snow- and ice-covered surfaces equals about 19% of the total dust deposition
onto these areas and around 57% of the deposition onto the areas explicitly located in the Arctic region. For fine dust, the
corresponding figures are 7% and 22%. Compared to the deposition of black carbon (anthropogenic sources and wildfires
combined; Fig. 15 of Section 3.5) onto snow and ice, the deposition of fine Arctic dust is about 70% higher globally and around
580% higher in the Arctic regions. While these figures provide a general quantification of the deposited amounts, detailed
calculations of the thermal and optical properties of dust and black carbon deposited on snow would be required to compare
the deposited substances' net impacts on the climate.

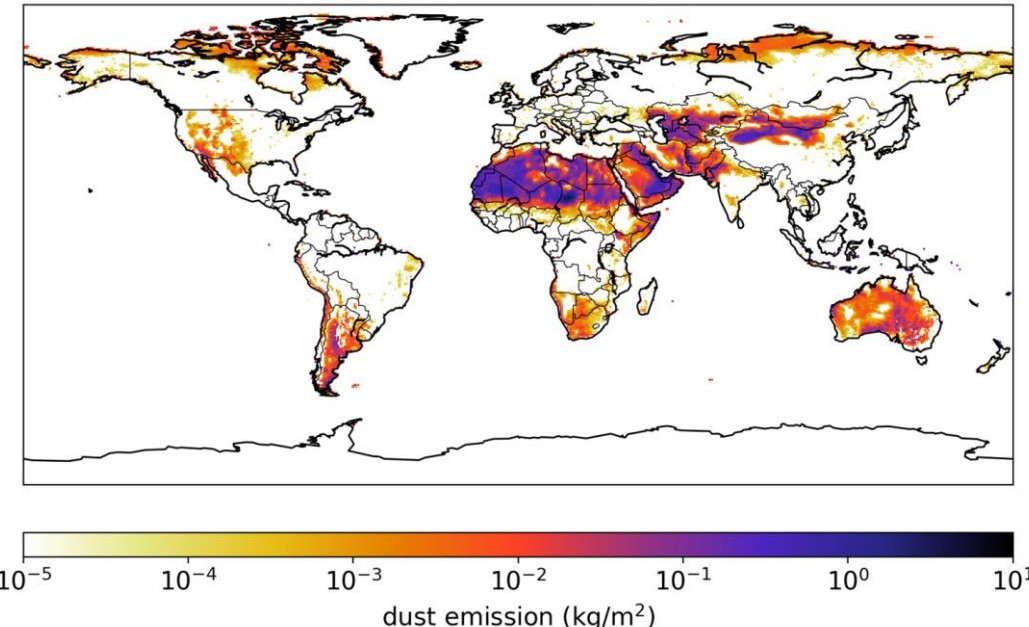

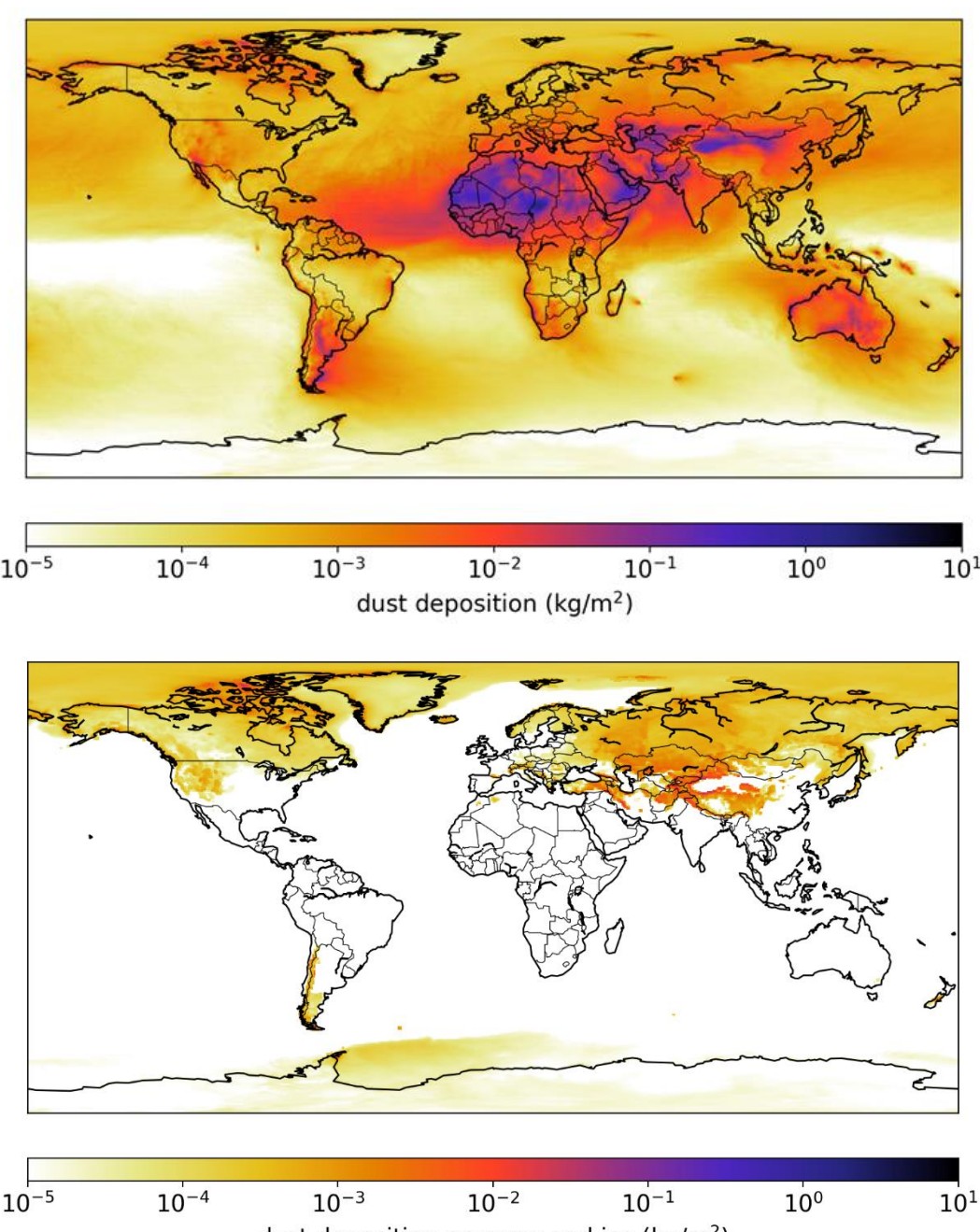

**Figure 4. SILAM emission and deposition modeling results of dust emission (above), dust deposition (middle) and dust deposition on snow and ice (below), in [kg/m²].**




## 3.4 The identified dust sources

Observations of the identified sixty-four dust sources in our collection (Fig. 1) are presented and discussed in alphabetical order as follows: 1. Alaska (sources no. 14 and 50 in Fig. 1); 2. Antarctica (no. 9, 19, 20, 52); 3. Canada (no. 2, 16, 48, 62); 4. Denmark and Sweden (no. 1, 15, 51); 5. Greenland (no. 53–61, 64); 6. Iceland (no. 23–45); 7. Russia (no. 2–11); 8. South America and Patagonia (no. 17, 21, 46, 47, 49, 52, 63); and 9. Svalbard (no. 13, 18, 22). Dust events originating simultaneously from Greenland, Iceland, and northern America are demonstrated in the Supplementary animation. The numbers are the identified 64 dust sources shown in Figure 1. Additional information, including latitude, longitude, and SI values, can be found in Supplement (Tables S1-S4).

### 3.4.1 Alaska, Copper River Valley, USA

Alaskan dust sources were identified over a century ago (Tarr and Martin, 1913). However, limited satellite detection due to abundant cloud cover and isolated location resulted in sparse information on this region (Crusius et al., 2011). The main identified sources are piedmont glaciers (Malaspina, Bering), resuspension of ash from past eruptions (Hadley et al., 2004), and major rivers carrying glacial sediment (Copper, Yukon, Tanana, and Alsek) (Gassó, 2020a,b; 2021a,b). Resuspension of glacial dust transported by these rivers can be abundant, often triggering air quality alerts by the Alaska Department of Environment (USGCRP, 2018). The largest and most active of such dust sources is the Copper River (Fig. 5), estimated to transport 69 million tons of suspended sediment annually (Brabets, 1997). Transported sediment is deposited on the Copper River Delta, an alluvial floodplain covering an area of 2800 km$^2$. When conditions allow, sediment is resuspended, resulting in dust plumes that can extend hundreds of kilometers over the Gulf of Alaska. Dust events, often lasting several days or weeks (Schroth et al., 2017), are most common in late summer and autumn when the river discharge and snow cover are at their minimum and high wind speeds are commonplace (Crusius, 2021). However, these occurrences have been observed year-round (Gassó, 2021a; January 2021). Dust reaches the open waters beyond the continental shelf and the influence of coastal sediments (Crusius et al., 2017). Thus, it has been proposed that dust from coastal sources such as the Copper River Delta can be an important source of bioavailable iron in the Gulf of Alaska (Crusius et al., 2011; Crusius, 2021; Schroth et al., 2017). Further work is also needed to investigate the relative importance of dust emissions from Alaska and East Asia (Bishop et al., 2002) in other areas. Also, dust from this region may initiate ice production in supercooled clouds, which is crucial for climate feedback (Murray et al., 2021). Regarding the magnitude and seasonal variability of emissions of sources in southern Alaska, a few dedicated studies have focused on dust from the Copper River Delta (Crusius, et al., 2017; Schroth et al., 2017; Crusius, 2021). However, to our knowledge, no dust activity and source characterization has been carried out along the coast of the Gulf of Alaska. Moreover, resuspended road dust is a major air quality issue throughout Alaska.

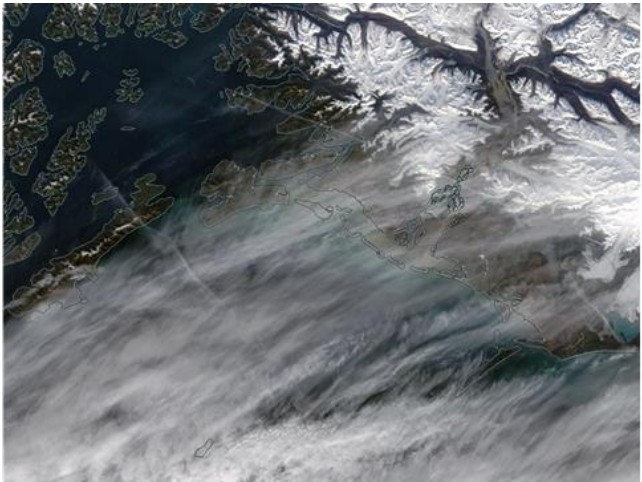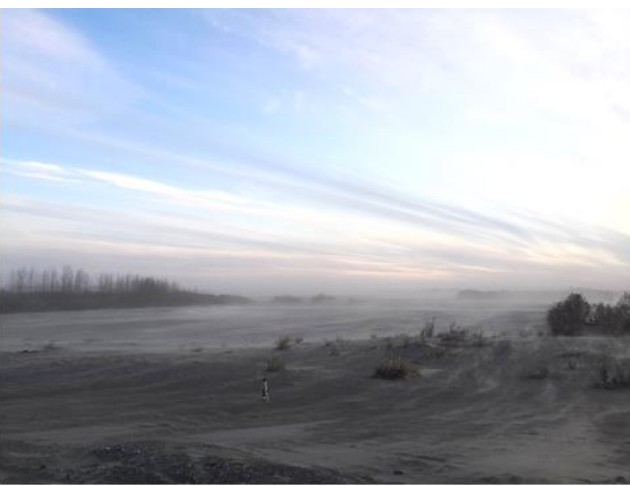

**Figure 5. Satellite image (left) of the Copper River region and photo (right) taken at the Copper River Delta on the same day (14th October 2019). The common occurrence of clouds prevents directly viewing the dust in suspension, illustrating the difficulty of observing dust activity from space. (Satellite image from NASA Worldview; photo by Sarah Barr).**

**3.4.2 Antarctica**

**3.4.2.1 James Ross Island, Ulu Peninsula**

The northern part of James Ross Island—Ulu Peninsula—represents one of Antarctica's largest ice-free areas (312 km$^2$). Its bare surface, consisting mainly of weathered sedimentary rocks, is an active HLD source (Kavan et al., 2018). Suspended sediments originate from outside the local fluvial systems based on the elemental ratios of Sr/Ca and Rb/Sr (Kavan et al., 2017). The wind speed threshold of 10 ms$^{-1}$ is needed for activating local dust sources, with most of the particles captured (by mass) in size bins between 2.5–10 µm. Mean (median) mass concentrations of the PM10 were 6.4 ± 1.4 (3.9 ± 1) µg m$^{-3}$, while the PM2.5 was 3.1 ± 1 (2.3 ± 0.9) µg m$^{-3}$ for the whole measurement period from January to March 2018. Mean PM10 values are comparable to background stations in Northern Europe. The highest daily aerosol concentration was 57 µg m$^{-3}$ for PM10, with hourly PM10 with > 100 µg m$^{-3}$. Higher aerosol concentration occurs in late austral summer when the soil water content in the upper soil layer is significantly lower than in early summer. Long-range transport of dust originating in Patagonia was observed during aerosol measurements (Kavan et al., 2018). A higher proportion of long-range transported dust was found in snow pits on higher elevated glaciers compared to a higher proportion of locally transported dust in lower elevated glaciers (Kavan et al., 2020b). Kňažková et al. (2020) identified a redistribution of mineral material within the HLD source area in Abernethy Flats, impacting the local microtopography.

### 3.4.2.2 Marambio, Antarctic Peninsula

The Marambio Base (64°14'S, 56°37'W, 198 m a.s.l.) on Marambio Island, Graham Land, Antarctic Peninsula, is a member of the Global Atmosphere Watch (GAW) programme of the WMO, with personnel available year-round. This region has ice-free areas and cold desert soils (Cryosols) that can be seasonally susceptible to wind erosion and weathering. The removal of fine materials occurs mainly by wind action. The Finnish-Argentinian co-operative research in Marambio includes measurements of ozone, solar irradiance, aerosols, and ultraviolet (UV) albedo (Aun et al., 2020). The UV Biometer Model 501 from Solar Light Co. (SL501) UV albedo data of 2013–2017 in Marambio were used to analyze the effects of local HLD on measured snow UV albedo and solar UV irradiance and differences in simulated UV irradiances (Meinander et al., 2018; data not presented here). For validating the UV albedo data, surface photos were taken regularly. The surface photos and UV albedo measurements show that local dust can be detected on the snow and ice. Also, the optical dome of the SL-501 sensor was found to be sandblasted by the windblown dust when returning to Finland for maintenance. These findings suggest that in Marambio, local dust can decrease surface snow/ice albedo, possibly enhance the cryosphere melt, and contribute to warming in the Antarctic Peninsula due to the ice-albedo feedback mechanism.

### 3.4.2.3 McMurdo Sound, Antarctica

The McMurdo Sound area of the Ross Sea region is widely recognized as the dustiest place in Antarctica, where locally sourced aeolian accumulation is up to two to three orders of magnitude above global background and dust fallout rates for the continent (Chewings et al., 2014; Winton et al., 2014). The area includes the McMurdo Dry Valleys (MDV), the largest ice-free area (4 800 km$^2$) in Antarctica. The MDV has high but extremely variable fluxes of locally derived aeolian sand (e.g., Speirs et al., 2008; Lancaster et al., 2010; Gillies et al., 2013; Diaz et al., 2020) and common aeolian landforms. Such has led to the assumption that the MDV is a significant regional dust source (e.g., Bullard, 2016). Some modeling studies suggest the MDV could supply large volumes of dust to a wide area of the Southern Ocean (e.g., Bhattachan et al., 2015). However, field-based observations show that very little sediment is transported out of the MDV (Ayling and McGowan, 2006; Atkins and Dunbar, 2009; Chewings et al., 2014; Murray et al., 2013) because the valleys have already been extensively winnowed into a well-developed deflation surface and large coastal piedmont glaciers form a topographic barrier, preventing aeolian sediment from escaping. The dominant source of aeolian sediment in the McMurdo Sound area is the debris-covered surface of the McMurdo Ice Shelf (1500 km$^2$), with minor contributions from local ice-free headlands. This ice shelf is unusual because it has high surface ablation and a continuously replenishing supply of fine-grained sediment advected from the seafloor. The sediment is blown off the ice shelf by frequent intense southerly wind events, forming a visible sediment plume onto coastal sea ice. Within a few km of the ice shelf, accumulation rates on sea ice are up to 55g m$^{-2}$yr$^{-1}$, reducing rapidly downwind to an average of 1.14 g m$^{-2}$ yr$^{-1}$, equating to 0.6 kt yr$^{-1}$ of aeolian sediment entering McMurdo Sound annually (Atkins and Dunbar, 2009; Chewings et al., 2014). Some sediment is transported at least 120 km from the source and could travel much farther, contributing iron-rich dust to the Ross Sea (Winton et al., 2014). Coastal areas and lowland parts of the MDV are on the

threshold of climatically driven change with observed increases in ablation and seasonal meltwater flow incising into
permafrost (Fountain et al., 2014), suggesting the dust potential of McMurdo Sound and MDV could rapidly change. The
McMurdo Dry Valleys (4800 km$^2$) is estimated to best fit Category 3 (source with unknown activity, Table S1). The McMurdo
Ice Shelf 'debris bands' are estimated to best fit Category 2 (moderately active source).

### 3.4.2.4 Schirmacher Oasis, East Antarctica

The Schirmacher Oasis (70° 45′ 30″ S, 11° 38′ 40″ E) is approximately 80 km from the coast of Lazarev Sea, Queen Maud
Land, East Antarctica. The oasis is an ice-free area of over 35 km$^2$ with typically hillocky relief. The oasis and surrounding
area have been explored since the early 1960s. However, no systematic studies of dust on local ice and snow have been done.
Most of this region's dust is assumed to be formed with the soils blown in the air because of strong winds. Human activity
produces some of the dust in this region: The oasis shelters four bases, which use diesel oil and petrol to supply heat and
transport operations. Two airports are nearby, which operate during the summer—lasting from late November to late February.
In December 2019, we collected the snow samples on eleven sites near the local ice roads, bases, and airports. These data will
contribute to our future study.

### 3.4.3 Canada

### 3.4.3.1 Lake Hazen, Ellesmere Island

Evidence of dust activity in Canada has been reported, e.g., in the prairie, crater lake, and river valley environments (e.g.,
Wheaton et al., 1990; Neuman, 1990; Wheaton, 1992; Hugenholtz and Wolfe, 2010; Fox et al., 2012). Satellite observations
of high-latitude dust events over water are relatively common (see, for example, Bullard et al., 2016). Whether directly
concerning explicit plume remote sensing or indirectly regarding plume deposition, the detection of such events has remained
largely unreported. Ranjbar et al. (2021) recently reported detecting a drainage-flow induced dust plume over (frozen) Lake
Hazen, Nunavut, Canada, using a variety of remote sensing techniques (Lake Hazen is the Arctic's largest lake, by volume, at
81.8°N latitude in the northernmost portion of Ellesmere Island). Figure 6 shows a true-color georeferenced RGB MODIS-
Terra image acquired on 19 May 2014 at 19:50 UT (15:50 EDT) over Lake Hazen. The authors employed MISR stereoscopy,
CALIOP, and CloudSat vertical profiling, as well as MODIS thermal IR techniques, to identify and characterize the plume as
it crossed over a complex springtime terrain of snow, ice, and embedded dust. While limited by the lack of dedicated dust
remote sensing algorithms over snow and ice terrain, the plume characterization boded well for developing systematic,
satellite-based, high-latitude dust detection approaches using current and future generations of aerosol and cloud remote
sensing platforms.

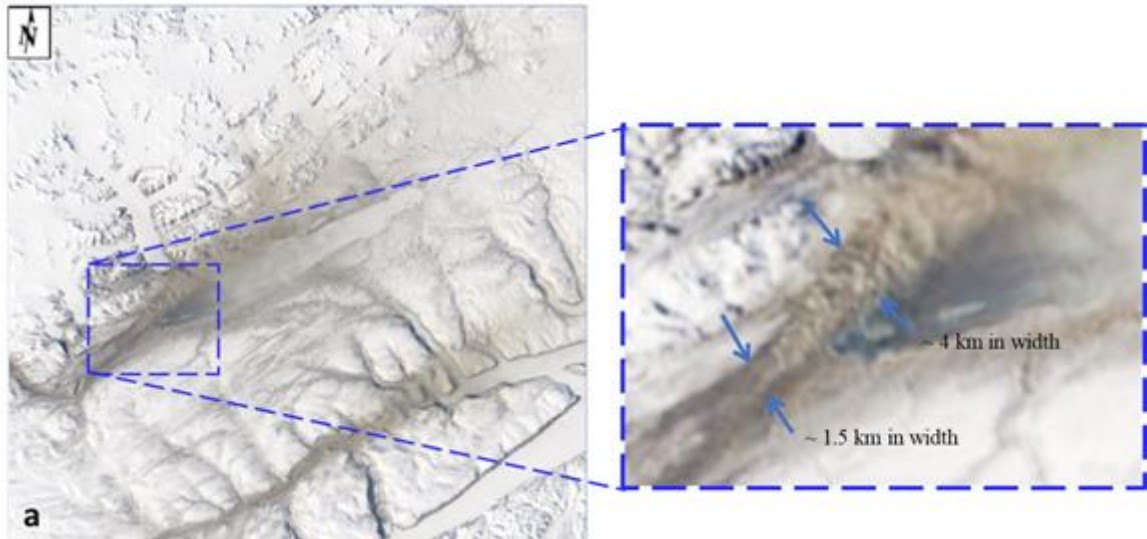


**Figure 6. MODIS-Terra satellite image on 19 May 2014 at 19:50 UTC (a) True-color image: MODIS channels 1 (620–670nm), 3 (459–479 nm), and 4 (545–565 nm) were loaded into the RGB channels of the display. The sub-image is a zoom of the most discernible part of the plume (outlined by the blue broken-line square).**

### 3.4.3.2 Kluane Lake, Yukon

Within the St. Elias Mountain range at the north end of the Pacific Coast Range on the continental side of the Yukon Territory lies the Kluane Lake region (KLR), which contains Łhù'ààn Mân' (Kluane Lake) (no. 50 in Fig. 1). The lake is fed primarily from the meltwater of the Kaskawulsh glacier down the A'äy Chù (formally the Slims River) and snowmelt from the surrounding regions in the springtime. This seasonal discharge has, in recent history, been known to be highly variable as the glacier terminates at the fork of two distinct watersheds—one draining into the Bering Strait through the Yukon River and the other into the Gulf of Alaska—supplying the two watersheds' inconstant ratios. In 2016, most of the glacier's discharge was diverted to the Gulf of Alaska in an intense discharge event, dramatically decreasing the Łhù'ààn Mân's water levels and increasing the dust emission potential from the A'äy Chù (Shugar et al., 2017). This drastic change makes the KLR an excellent natural laboratory for investigating the impact of pro-glacial hydrology on dust emission potential under past and future climates. Research was conducted in the early 1970s in this same valley as a comprehensive set of dust flux measurements as part of several publications (Nickling, 1978; Nickling and Brazel, 1985). Nickling (1978) concluded that there is a dynamic relationship between soil moisture (driven by precipitation and nighttime radiation insolation) and wind, resulting in periodicity of dust emissions from the valley in all but the mornings throughout the snow-free seasons. Within a more recent study by Bachelder et al. (2020), soil and aerosol samples were collected within the Ä'äy Chù delta, where air quality thresholds were exceeded, indicating a negative impact on local air quality throughout May. Notably, daily particle size distributions of PM10

were very fine (mode of 3.25 µm) compared to those measured at more well-characterized, low-latitude dust sources. Moreover, mineralogy and elemental composition of ambient PM10 were found to be enriched in trace elements (e.g., As and Pb) compared to dust deposition, bulk soil samples, and fine soil fractions (d < 53 µm). Finally, through a comparison of the elemental composition of PM10, dust deposition, and fine and bulk soil fractions, as well as meteorological factors measured, Bachelder et al. (2020) propose that the primary mechanisms for dust emissions from the Ä'äy Chù are the rupture of clay coatings on particles and the release of resident fine particulate matter.

### 3.4.4 Denmark and Sweden

In Denmark, large areas with severe wind erosion have been documented (Kuhlman, 1960). Published literature on the activity of dust sources in Denmark is rare; some documentation is only in Danish. On 23 April 2019, a dust plume from Denmark's west coast, with dust plumes from Sweden 12 km long Mellbystrand around the mouth of the Lagan River (no. 51 in Fig. 1); Poland could be observed in Meteosat-11 Dust RGB and Natural Colour images, 23 April 12:30 UTC. These dust plumes were observed to travel to the North Sea (Meteosat, 2019). The source in Denmark appears to be from Holmsland Dunes (no. 15 in Fig. 1). Other potential dust sources in Denmark include, e.g., the Råbjerg mile (no. 1 in Fig. 1), the largest moving dune in Northern Europe with an area of around 2 km$^2$ (Doody et al., 2014), located between Skagen and Frederikshavn. Råbjerg Mile moves at approximately 15 meters per year due to wind and has moved around 1.5 km further east in the last 110 years. The drifting sand is not considered to be transported very far. In general, dust storms in Denmark are considered small, and locally based dust storms can be expected when farmers prepare the arable soils in spring, creating dust in case of a very dry April month. In Tilviden, flying sand took over (after King Frederik II cut the oak trees for building ships in 1600). Also, a regional soil and sand event in Denmark, reportedly common in April, was recently documented between Mejrup and Holtebro on 6 April 2021 (Television Midtvest, 2021; not identified in Fig. 1; coordinates are estimated as 56°23'N, 8°41'E). This location between Mejrup and Holtebro remains to be marked as a potential dust source for future observations. The event was observed over roadways in several parts of the region, reducing visibility due to a long period without rain and with strong winds for > 24 hours, causing the soil to blow off the harrowed fields.

### 3.4.5 Greenland

Greenland's ice-free areas have long been identified as locally important dust sources (Hobbs, 1942), with dust storms described as reaching >100 m high (Dijkmans and Törnqvist, 1991). These storms can cause the darkening of the Greenland Ice Sheet by deposition, which may affect albedo and rates of ice melt (Wientjes et al., 2011; McCutcheon et al., 2021). Potential dust source areas in Greenland are mapped in the recently issued global dust atlas by A. Vukovic (UNCCD, 2021). Dust input to soils and lakes may also have substantial ecological impacts (Anderson et al., 2017). Bullard and Mockford (2018) investigated the seasonal and decadal variability of dust emissions in southwest Greenland and presented the first long-term assessment of dust emissions. Dust emissions occur all year but peak in spring and early autumn. The evidence linking increased dust emissions to preceding jökulhlaup (a type of glacial outburst flood) events is inconclusive, requiring further

exploration. The decadal record confirmed that dust-storm magnitude may have increased from 1985 to the 1990s (Bullard and Mockford, 2018). Amino et al. (2020) also showed that dust deposition on the southeastern dome in Greenland has increased in recent decades. They link this increase to dust emissions in coastal Greenland, where snow cover is decreasing. However, further work is needed to characterize the magnitude of dust events at the source and how their emissions are changing. Bullard and Mockford (2018) also presented preferential dust-event pathways from Kangerlussuaq, indicating that most events travel toward the Davis Strait and the Labrador Sea, where the dust might impact boundary layer of mixed-phase clouds (Murray et al., 2021).

Modern satellite remote sensing methods can detect dust storm events in Greenland's different valleys and coastal areas. The new HLD sources identified in this study based on satellite observations are in Supplementary Table S3. Figure 7 illustrates one such dust storm episode on the Nuussuaq Peninsula, Greenland, on 1 October 2020 (Markuse, 2020). One example of DREAM regional-scale modeling of atmospheric transport of dust from Greenland potential dust sources is demonstrated in Figure 8 (animation available in Supplementary), where the DREAM circumpolar prediction experiment example shows the predicted surface dust concentration for 4 November 2013 and Icelandic volcanic desert dust to reach Greenland, as discussed, e.g., in Meinander et al. (2016).

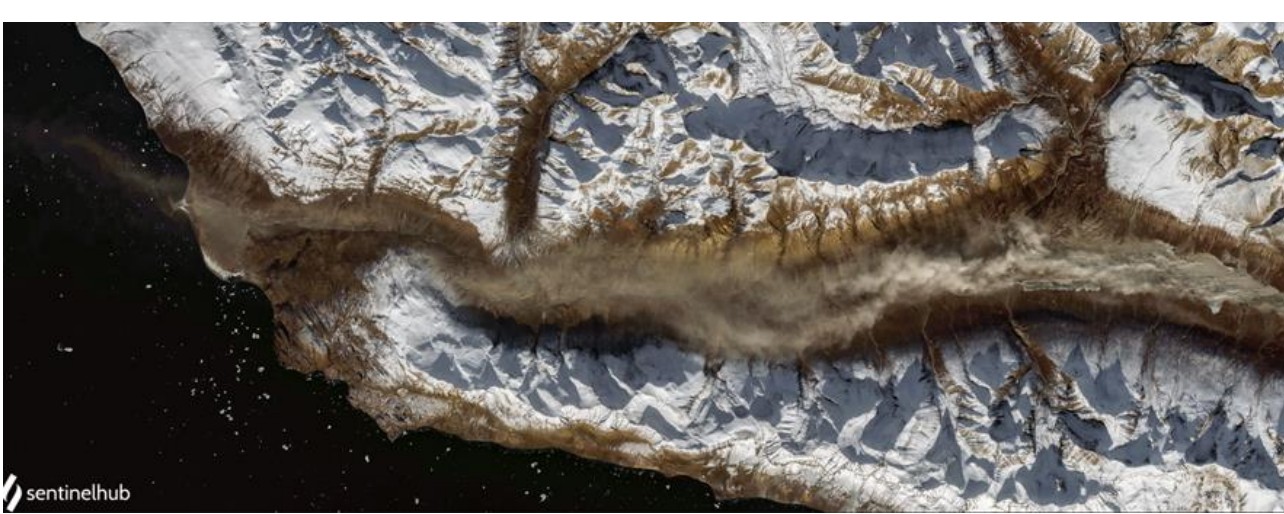

**Figure 7. High-latitude dust storm on the Nuussuaq Peninsula, Greenland – 1 October 2020 (Markuse, 2020; cc-by-2.0.2020)**

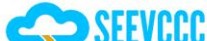

NMMB–DREAM8–circumpolar: Dust load (g/m²) and 10m wind streamlines
Forecast base time: 04NOV2013 00UTC   Valid time: 04NOV2013 20UTC

Figure 8. DREAM model predicted dust load for 4 November 2013 (animation available in Supplementary).

### 3.4.6 Iceland

Iceland has been recognized for a while as a potentially important dust source. In our collection, 13 new sources in Iceland were included (Table S2), compared to previous sources, in which eight Icelandic dust hot spots were identified (Arnalds et al., 2016). Sandkluftavatn, Kleifarvatn, Skafta jökulhlaup deposits and other areas have also been found to produce large amounts of dust (Dagsson-Waldhauserová et al., 2019). In recent years, increased dust activity has been reported in Flosaskard and Vonaskard (Gunnarsson et al., 2020). These dust hotspots cover almost 500 km$^2$, while deserts are over 45 000 km$^2$ (Arnalds et al., 2016). Most of the dust hotspots are near glaciers: glacial floodplains, old lakes, jökulhlaup (a type of glacial outburst flood) deposit areas, or sandy beaches. Glacio-fluvial plains receive a massive amount of unconsolidated silty material during the melting of nearby glacial regions.

New dust sources with the number of events are identified here and presented based on satellite image observations from 2002
to 2011 (Supplementary Table S2), suggesting that Iceland's entire southern coast could be considered one source. However,
previous results on Icelandic dust suggest that nearby locations may have different particle characteristics (Fig. 9). Therefore,
each source must be studied independently. For example, the grain size distribution curves of the samples from Dyngjusandur,
Hagavatn, Landeyjarsandur, Maelifellsandur, Myrdahlsandur, and Sandkluftavatn showed generally unimodal distributions
with a rather diverse character (average diameters ranging from 19.8 to 97.7 µm, Fig. 9). Richards-Thomas et al. (2021)
identified a range in particle diameter between 0.4 µm and 89 µm, with the medians (d50) of the distributions from 12 to 25
µm). Some hotspot particles are bimodal with peaks at 2 µm and 30 µm and a more significant proportion of the sample within
the silt-size range.

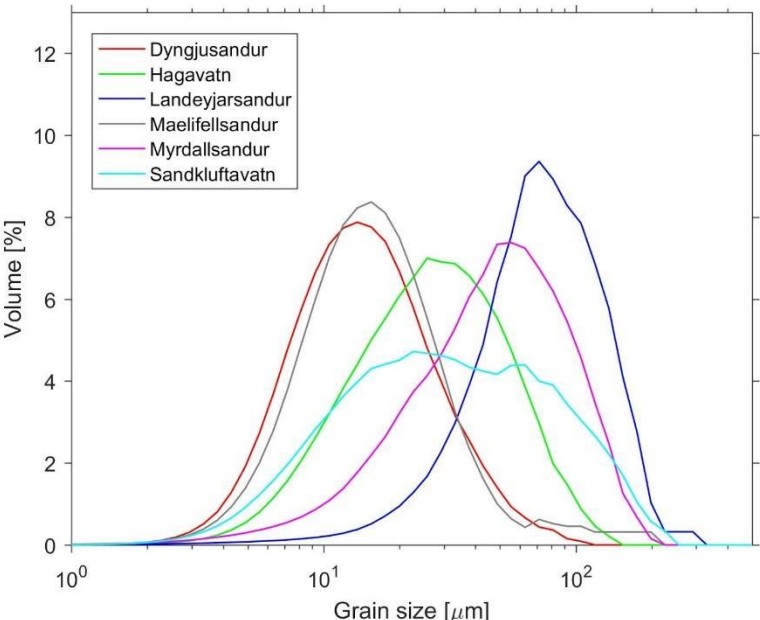


**Figure 9. Grain size distributions of samples from Icelandic source areas (redrawn from Varga et al., 2021)**
Icelandic dust particles have different shapes, lower densities, higher porosity, increased roughness, and darker colors than
other desert dust (Butwin et al., 2020; Richards-Thomas et al., 2021). Those greater than 20 µm retain the volcanic
morphological properties of fresh volcanic ash. Dust and fresh volcanic ash particles less than 20 µm are crystalline and blocky.
Icelandic dust particles contain amorphous glass, large internal voids, and copious dustcoats comprised of nano-scale flakes.
The amorphous basaltic material is mostly aluminosilicate glass ranging from 8 wt% (Hagavatn hotspot) to 60–90 wt%, with
relatively high total Fe with higher Fe solubility and magnetite fraction than low-latitude dust (10–13 wt%, Baldo et al., 2020).
PM10 concentrations measured during severe Icelandic dust storms well exceeded 7000 µgm$^{-3}$ (Dagsson-Waldhauserová et
al., 2014, 2015; Mockford et al., 2018). Submicron particles contribute with high proportions (> 50%) to PM10 mass
concentrations and number concentrations (Dagsson-Waldhauserová et al., 2014, 2016, 2019). Aeolian transport of 11 t of
dust over one meter transect was measured during the severe dust/ash storm in 2010, when grains > 2 mm were uplifted
(Arnalds et al., 2013).
As well as differences in Icelandic dust sources, the chemical composition of the aircraft-collected Icelandic dust particles has
a different chemical signature than, e.g., airborne Saharan dust particles transported to Barbados (Sanchez-Marroquin et al.,
2020). This difference can be observed in Figs. 10a and 10b, where the chemical composition of most Icelandic dust particles
falls in a different area of the chemical composition ternary diagram than the Saharan dust particles from Barbados. One of the
most prominent differences between these types of dust is Ti's presence in ~ 30% of the Icelandic dust particles, while this
element is almost absent in the Saharan dust particles and dust collected elsewhere, shown in Fig. 10c. Furthermore, the
chemical composition of the aircraft-collected Icelandic dust is consistent with surface scooped samples of dust or volcanic
ash from Iceland. Moreover, a droplet freezing-based assay confirmed that the sampled Icelandic dust has a high ice-nucleation
ability and can influence the radiative and lifetime properties of clouds containing water and ice.

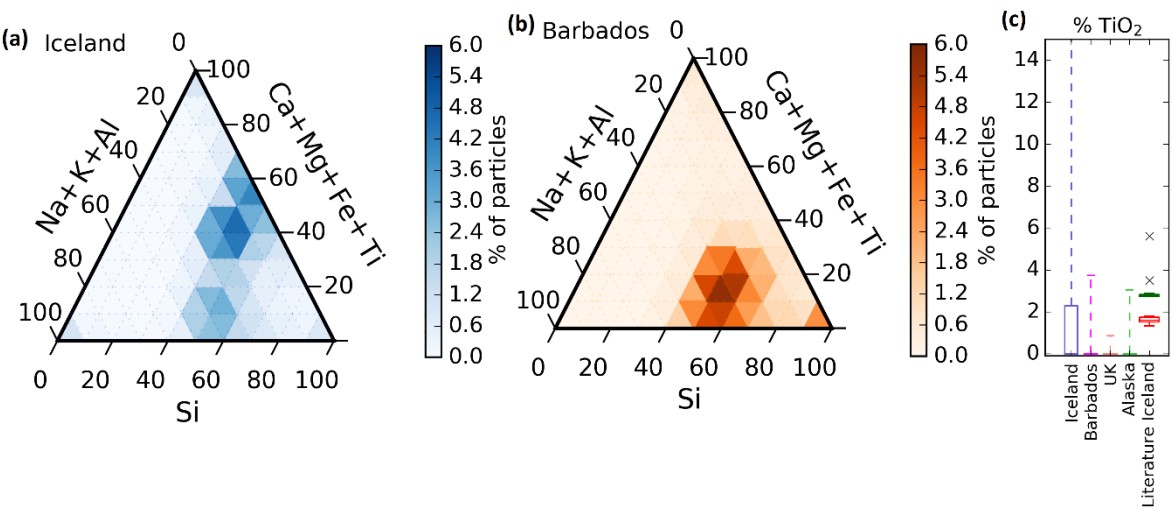


**Figure 10. Ternary graphs of the chemical composition of Icelandic dust particles (a) and Saharan dust particles collected in Barbados (b). Each graph contains a heat map with the percentage of dust particles in each sample compositional bin. The chemical composition of each aerosol has been recalculated from the weight percentages given by the SEM software, excluding elements that are not Si, Al, Fe, Mg, Ca, Na, K, Ti, Mn, and P. (c) The box represents particles in the Q3 percentile of the percentage of the composition of Ti in all the dust particles in each sample (Icelandic dust, Saharan dust collected in Barbados, dust collected in the UK, and dust collected in Alaska). The whiskers represent the composition of all particles between the median plus and minus two**

**standard deviations. The data has been compared with the Ti weight percentage of different Icelandic dust and ash samples from**

**the literature. (Figure extracted from the Supplementary Material of Sanchez-Marroquin, 2020).**

No direct observations or measurements of the new sources were available. Instead, two model computations are presented for
Iceland because of the lack of observations and complexity of the AOD interpretation in polar and subpolar regions. Without
high uncertainty of direct measurements, the importance of the HLD modeling rises; models validated over better-observed
regions may become an important or primary source of information. Results using the DREAM model, with a horizontal
resolution of ~3.5 km, were used here to resolve the heterogeneous and small-scale character of the Icelandic dust sources
(Fig. 11). As the first operational numerical HLD model, DREAM-ICELAND predicted the Icelandic dust for the example
case of 18 September 2020 (Fig. 11).


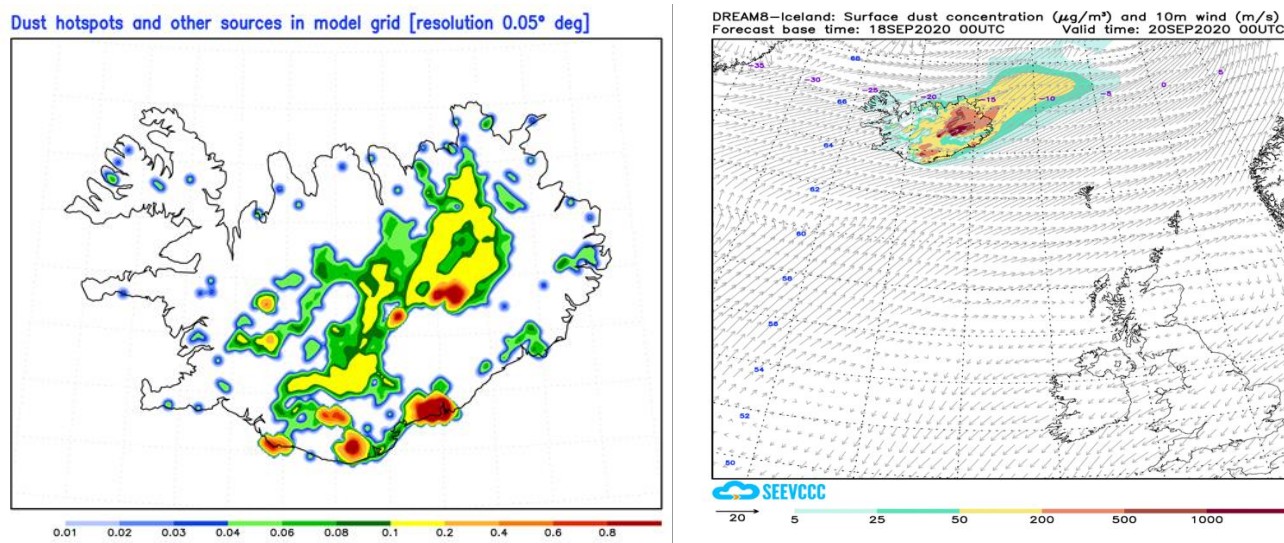


**Figure 11. Left panel: dust sources in DREAM-ICELAND model grid with areas vulnerable to erosion and containing hot spots**
**(Arnalds et al., 2016). Right panel: An example of the operational Icelandic dust surface concentration forecast for 18 September**
**2020 (available at the Republic Hydrometeorological Service of Serbia site, http://www.seevccc.rs/?p=8).**

In Figure 12, dust emissions in Iceland are presented in three-months periods for March 2020–August 2021. The modeled
results clearly show the seasonal nature of the dust sources. The summer season (June–August) appears to be the strongest
dust season. However, there are also dust emissions in wintertime with snow-covered land surfaces, according to observations
of dust events during snowfall (e.g., Dagsson-Waldhauserová et al., 2015). The 2021 summer season in these modeled emission
results appears in the same locations as summer 2020 but with more severe emissions in the highlands in 2021, agreeing with
the field observations in Vatnajökull national park during the HiLDA measurement campaign in the 2021 season
(https://gomera.geo.tu-darmstadt.de/wordpress/), where the most severe dust events were measured.

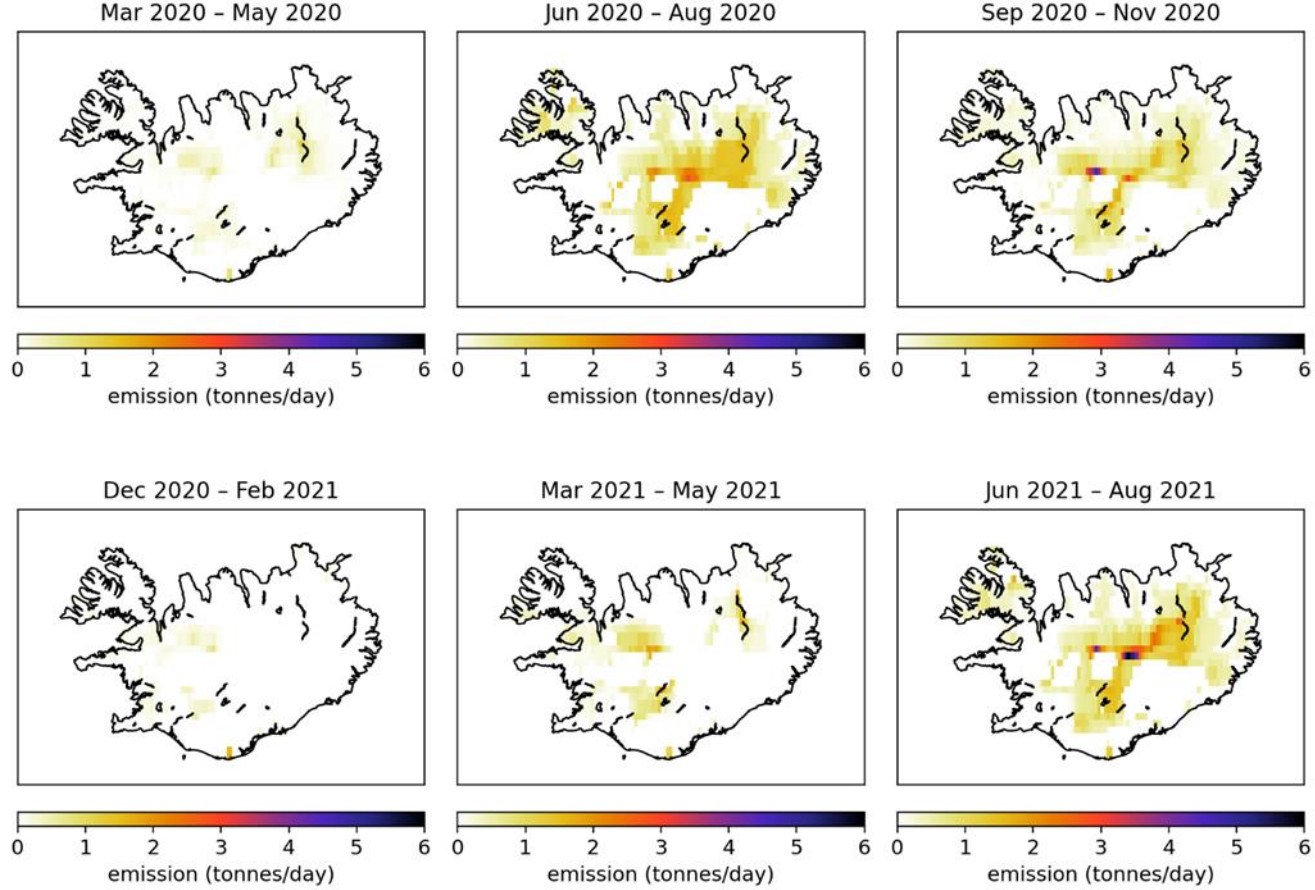

**Figure 12. SILAM modeled dust emissions (tonnes/day) in Iceland for three months periods in March 2020 – August**
**2021.**


**3.4.7 Russia**
The Russian Arctic and subarctic are the most relevant regions connected to the HLD sources. In these territories, atmospheric
dust is produced due to burning gas (Novy Urengoy is named the gas capital of Russia) and forest fires (especially in Siberia;
see MODIS or Sentinel images for Novy Urengoy on 3-8 August 2021), dusting of abandoned and non-reclaimed heaps).
Wind erosion is followed by vegetation destruction from gas and oil extraction, especially in Western Siberia. Some Russian

sources included in our collection (e.g., no. 7 and 8 of Fig. 1) could be identified as dust sources on the periphery of HLD and low-latitude source regions. Source no. 7 of Fig. 1 is the Altai Mountains. Some parts of these territories are covered by permafrost, where winter lasts for 5–6 months. From October, in lower mountains (less than 1000 m a.s.l.), and from September, in higher mountains (more than 1500 m a.s.l.), a stable snow cover persists. The mean daily air temperature during winter within the lower, middle, and higher mountains is –21°C, –29°C, and below –30°C, respectively. Source no. 8 is in Central Kazakhstan. From late December to early March, a stable snow cover from 5 cm to 30 cm occurs within plains and up to 50 cm within hollows. Periods of snow cover and thaw correspond to transitions of the mean daily temperature of air through 0°C, which, on average, are the 7 November and 23 March plus/minus 10–12 days. From early January to late February, the air's mean daily temperature can be as low as –20°C. Soil Atlas of the Northern Circumpolar Region (https://esdac.jrc.ec.europa.eu/content/soilatlas-northern-circumpolar-region) covers all land surfaces in Eurasia and North America above the latitude of 50°N. Thus, these territories are considered high-latitude.

### 3.4.7.1 Western Siberia, Altai Mountains, and Central Kazakhstan

In the most widespread undisturbed soils (Gleysols, Phaeozems, Podzols, Retisols, and Stagnosols) in Western Siberia (Semenkov et al., 2015a, 2015b)—the vastest plain in the world—mineralogical and elemental composition (Supplementary Table S6) were studied using X-ray diffractometry, X-ray fluorescence spectrometry, ICP-MS, ICP-AES, and content of total organic carbon (TOC), as reported in detail in (Semenkov et al., 2019; Semenkov and Koroleva, 2019; Semenkov and Yakushev, 2019). At locations no.7 and no. 8 of Fig. 1 (Table 4), the concentration of N-containing substances, pH values, dust content and dust deposition rate were measured in snow in winter from 2009 to 2019 (Koroleva et al., 2016, 2017; Semenkov et al., 2021; Sharapova et al., 2020).

**Table 4. Major ions (mg/L), pH value, dust content ($mg/m^2$ in snow), and deposition rate ($mg/m^2/d$) during winter at HLD sources no. 7 and no. 8 in Fig. 1.**

| HLD no | M | SD | Me | Min | Max | N |
|---|---|---|---|---|---|---|
| No. 7 | | | | | | |
| Dust content $mg/m^2$ | 316 | 439 | 112 | 0 | 1542 | 30 |
| $NH_4^+$ mg/L | 0.75 | 0.98 | 0.30 | 0 | 3.60 | 43 |
| $NO_2^-$ mg/L | 0.015 | 0.019 | 0.008 | 0 | 0.08 | 107 |
| $NO_3^-$ mg/L | 2.3 | 3.4 | 1.4 | 0 | 20.4 | 118 |
| pH | 6.6 | 0.8 | 6.7 | 4.1 | 8.4 | 129 |

| No. 8 | | | | | | |
|---|---|---|---|---|---|
| Dust deposition rate mg/m$^2$/d | 1.67 | 1.67 | 1.08 | 0.05 | 6.6 | 38 |
| NH$_4^+$ mg/L | 0.20 | 0.009 | 0.10 | 0 | 1.34 | 682 |
| NO$_2^-$ mg/L | 0.027 | 0.007 | 0 | 0 | 0.61 | 127 |
| NO$_3^-$ mg/L | 0.47 | 0.02 | 0.19 | 0 | 3.93 | 697 |
| pH | 6.1 | 0.02 | 6.1 | 4.6 | 8.0 | 585 |

M – mean, max – maximum, Me – median, min – minimum, N – number of observations, SD – standard deviation

**3.4.7.2 Murmansk region: Apatity, Kirovsk, Kovdor**
Large amounts of displaced rock have been breaking the balance of geological emissions of gas and dust from mining, dumps,
and tailing pits (e.g., Csavina, et al. 2012). Over 150 Mt of industrial wastes are disposed of in the Murmansk region annually,
achieving about 8 Gt (Supplementary Table S7). The dusting of processing tailing is one of the main sources of air pollution
resulting from suspended matters near the mining enterprises. About 30% of all suspended matter is released from the mining
enterprises into the atmosphere due to wind-induced dusting of beaches and slopes of tailings dumps. Elevated concentrations
of suspended matter are registered every summer in Apatity's atmosphere. Dust storms from technogenic dust sources of the
mining industry on the Kola Peninsula are presented, e.g., in Baklanov and Rigina (1998), Baklanov et al., (2012), and Amosov
and Baklanov (2015).
**3.4.7.3 Tiksi**
Aerosol characterization was performed at the Hydrometeorological Observatory (HMO) Tiksi (71.36N; 128.53E) on the coast
of the Laptev Sea in Northern Siberia from 2014 to 2016 (Popovicheva et al., 2019). FTIR analyses of functionalities and ionic
and elemental components provided insight into the dust source-influenced and season-dependent composition of East Siberian
Arctic aerosols. Analysis of wind and aerosol pollutants roses, with long-range transport analysis, helped identify the dust
sources at Tiksi, demonstrating impacts from lower latitudes or local emissions from the adjacent urban Tiksi area. In warm
periods, Na$^+$, Cl$^-$, K$^+$, and Mg$^{2+}$ are found to be the major ions in the sea-salt aerosols, which are ubiquitous in the marine
boundary layer, significantly impacting the dust concentrations in the coastal region. However, Cl$^-$ and K$^+$ could also originate
from biomass burning during the warm period. Ammonium is mainly produced by the soil and emission from biota and the
ocean, commonly found in the form of $(NH_4)_2SO_4$ and $NH_4Cl$. Like sulfates, ammonium is influenced by regional sources of
secondary aerosol formation and transport. Bands of carbonates $CO_3^{2-}$ (at 871 cm$^{-1}$) and ammonium $NH_4^+$ (3247 cm$^{-1}$) indicate
the dominance of dust carbonates in the natural inorganic aerosol. Also, S, Fe, Na, Al, Si, Ca, Cl, K, Ti, Mn, Co, Cu, Zn, Ga,
Sr, Ba, Hg, and Pb were detected in the background dust, with sulfur displaying the highest concentration, followed by Fe, Na,
and Al.
According to individual particle analyses by SEM-EDX, during the summer and autumn, when the wind comes from the
southwest and air masses arrive from the ocean, aerosol particles demonstrate a large variability in shapes, sizes, and
composition (Fig. 13.1). Elemental composition is characterized by a dominant weight percentage of C, K, Na, Cl, O, and Fe.
The distribution of elements over particles is heterogeneous, with greater amounts of Cl, K, and Na than C and O in around
50% of particles, indicating that background aerosols contain soil, salts, minerals, and carbonaceous compounds. Group Na-
rich with dominant Na and Cl is the most abundant at 32.5%, originating from sea spray near the ocean (Fig. 13.2). The other
particles contain small amounts of K, Ca, and Mg from seawater impurities, as well as S, gained through acid displacement.
The second most abundant group of individual particles is Group K-rich at 28.8%, dominated by K and Cl, which are not of
marine origin because the concentration of NSS K$^+$ ions significantly exceeds K's possible concentration in SSA. Instead,
Group K-rich particles are of natural mineral sylvite (KCl), transformed from genuine ones because the average weight ratio
of K/Cl was found to be equal to 3.3—significantly higher than 1.1—in sylvite (Fig. 13.3). KCl is water-soluble and may react
in a polluted atmosphere. The variation of wt% of K vs. Cl shows the lack of Cl compared to genuine sylvite and the formation
of complex chemical compounds $K_xCl_y$ with various K and Cl atoms. A representative micrograph of particles in Group K-
rich demonstrates the reacted sylvite in Fig. 13.3, with slight damage by an electronic beam that can prove the presence of
nitrates that were easily evaporated during EDX analyses. A part of Group Na-rich and K-rich, 20% and 5%, respectively,
contains Na, Cl, and K and is assumed to be particles comprised of natural sylvite from alternative layers of halite and sylvite
(nNaCl + mKCl) (Fig. 13.4). They have distinctive mineral shapes and are stable regarding evaporation by an electron beam.
About 14.8% of individual particles composed of Group Organic made almost exclusively from C and O. These particles are
roughly spherical or liquid-like shaped (Fig. 13.5): Around half contain only C and O, being probably secondary organic
aerosol from the biogenic source; the other half come from the seawater of the Arctic Ocean, as demonstrated by trace amounts
of Na, Cl, and Mg. The oxidation of volatile organic compounds, humic-like substances (HULIS) in the marine environment,
perhaps contributes to observed organic matter.
Finally, a few biogenic particles such as pollen, spore, algae, bacteria, and plant or insect remnants are found in natural aerosols,
indicated by the specific shape and presence of K, S, Si, and Cl with C. The remaining groups—Fe-rich (14.4%), Ca-rich
(6.4%), and Al, Si-rich (3%)—are representative of atmospheric dust derived from the Earth's crustal surface. Dust particles
have solid irregular shapes of round and euhedral morphology. Analyses of the soil sample taken near the CAF showed stony
material with minimal fertile ground cover. EDX analyses demonstrated 27.7 and 9.8 wt% of Si and Al, 46 and 10.6 wt% of
O and Fe, respectively, and 3.5 w% of K in various Fe,K—aluminosilicates containing small additives (less than 1.7 wt%) of

Na and Mg. Since the tiny dust of stony soil may be easily dispersed into the atmosphere by wind, we assume that Group Al, which is Si-rich, and around half of Group Fe-rich, is composed of Fe,K—aluminosilicates (Fig. 13.6). Group Fe-rich containing Fe, Ni, Ca, and Si is composed of soil particles of iron-nickel ore (Fig. 13.7). Finally, Ca carbonates and sulfates with Ca, C, S, and O are found in Group Ca-rich (Fig. 13.8), according to the observation of $Ca^{2+}$, $CO_3^{2-}$, and $SO_4^{2-}$ ions described above. With aluminosilicates, they are most likely windblown dust.

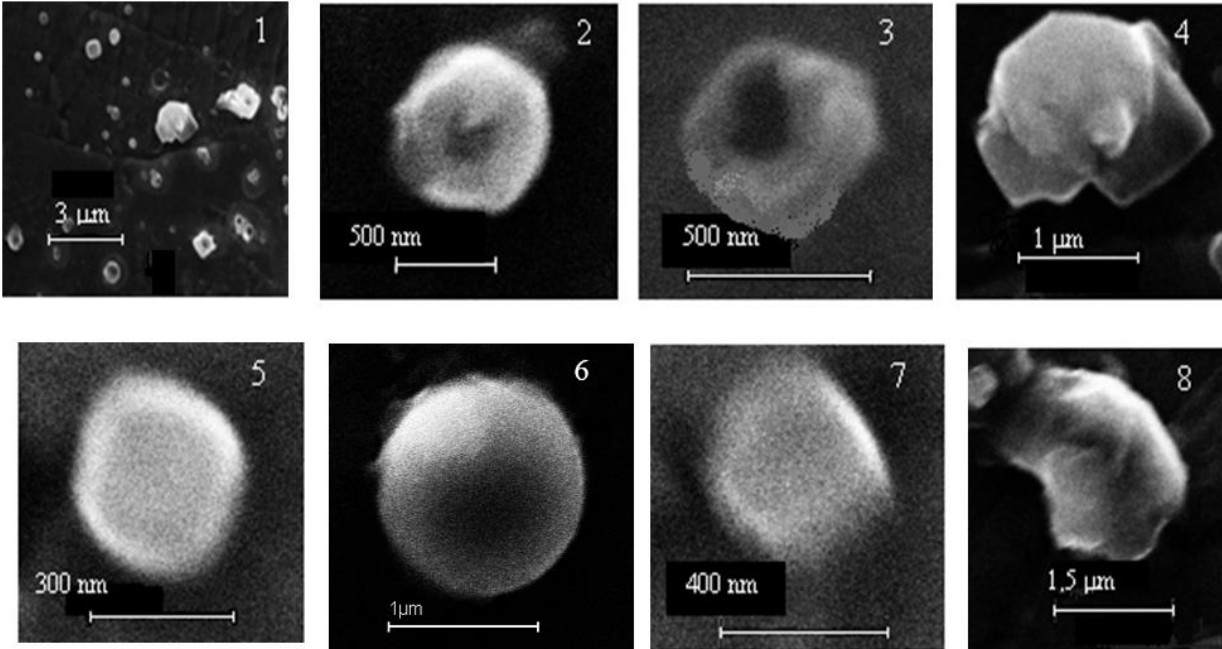

**Figure 13. 1. Panorama and representative micrographs of natural background aerosols at HMO Tiksi; 2. reacted sea salt NaCl in Group Na-rich; 3. reacted sylvite KCl and 4. sylvinite (nNaCl + mKCl) in Group K-rich; 5. an organic particle in Group Organic; 6. Fe, Ca- aluminosilicate in Group Al, Si-rich; 7. Fe/Ni particle in Group Fe-rich and 8. $CaCO_3$ in Group Ca-rich of natural aerosols on 27.09.2014. New unpublished results of Popovicheva et al. (2019) investigation.**

### 3.4.8 South America and Patagonia

Extending from 39ºS to 54ºS, with an area of 600 000 $km^2$, dust activity (Fig. 14) from this large desert remains largely unknown. Some basic facts must be formally assessed, such as the location of sources and geomorphological features associated with dust, as well as the seasonality and frequency of the dust's activity. To date, limited surveys of dust activity

(Crespi-Abril et al., 2017; Gaiero et al., 2003; Gassó and Torres, 2019) and case studies of individual sources exist (Gassó et
al., 2010; Gassó and Stein, 2007; Johnson et al., 2011). Recently, a list of dust activities and sources in Tierra del Fuego
(Cosentino et al., 2020) has been published. Generally, dust sources in Patagonia are at topographic lows, and the river valleys
(e.g., the Deseado and Santa Cruz rivers) (Coronato et al., 2017; Hernández et al., 2008) are associated with the late Holocene
para-glacial environments. The most active modern source of dust is the drying of Colhué Huapi Lake (CHL) in Central
Patagonia (45.5°S and 68°W) (Montes et al., 2017)—a shallow lake with variable water levels exposed to intense
evapotranspiration. An anthropogenic component appears to be linked to intense farming, oil prospection, and supplying water
to urban centers (Gaitán et al., 2009; Hernández et al., 2008; Mazzonia and Vazquez, 2009; Valle et al., 1998). CHL has been
steadily shrinking (Llanos et al., 2016) and was dried up by the summer of 2020. Consequently, dust activity originating in
CHL has increased with frequent blowouts large enough to be easily detected from space (Gassó and Torres, 2019).

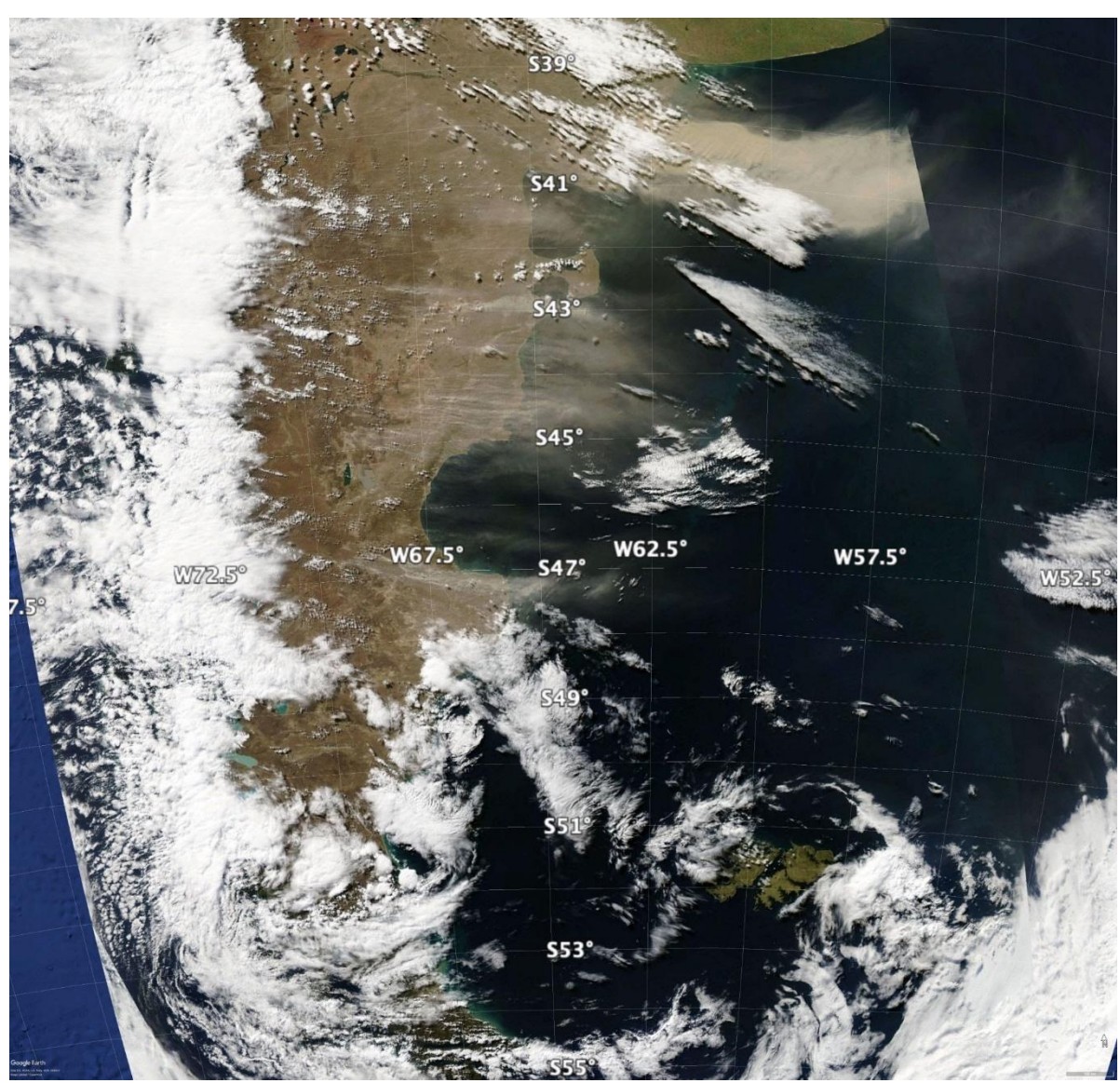

**Figure 14. A dust event spanning the north and central sections of the Patagonian Desert (+1000 km) on March 28, 2009. Events this large occur about once every one to three years. This event is typical in that it was triggered by the passage of a powerful low-pressure center commonly found in these high latitudes. Also, this event is singular in that a large portion of it is cloudless, enabling a direct view from space (most dust activity in Patagonia occurs under cloudy conditions). The thick dust cloud in the upper right corner is from an area used for cattle farming, which was undergoing a drought, whereas the active sources further south can be considered more naturally occurring with less anthropogenic interference. Source: NASA's Worldview interface image processed with Google Earth.**

Overall, satellite detection in the Patagonia region remains challenging. There are several difficulties in surveying dust activity
in the area: obstructed views from space because of cloudiness, nighttime dust activity, and sparse population. Also, except
for a few sources, the lack of recurrence in dust emission is a general feature of the desert: Sources that were active during one
season do not reactivate until two or three seasons later. A comprehensive and dedicated survey combining surface and space-
based detection networks is needed for a better understanding.
**3.4.9 Svalbard**
Evidence of the presence and activity of dust sources in Svalbard is only recent and quite rare. Yet, for example, dust storms
in Longyearbyen are reported as a regular feature in autumn. Dörnbrack et al. (2010) documented and characterized a strong
dust storm in the Adventdalen valley—the center of Spitsbergen Island—in May 2004, using airborne lidar observations and
mesoscale numerical modeling. In the same area, near Longyearbyen, dust emissions from an active coal mine were
documented by Khan et al. (2017). Kandler et al. (2020) also reported Svalbard measurements in Longyearbyen in September
2017, with high iron and chlorite-like contributions in dust.
The accelerated ablation of Svalbard's glaciers (Schuler et al., 2020) and the increasing melt rate of permafrost are causing
accelerated growth in periglacial and proglacial areas. The significance of the morphogenetic processes of deflation,
denudation, and sediment transport on slopes and in river channels in glaciers' marginal zones is increasing (Zwolinski et al.,
2013). Thus, these areas have become potential sources of dust and, as such, have been investigated for the physical and
chemical properties of their sediments, regardless of the documented occurrences of the dust events these areas have
experienced.
Fluvial, glaciofluvial, and weathering deposits at five different sites on the coastal plains near the Ny-Ålesund Research Station
(78.92481°N, 11.92474°E), NW Spitsbergen were investigated (Moroni et al., 2018). The mineralogical assemblage is
characterized by dolomite, calcite, quartz, albite, and sheet silicates (vermiculite, muscovite, clinochlore) in variable amounts,
along with monazite, zircon, apatite, baryte, iron sulfate, Fe, Ti, Cu, and Zn ores as accessory minerals. With a weight fraction
of 4 to 53% of particles smaller than 100 μm, these deposits should be considered a valid dust source. However, the contribution
is influenced by the modest extension of bare soils (less than 4 km$^2$) and the brief duration of the area's driest summer. The
composition of the aerosols collected at the Gruvebadet lab near Ny-Ålesund during the summer-fall period reveals the
presence of such a local dust component (Moroni et al., 2016; Moroni et al., 2018). Further evidence of local dust sources in
the Ny-Ålesund area and Brøgger Peninsula also results from the annual snowpack's chemical composition (Gallet et al., 2018,
Jacobi et al., 2019). The contribution of local dust sources on this site is of secondary importance compared to that of long-
range transport (Moroni et al., 2015; Moroni et al., 2016; Moroni et al., 2018, Conca et al., 2019).

A similar study was conducted on the loose sediment deposits in the neighborhood of the Polish Polar Station Hornsund (77.00180°N, 15.54057°E), SW Spitsbergen, where a belt of nearshore plains consisting of marine terraces and nival moraine bars, with bare surfaces available for mineral dust uplift from late spring, widely outcrop (Zwolinski et al., 2013). The mineralogical assemblage consists of quartz, alkali feldspar, plagioclase, dark mica, and chlorite, with zircon, apatite, monazite, iron sulfide, and Fe ore as accessory minerals. The same assemblage was found in the aerosols and snow cover collected at the base station and the surrounding glaciers in the same period. This fact, along with the significant proportion of particles smaller than 50 µm in the loose sediment deposits, supports the prevalence of the local dust source in the melting season. Further evaluation of the impact of local dust sources was obtained from analyzing shallow and deep cores from different glaciers in the Hornsund area (Lewandowski et al., 2020; Spolaor et al., 2020). The results suggest that for Spitsbergen glaciers with the summit close (Ny-Ålesund) or below (Hornsund) the equilibrium line, the summer dust deposition from the local sources is predominant, affecting the glacier ice's chemical composition. However, the dating of monazite grains and presence of magnetite and iron sulfide (magnetic susceptibility and SEM data, Lewandowski et al., 2020) also suggest the existence of regional wind transport from Nordaustlandet and Edgeøya, respectively. Further, a long-range component from Northern Europe, Siberia, and, to a limited extent, Greenland, Iceland, and Alaska, was also evidenced (Moroni et al., 2018; Crocchianti et al., 2021).

Recent estimations of dust load in Central and Southern Svalbard from different sources range from 4 g up to 4 – 5 kg m$^{-2}$ (Rymer, 2018; Rymer et al. 2022), with the highest values in the Ebba Valley due to frequent dust storms in this area (Strzelecki and Long, 2020). Kavan et al. (2020a) found a negative correlation between deposition rate and altitude at Pyramiden (78.71060°N, 16.46059°E), the west coast of Petuniabukta, and Ariekammen (77.00035°N, 15.53674°E), and Hornsund area. The pattern was clear up to the altitude of approximately 300 m a.s.l., suggesting the influence of local sources in the lower levels of the atmosphere and long-range transport at higher altitudes. The lower values of the deposition rates found at Ariekammen were ascribed due to the more frankly maritime climate of the Hornsund region.

**3.5 Climatic and environmental impacts of HLD**

Climatic and environmental impacts of HLD on clouds and climate feedback, atmospheric chemistry, marine environment and cryosphere-atmosphere feedback (Fig. 15) were investigated with the help of topical literature surveys (Sections 3.5.1 - 3.5.4). Direct radiative forcing of HLD dust (blocking sunlight) and comparison of dust and black carbon as SLCF in the cryosphere are included in the cryosphere-atmosphere feedback section.

The amounts of dust emission and deposition (megatonnes) of annual global and Arctic dust (data for 2017), as compared to anthropogenic and wildfire black carbon (Fig. 15), were studied using the SILAM model (Sofiev et al., 2015). The results of black carbon emissions presented in Figure 15 were based on the Copernicus Atmosphere Monitoring Service (CAMS) global

emission inventory version 4.2 and black carbon originating from wildfires from the SILAM IS4FIRES fire emission model, equaling 5 % of the total primary fire PM emissions of the model. The IS4FIRES model is based on fires observed by the MODIS instrument onboard the Terra and Aqua satellites.

| FLUX (MEGATONNES) | GLOBAL DUST < 30 μm | GLOBAL DUST < 2.5 μm | ARCTIC DUST < 30 μm | ARCTIC DUST < 2.5 μm | ANTHROPOGENIC BLACK CARBON (CAMS global emissions v4.2) | WILDFIRE BLACK CARBON (5% of fire PM emissions) |
|---|---|---|---|---|---|---|
| Total emission | 3000 | 160 | 30 | 1,6 | 4,6 | 1,9 |
| Deposition on snow | 32 | 5,2 | 4 | 0,21 | 0,18 | 0,029 |
| Deposition on sea ice | 5,5 | 0,59 | 3 | 0,17 | 0,009 | 0,008 |
| Deposition on Arctic snow | 7,6 | 1,1 | 4 | 0,19 | 0,027 | 0,013 |
| Deposition on Arctic Sea ice | 4,7 | 0,52 | 3 | 0,17 | 0,0055 | 0,0074 |
| Deposition on sea surface | 500 | 86 | 15 | 1,0 | 1,7 | 0,9 |
| Deposition on Arctic Sea surface | 21 | 2,4 | 12 | 0,68 | 0,035 | 0,063 |

**Figure 15. Climatic and environmental impacts of high latitude dust include direct radiative forcing (blocking sunlight), indirect radiative forcing (clouds and cryosphere) as well as effects on atmospheric chemistry and marine environment. The amounts of dust emission and deposition (megatonnes) of global and Arctic dust, as compared to black carbon, were estimated using the SILAM model (Sofiev et al., 2015). The black carbon emissions are based on the CAMS global anthropogenic emission dataset v4.2 and the wildfire black carbon emissions are based on the IS4FIRES fire emission model, equaling 5 % of the total primary fire PM emissions of the model.**

### 3.5.1 Impacts of HLD on atmospheric chemistry

Icelandic dust, a specific HLD of volcanic origin, is constantly resuspended from the deserts. Regarding atmospheric chemistry, the most substantial impact comes from the particles in the 0.002 to 10 µm range, as they can be carried over more considerable distances (Finlayson-Pitts, 1999). The Icelandic dust in the troposphere is not as addressed as the impact of desert dust. This HLD is very likely a long-range transporting carrier for many species adsorbed on its surface, which can act as a sink of trace gases and a subsequent platform for transferring taken-up species. Along with transport, adsorbed species may undergo different heterogeneous reactions that can lead to secondary compound formation. Such processes can influence the reactivity and balance of atmospheric species. Optical, hygroscopic, and, more generally, physicochemical properties of the HLD can change due to surface processes, implying atmospheric trace gases due to heterogeneous interactions (Usher et al., 2003). The consequences can be starkly different depending on the nature of atmospheric trace gases interacting with HLD. This section aims to illustrate the diversity of interactions between HLD and atmospheric trace gases to emphasize the various impacts of these aerosols on atmospheric physics and chemistry. In the case of ozone, if the direct heterogeneous interaction with dust does not play a major role in the atmospheric concentration decrease of the primary compound, surface processes are triggered, affecting the atmospheric budget of ozone. In the case of $NO_2$, heterogeneous processes on dust can significantly lead to HONO species forming, with direct impacts on gas-phase atmospheric reactivity. In the case of $SO_2$, beyond a complex reaction pathway, the heterogeneous process dually affects the budget of the taken-up species and the chemical and physical properties of the dust surface.

If the heterogeneous reaction of $NO_2$ on various types of atmospheric particles, e.g., salts, soot, mineral dust, and proxies, was addressed in the literature (George et al., 2015), the interaction of $NO_2$ with volcanic particles, typical HLD desert dust, under atmospheric conditions, has only been studied by Romanias et al. (2020). They explore the possible formation of a short lifetime key atmospheric species, considered a trigger of numerous atmospheric processes: HONO, a precursor of OH radicals in the atmosphere. To that end, $NO_2$ uptake on Icelandic HLD is explored under various and contrasting atmospheric conditions. Despite the relatively close volcanic regions where the selected samples originate, uptake coefficients of $NO_2$ contrasted significantly with the dust location due to magmatic and morphological differences among samples. This point confirms that concerning heterogeneous atmospheric chemistry, sample behavior can dramatically deviate from one class of dust to another, with physical and chemical characterizations of the samples remaining key intrinsic descriptors. Nonetheless,

volcanic dust appears as effective $NO_2$ scavengers from the atmosphere. The interaction of $NO_2$ with HLD is evidenced as a source of NO and, more interestingly, HONO, with kinetics and formation yields highly dependent on relative humidity. Higher HONO formation yields on volcanic samples are observed for RH values exceeding 30% RH. Heterogeneous construction of HONO from $NO_2$ interaction with Icelandic dust is estimated as atmospherically significant under volcanic eruptions or, more frequently in Iceland, during typical volcanic dust storms. Such leads to HONO formation rates up to 10 pptV/hr, which can significantly influence the regional atmosphere's oxidative capacity. The experimental determination of $NO_2$ uptake coefficient $\gamma$ allows including such processes in atmospheric modeling, improving their representativeness.

A transient uptake of $SO_2$ – an initially important uptake of $SO_2$ that is progressively reduced – leads to low steady-state uptake coefficients of $SO_2$ after several hours of exposure in the range of 10–9 to 10–8. The surface coverages were in the range of 1014 molecule cm-2 or 1016 molecule cm-2 using the total surface area or the geometric surface area of aerosols, respectively (Urupina et al., 2019). Zhu et al. (2020) estimated that around 43% more volcanic sulfur is removed from the stratosphere within months due to $SO_2$ heterogeneous chemistry on volcanic particles than without. Concomitantly with $SO_2$ uptake, sulfites and sulfates are monitored on the surface of volcanic dust, with sulfates being the final oxidation product, attesting to $SO_2$ surface reaction. Through surface hydroxyl groups, the dust surface's chemical composition plays a crucial role in converting $SO_2$ to sulfites, as evidenced experimentally using lab scale but atmospheric relevant experimental setups (Urupina et al, 2019). This provides original insights into the kinetics and mechanism of $SO_2$ uptake and the transformation on volcanic material under simulated atmospheric conditions, bringing an accurate perspective on $SO_2$ heterogeneous sinks in the atmosphere on the HLD surface. The model simulations of Zhu et al. (2020) suggested that the transformation of $SO_2$ on such particles plays a key role in the stratosphere's sulfate content. Interestingly, this transformation and accumulation of sulfates on the surface of particles could turn the unreactive ozone material into reactive, especially in the stratosphere, where volcanic particles have longevity.

The case of $SO_2$ uptake points to the aging of the HLD surface with subsequent impacts on its chemical (e.g., hygroscopicity) and physical (e.g., optical) properties. Changes in hygroscopic properties can correlate with HLD's erratic behavior to act as cloud-or ice-nucleating particles, depending on their interactions with atmospheric gases. Similarly, sulfate and sulfuric acid's high surface coverage for volcanic dust, as reported by Urupina et al. (2019), questions the variability of the HLD refractive index and the impact on remote sensing of fresh vs. aged dust.

**3.5.2-Impacts of HLD on clouds and climate feedback**

Clouds across the mid- and high latitudes are of first-order importance. Climate and HLDs may play a first-order but highly uncertain role in defining their properties through the initiation of ice formation. Clouds frequently persist in a supercooled state. However, even a few droplets converting to ice crystals through heterogeneous freezing can lead to microphysical processes that dramatically reduce a cloud's liquid water content, reducing its albedo and exposing the surface underneath

(Murray et al., 2021; Tan and Storelvmo, 2019). Only a small subset of atmospheric aerosol can nucleate ice; concentrations of around only 1 INP per liter of air active at the cloud temperature can dramatically alter cloud albedo. In contrast, the concentration of aerosol particles capable of serving as cloud condensation nuclei (CCN) are orders of magnitude larger. Hence, dust particles in the high latitudes will rarely exist in high enough concentrations to dramatically impact cloud droplet numbers by providing additional CCN. However, high-latitude dust has been shown to serve as an effective INP in sufficient concentrations to potentially impact mixed-phase clouds (Sanchez-Marroquin, 2020). Ice formation's role in climate projections depends on the clouds' location. In the following paragraphs, we discuss two distinct classes of clouds that may be influenced by HLD particles serving as INPs.

For boundary layer clouds over oceans between approximately 45–70°, the amount of ice versus supercooled water, as well as albedo, is critical for global climate (Vergara-Temprado et al., 2018; Bodas-Salcedo et al., 2014). These clouds are where substantial solar insolation exists, and the contrast between a high albedo cloud and a dark ocean surface is significant. Hence, these clouds are implicated in the cloud-phase feedback, where water replaces ice, increasing their albedo as the world warms with increased carbon dioxide (Storelvmo et al., 2015). This feedback's uncertainty is very high, with the temperature rise associated with a doubling of carbon dioxide, rising from around 4 K to well above 5 K, simply by increasing the amount of supercooled water in clouds in the current climate (Frey and Kay, 2018). Hence, understanding the sources of ice-nucleating particles in the high latitudes, including HLDs, is critical to understanding these climate-relevant issues (Murray et al., 2021).

The second group of clouds is those occurring at high latitudes. For example, in the central Arctic, mixed-phase clouds play a critical role in the local Arctic climate and the phenomenon of Arctic amplification. In a corollary to the cloud-phase feedback, water replacing ice leads to more downward longwave radiation, resulting in positive feedback (i.e., amplification) (Tan et al., 2019). Hence, the phase of clouds and, therefore, the INP population in clouds in the present Arctic atmosphere are key for defining this feedback's strength. Moreover, any changes in the INP population with a changing climate may also provide feedback on cloud properties (Murray et al., 2021).

Given the apparent importance of INPs in defining cloud properties and climate feedback, surprisingly little is known about the ice-nucleating properties of HLDs. Mineral dust is one of the most important types of atmospheric INPs in clouds below approximately -15°C around the globe because of its relatively high ice-nucleating activity and abundance in the atmosphere (Murray et al., 2012). A handful of papers have also identified HLDs as significant contributors to the Arctic's INP population (Irish et al., 2019; Sanchez-Marroquin, 2020; Tobo et al., 2019; Šantl-Temkiv et al., 2019). HLDs may differ in their ice-nucleating ability from LLDs for several reasons: Firstly, the HLDs from glacial valleys, for example, are often richer in primary minerals (olivenes, pyroxenes, feldspars, and amphiboles) and less rich in clays compared to LLDs. This is crucial because K-rich feldspars are known for their exceptional ice-nucleating ability, whereas clays are much less active (Harrison et al., 2019; Atkinson et al., 2013). Secondly, the most prominent LLD sources, like those in Africa, are abiotic (Price et al., 2018), whereas it has been found that HLDs can be associated with highly effective biogenic ice-nucleating material (Tobo et

al., 2019; Šantl-Temkiv et al., 2019). The inclusion of biological ice-nucleating material, which can be ice-active at temperatures much higher than -15 °C, may mean that these dust sources have a disproportionately greater impact on cloud glaciation and climate than their low-latitude counterparts. Much more research is needed to define and understand the ice-nucleating ability of these HLD sources.

### 3.5.3 Impacts of HLD on the marine environment

Mineral dust particles are a source of essential nutrients such as phosphorus (P) and iron (Fe) to the ocean ecosystems (e.g., Jickells et al., 2005; Mahowald et al., 2005; Stockdale et al., 2016). Dust deposition onto the ocean's surface can stimulate primary productivity and enhance carbon uptake, indirectly affecting the climate (e.g., Jickells and Moore, 2015; Mahowald, 2011). The extent of these impacts primarily depends on the dust deposition fluxes, its chemical properties, and the nutrients of (co)limitations patterns in the ocean waters (e.g., Boyd et al., 2007; Boyd et al., 2010; Kanakidou et al., 2018; Mahowald et al., 2010; Mills et al., 2004; Moore et al., 2013; Shi et al., 2012; Stockdale et al., 2016). Arctic Ocean is often nitrogen-limited (von Friesen and Riemann, 2020).

The aerosol fractional Fe solubility (%) is defined as the ratio of dissolved Fe (in the filtrate, which has passed through 0.2 or 0.45 µm pore size filters) to the total Fe in the bulk aerosol (e.g., Meskhidze et al., 2019; Shi et al., 2012). This is typically used to indicate the fraction of Fe, which is likely to be bio-accessible for marine ecosystems (Meskhidze et al., 2019).

Sub-Arctic oceans are Fe-limited or seasonally Fe-limited. Fe limits primary productivity in the Sub-Arctic Pacific Ocean (Martin and Fitzwater, 1988). The atmospheric Fe deposition in the Gulf of Alaska is dominated by dust transported from glacial sediments from the Gulf of Alaska coastline (Crusius et al., 2011), with relatively high fractional Fe solubility—around 1.4% (Schroth et al., 2017). Although the upwelling of deep water is the major source of dissolved Fe, the atmospheric flux of dissolved Fe to the Gulf of Alaska's surface water is comparable to the Fe flux from eddies of coastal origin (Crusius et al., 2011). The magnitude of glacial dust's deposition to the Gulf of Alaska varies significantly depending on the regional weather conditions. However, the extent of its impacts is still unclear (Schroth et al., 2017). Currently, the spatial resolution of global dust models is too low to accurately reproduce Alaskan dust flux, generated by anomalous offshore winds and channeled through mountains (Crusius, 2021). Recently, Crusius (2021) determined dissolved Fe inventories based on time series of dissolved Fe and particulate Fe concentrations from the Ocean Station Papa in the central Gulf of Alaska, including measurements from September 1997 to February 1999. The analysis showed 33%–70% increases in dissolved Fe inventories between September and February of successive years. These increases were possibly linked to dust fluxes from the Alaskan coastline—known to occur mostly in autumn (Crusius et al., 2011; Schroth et al., 2017). These new results support the importance of atmospheric Fe's contribution, although more work is needed to confirm the sources of dissolved Fe in the Gulf of Alaska.

The Sub-Arctic North Atlantic Ocean is seasonally Fe-limited (Nielsdottir et al., 2009; Ryan-Keogh et al., 2013). Natural dust from Iceland largely contributes to the atmospheric dust deposition in the North Atlantic Ocean (Bullard, 2016). Icelandic dust originates from volcanic sediments and has a relatively high total Fe content—about 10% (e.g., Arnalds et al., 2014, Baldo et al., 2020). The estimated total Fe deposition from Icelandic dust to the ocean's surface is 0.56–1.38 Mt yr-1 (Arnalds et al., 2014). The initial Fe solubility observed in dust samples from Icelandic dust hotspots is from 0.08% to 0.6%—comparable to that of mineral dust from low-latitude regions such as Northern Africa, while the fractional Fe solubility at low pH (i.e., 2) is significantly higher than typical low-latitude dust (up to 30%) (Baldo et al., 2020). Achterberg et al. (2018) argued that deep-water mixing is the dominant source of Fe in the Sub-Arctic North Atlantic Ocean's surface water, which is up to ten times higher than the Fe supply by atmospheric Fe deposition. However, during the 2010 eruption of the Icelandic volcano Eyjafjallajökull, Achterberg et al. (2013) observed elevated dissolved Fe concentration and nitrate depletion in the Iceland Basin, followed by an early spring bloom. They measured an initial fractional Fe solubility of 0.04 %–0.14 % for Icelandic ash, which is below or towards the lower end of the range of values estimated for Icelandic dust (0.08%–0.6%) (Baldo et al., 2020). High deposition flux (Arnalds et al., 2016) and higher Fe solubility of Icelandic dust (Baldo et al., 2020) suggest that they may impact Fe biogeochemistry and primary productivity in the surface ocean. However, more research is needed to confirm this.

The Southern Ocean is known to be Fe-limited (Moore et al., 2013). Major atmospheric dust sources include, for example, Australia, southern South America, and Southern Africa (e.g., Ito and Kok, 2017). Contribution from local sources in Antarctica is also observed (e.g., Chewings et al., 2014; Winton et al., 2014; Winton et al., 2016). Winton et al. (2016) reported a background fractional Fe solubility from Antarctic dust sources of 0.7%—similar to the upper limit of Fe solubilities observed in Icelandic dust (Baldo et al., 2020). However, mineral dust originating from glacial sediments from the Gulf of Alaska coastline showed higher Fe solubilities (1.4%) (Schroth et al., 2017)—likely due to the different mineralogy and Fe speciation in the samples. The various methods used to determine the fractional Fe solubility in these studies may also contribute to this difference (Perron et al., 2020).

Although the upwelling of deep water is a major source of dissolved Fe, the atmospheric deposition of dissolved Fe can locally contribute to the phytoplankton bloom (Winton et al., 2014). However, evidence exists that increased dust flux enhanced primary production in the Southern Ocean in the last glacial age (Martínez-García et al., 2014). The Ross Sea is a continental shelf region around Antarctica and a highly biologically productive area in the Southern Ocean, which has important implications for global carbon sequestration (e.g., Arrigo et al., 2008; Arrigo and Van Dijken, 2007). In the Ross Sea, an additional Fe supply is required to sustain the intense phytoplankton bloom during the austral summer (Tagliabue and Arrigo, 2005). Measurements conducted on snow pits and surface snow samples showed that local Antarctic dust contributes to Fe deposition. However, this contribution is only a minor component of the total Fe supply to the Ross Sea, with most being supplied by the upwelling of deep water (Winton et al., 2014; Winton et al., 2016a,b).

In the Polar regions, atmospheric dust is mostly delivered to the sea ice, where melting and freezing cycles (ice processing)
can enhance the formation of relatively more soluble phases of Fe oxide-hydroxide minerals such as ferrihydrite. This
formation can increase the flux of atmospheric dissolved Fe to the ocean (Raiswell et al., 2016).
**3.5.4 HLD impacts on cryosphere and cryosphere-atmosphere feedback**
The cryosphere is the frozen water part of the Earth system, including sea, lake, and river ice; snow cover, glaciers, ice caps,
and ice sheets; permafrost and frozen ground. These components play a crucial role in the Earth's climate (IPCC, 2019).
Temperatures in fragile areas, such as the pristine polar regions, have been increasing at twice the global average; the highest
increase in the temperature on the coldest days—up to three times the rate of global warming—is projected for the Arctic
(IPCC, 2021). Warming in vulnerable cold climate land areas causes glacier retreat, permafrost thaw, and a decrease in snow
cover extent (IPCC, 2019). Consequently, potential HLD sources, such as glacial sediments, can increase (e.g., Nagatsuka et
al., 2021). When dust is long-range transported and wet- or dry-deposited or windblown from local dust sources, the ice and
snow albedo decrease and influences glacier melt (e.g., Boy et al., 2019) via the positive ice-albedo feedback mechanism
(AMAP, 2015; Flanner et al., 2007; Gardner and Sharp, 2010; IPCC, 2019). Cryospheric melt processes are controlled by
many environmental factors (IPCC, 2019), such as solar irradiance, ambient temperature, and precipitation (e.g., Meinander
et al., 2013, 2014; Mori et al., 2019). Kylling et al. (2018) used dust load estimates from Groot Zwaaftink et al. (2016) (using
low-latitude dust complex refractive index for high-latitude dust) to quantify the mineral dust instantaneous radiative forcing
(IRF) in the Arctic for 2012. They found that the annual-mean top of the atmosphere IRF ($0.225$ W/m$^2$) had the largest
contributions from dust transported from Asia south of 60°N and Africa; high-latitude (>60°N) dust sources contributed about
39% to the top of the atmosphere IRF. However, HLD had a larger impact (1 to 2 orders of magnitude) on IRF per emitted
kilogram of dust than low-latitude sources. They also reported that mineral dust deposited on snow accounted for nearly all
the bottom of the atmosphere IRF ($0.135$ W/m$^2$), with over half caused by dust from high-latitude sources north of 60°N.
For snow and ice (glacier) surface radiation balance, the net energy flux $E_N$ is due to differences between downward ($\downarrow$) and
upward ($\uparrow$) non-thermal shortwave (SW) and thermal longwave (LW) radiative fluxes. Such is most critically influenced by
the surface characteristics of the bi-hemispherical reflectance (BHR), i.e., albedo (Manninen et al., 2021). Therefore, melt is
also controlled by dark impurities in snow and ice (IPCC, 2019). Black carbon (BC) is, climatically, the most significant and
best-studied dark light-absorbing aerosol particle in snow (e.g., Bond et al., 2013; Dang et al., 2017; Evangeliou et al., 2018;
Flanner et al., 2007; Forsström et al., 2013; Mori et al., 2019; Meinander et al., 2020a,b). Radiation-transfer (RT) calculations
indicate that seemingly small amounts of black carbon (BC) in snow, in the order of 10–100 parts per billion by mass (ppb),
decrease its albedo by 1–5% (Hadley and Kirchtetter, 2012). BC has been shown to enhance snowmelt (AMAP, 2015; Bond
et al., 2013; IPCC, 2019). Other light-absorbing particles include organic carbon (OC, including brown carbon BrC) and dust.
Also, blooms of pigmented glacier ice algae can lower ice albedo and accelerate surface melting (McCutcheon et al., 2021),

showing a direct link between mineral phosphorus in surface ice and glacier ice algae biomass. They say nutrients from mineral dust likely drive glacier ice algal growth, identifying mineral dust as a secondary control on ice sheet melting. Some Icelandic dust sources have particles almost as black as black carbon by the reflectivity properties when measured as bulk material or on snow and ice surfaces (Peltoniemi et al., 2015). Unlike black carbon, Icelandic dust has been shown to melt snow quicker in small amounts and insulate and prevent melt in larger amounts (e.g., Dragosics et al., 2015; Möller et al., 2016; Boy et al., 2019). Changes related to permafrost thaw and snow and ice melt, including disappearing glaciers, rising sea levels, and drinking water shortages, are among the most serious global threats (IPCC, 2019). Water availability is vital in regions where crops depend most on snowmelt water resources (Qin et al., 2020). Snow is also essential in the catchment areas (i.e., areas supplying watercourses) and for many snow-dependent organisms, including plants, animals, and microbes (Zhu et al., 2019). Melt can also run hydroelectric power plants that supply electricity (e.g., Lappalainen et al., 2022). This all highlights the importance of investigations and continuous assessment of the temporal and spatial significance and contribution of different light-absorbing impurities in enhancing or initiating cryospheric melt in the changing climate.

## 3.6 Understanding the HLD sources

The HLD results are further discussed from the perspective of HLD source intensity values; comparison with available HLD information on the various regions; geological perspective on sources, focusing on a gap identified in HLD observations for the Central part of the East European Plain and dust particle properties; and local HLD sources and long-range transport of dust with the focus on results from the observations in Svalbard and Antarctica.

### 3.6.1 Source intensity values

Most of the HLD study sites agree with UNCCD G-SDS-SBM source intensity (SI) values of the highest dust productive areas, identifying an environment from a given location within a distance $\leq 0.1°$. Surfaces with higher maximum SI include a significant portion of the land surface in HLD regions. In the south HLD region, an annual change of SI exists. However, approximately half the dust productive surface stays exposed to wind erosion during the year. In the north HLD region, SI intensity varies significantly with the weather. High values of SI may not always coincide with high surface winds, meaning high values may exist but not necessarily result in a dust storm. In case emissions occur, dust may remain undetected because of the absence of ground observations over most of the HLD region and frequent cloud cover over airborne dust, obscuring remotely sensed imagery.

Based on the SI values, the East Greenland sources in this study (no. 58–64 in Fig. 1) are seasonal, meaning their SI minimum value is zero. Conversely, the West Greenland sources are not necessarily seasonal since their SI minimum values are somewhat reduced (but not to zero). However, the term "seasonal" regarding the SI values means the soil surface conditions

are suitable for dust emissions, although that doesn't mean emissions will happen. Similarly, the seasonality of all sources in this collection can be further studied.

When the newly identified sources are close together, such might indicate they are part of the larger dust source area, like South Iceland, West Greenland, or East Greenland. The discovered sources could be considered to represent the hot spot locations, i.e., the most emissive or active locations, of those dust-productive areas. Simultaneously, however, the land surface and soil composition can be very complex and spatially variable, and the identification of single sources justified until the source characteristics and particle properties have been characterized in more detail. For example, Icelandic sources have shown that each source, even proximate ones, may have different particle size distributions and optical properties. The results (Fig. 2) suggest two northern high-latitude dust belts. The first HLD belt would extend at 50–58°N in Eurasia and 50–55°N in Canada, and the second dust belt at >60°N in Eurasia and >58°N in Canada, with a "no dust" belt between the HLD and LLD dust belts (except for British Columbia).

Uncertainties about the detected locations of the HLD sources and G-SDS-SBM source intensity values arise from the methodology for determining HLD source locations. These locations are ad-hoc location sources from satellite images of dust plumes; other kinds of airborne dust observations may introduce some errors in location estimation compared to on-site land surface monitoring and the precision of available data locations. The resolution of G-SDS-SBM may be too coarse for small-scale source areas (in this case, the representative grid point value shows reduced source intensity value since it represents the whole grid box). However, the in-point (at location) values are also given maximum values in the area around the given location (one point distance: 30 arcsec, 0.1°, 0.5°, and 1° distance). Values of source intensity above 0.9 have topsoil potential for SDS production in the top 10% of grid boxes with some emission potential in G-SDS-SBM (or in the top 10% of most dust-productive surfaces globally in case of favorable weather conditions), above 0.8 in the top 20%, above 0.7 in the top 30%, and so forth. Factors reducing source function value or topsoil dust productivity are sparse vegetation, coarser soil texture, higher moisture, and temperatures near the freezing point. Uncertainties in methodology for deriving G-SDS-SBM arise from the quality and resolution of available global datasets and the determination of thresholds for EVI in defining bare land fraction (primarily for brown grassland, which may appear as potential dust sources but with lower productivity). Surfaces with low SI values in favorable conditions for dust emission, in case of high winds, may produce some blowing dust events; sources with higher values of SI may generate dust storms. Real dust production from sources depends on high winds occurring while SI is high.

Forty-nine locations were in the north HLD region (47 according to HLD definition by Bullard et al. (2016), except for two: no. 8 and no. 48, with latitudes 47.7°N and 47.6°N, respectively), while 15 were in the south HLD region, including four south of 60°S, where the values of SI are not provided. In the north HLD region, higher dust productive potential (SI 0.5) has 17 of 49 marked locations at the HLD source marks exact location. Also, 38 sites are where a distance from a mark point (D) is equal to or less than 0.1° (Supplementary Table S4). Very high dust productivity, with SI 0.7, has 33 locations within D 0.1°,

and 42 and 46 within 0.5° and 1°, respectively. The highest dust productive potential, with SI 0.9°, has 27 locations within D
0.1°, and 39 and 44 within 0.5° and 1°, respectively. One point has the highest SI value, with less than 0.5° and five less than
0.9° away, when considering the largest environment of the HLD source mark. Three HLD source region marks are in the sea,
so their source values are marked as -99 (undefined). In the south HLD region, 11 locations are considered (situated between
40°S and 60°S). Seven sources have very high dust productivity with SI 0.7 where the HLD source marker is; three more have
SI 0.7° in the area of the source marker with D 0.1°. The highest dust productive potential, with SI 0.9°, has seven sources in
the area with the source marker with D 0.1°, and three more in the area with D 0.5°. The source maximum and minimum
intensities in these south HLD regions differ much less than in the north HLD region.
As a summary, our modeling results on the spatial distribution of the dust sources (Fig. 2) showed evidence supporting a
northern High Latitude Dust (HLD) belt, defined as the area north of 50°N, where we distinguish the following HLD-source
areas: (a) "transitional HLD-source area," which extends at latitudes 50–58°N in Eurasia and 50–55°N in Canada, and (b)
"cold HLD-source area," which includes areas north of 60°N in Eurasia and north of 58°N in Canada, with currently "no dust
source" area between the HLD and LLD dust belts (except for British Columbia).

### 3.6.2 Comparison of various regions

For the HLD sources identified and included in our collection, the available information varied from detailed characterizations
to the first satellite observations, waiting to be complemented with measurement data. Model output of dust transport can
provide valuable additional information. The sources are in the northern and southern high latitudes and include a variety of
environments. Particle properties, such as particle size distributions, have been determined for only some of the identified
HLD sources. For example, our study's many Iceland south coast sources have not had any characterization done. Previous
results on the known sources in Iceland's south coast region show that the particle size distributions vary substantially from
location to location. No assumptions can be made based on characterization in one place.
For Iceland seasonality, the correlation of SILAM modeled and measured PM10 and PM2.5 total aerosol concentration in
Iceland is low, especially in 2018, which can mainly be explained by the measurement locations being far from the source
locations and showing the effects of road dust rather than long-range transported dust. Also, the Reykjavik and Akureyri dust
inventories are unrepresentative due to the challenge of fitting the modeled long-range transported dust emissions 695 to the
measurement data within the 0.1 degrees model resolution. Near Reykjavik, dust emissions, e.g., from Landeyjasandur, may
contribute to the measured dust concentrations. However, the 0.1 degrees resolution of the model is too scarce to simulate
them.
The end of summer and autumn (October) are the seasons for dust activity in Greenland. For example, on 19 October 2021,
there was significant dust activity in western Greenland; several glacial valleys emitted dust along the 700-km coast. During

that dust event was a good Sentinel overpass showing a long narrow valley with a great deal of haze(dust) suspended (appearing as fuzziness in the image) (Gassó, 2021b). As far as we know, no previous observations for this source exist. Greenland's HLD sources (no. 53–58 of Fig. 1) from its west coast are considered new and identified here using satellite observations. Currently, further knowledge on the recurrency and area of the emission source is lacking. It is probable that these Greenland HLD sources from the west coast have been unidentified due to frequent cloudy conditions. The representation of dust sources in modeling approaches requires information on the location, soil characteristics, and temporal changes. A detailed specification of the geographic distribution of potential dust sources and their physical (e.g., particle size distribution, optics) and mineralogical/chemical (mineral fractions, chemical composition, etc.) properties is critical to accurately parameterize dust emission's potential in numerical dust models. Various methods exist to detect new sources; remote sensing is one of the most powerful tools, as demonstrated in Iceland's southern coast and Greenland's west and east coasts.

The central part of the East European Plain, with the wide occurrence of silty soils derived from loess-like sediments and reduced natural vegetation, is a potential aeolian dust source (Bullard et al., 2011; Sweeney and Manson, 2013). However, this region currently lacks observations on dust lifting and transport. Therefore, this region was not included in our collection of HLD sources. The gap for observations in the central part of the East European Plain for potential HLD source updates is filled here with new data in the Supplement Figures S1–S4 on the partitioning of elements among the five particle-size fractions separated from the natural soils of a rural area 100 km southwest of Moscow (Fig. S1). The study area (55°12–13'N, 36°21–22'E) belongs to the southeastern part of the Smolensk-Moscow Upland (314 m a.s.l.), representing a marginal area of the Middle Pleistocene (MIS 6) glaciation with moraine topography modified by post-glacial erosional and fluvial processes. The major soil reference group is Retisol (IUSS Working Group WRB, 2015), developed on the loess-like loam. About 50% of the soils in the interfluve area were subjected to arable farming. A new and unpublished independent dataset on 33 elements in topsoil horizons was obtained with a higher accuracy ICP-MS/AES analysis (compared to the DC-ARC-AES data set of Samonova and Aseyeva, 2020).

Additional dust sources with massive dust storms causing severe traffic disruption have been documented outside the dust belt in higher latitudes. These sources were mainly arable fields, such as those in Germany and Poland, as well as Montana and Washington state (in the US) (Hojan et al., 2019).

### 3.6.3 A geological perspective on HLD sources and particle properties

Dust sources involve very different formations and geological environments, each leaving its own imprint on the sediments. Thus, the geomorphological, sedimentological, petrological, and geochemical study of the loose sedimentary formations in the source areas provides information on the origin and provenance of dust when it is transported out or far away. These types of studies—quite typical for Saharan dust—are not so well-established in the case of HLD sources. These territories are not all

easily accessible. Even when they are, the time may not coincide with the dust production and/or dust emission period, which may be one reason for this missing source area characterization.

Geomorphological studies cover a wide range of subjects and topics, from characterizing specific dust sources (e.g., Arnalds et al., 2016; Bullard and Mockford, 2018; Bertran et al., 2021) to analyzing processes (e.g., Bullard and Austin, 2011; Hedding et al., 2015; Wolfe, 2020) to landform evolution (Heindel et al., 2017). Sedimentological studies on dust sources focus on the particles' morphological characteristics and textural details of the loose sediment formations. The size, shape, and surface characteristics of the particles result from morphogenetic processes. As such, these particles say a great deal about the source areas. Furthermore, the particles' size and shape influence their lifting and transport capacity and, finally, the distance they can reach from their site of origin. Such applies to the studies of the properties of volcaniclastic dust sources in Iceland (e.g., Butwin et al., 2020; Richard-Thomas et al., 2021). From the petrological and geochemical perspective, the panorama is even wider and more varied. Save a few (e.g., Baratoux et al., 2011; Moroni et al., 2018), most studies are not aimed at studying dust sources but comprise different targets involving the parental soils (e.g., Antcibor et al., 2014; Brédoire et al., 2015). Although providing information on the (possible) source areas for dust, these latter studies are not explicitly aimed at studying dust sources, so they are not functional for that purpose. Specific survey and sampling activities by a team of experts would be required to address all aspects of dust sources and properties adequately. Thus, obtaining a database as rich and articulated as possible on the particles' physico-chemical properties within dust would be feasible, providing the ability to predict dust behavior within the aerosols and understand medium and long-range transport phenomena is present. A further aspect regarding dust sources and properties is the evolution of the particles' physico-chemical properties due to the lifting and transport mechanisms. The aerosols must be sampled in different places at different distances from the source. However, this approach is complicated by the air masses mixing during transport, requiring a deep investigation of air mass back trajectories. Conversely, treating the soils in the lab by re-suspending and sampling them using impactors at well-defined cut-off size ranges can be very advantageous. Such work has been carried out on Australian soils and southern African soils (Gili et al, 2021) to study the dust sources in Antarctica, which is now underway in Iceland (Moroni, 2021, personal communication).

**3.6.4 Local HLD sources versus long-range transported dust: discussing Svalbard and Antarctica**

The same areas of dust lifting can also be deposition sites when particles leaving their respective source regions are deposited there after prolonged transport pathways. The extent of the contribution of local and long-range sources may vary during the year depending on the type of atmospheric circulation and state of the exposed surfaces, particularly the presence of bare deglaciated soils. Such is the case of Svalbard, where the local dust sources prevail over the long-range ones, especially in summer; the contrary occurs the rest of the year (Moroni et al., 2016; Spolaor et al., 2021). Conversely, and always in Spitsbergen, the type of contributions—local and long-range—may also depend on the altitude due to the stratified structure of the lower atmosphere frequently found at high latitudes (e.g., Moroni et al., 2015; Kavan et al., 2020a).

Investigating the physico-chemical properties, and possibly estimating their contributions at different times of the year, is key to identifying the source regions of dust. For example, in Spitsbergen's case, the potential Source Contribution Function (PSCF) analysis of aerosol samples taken in Ny-Alesund clearly identified four different HLD sources in Eurasia, Greenland, Arctic-Alaska, and Iceland (Crocchianti et al., 2021). Conversely, chemical-mineralogical investigation and single-particle analysis recognized and estimated Icelandic dust's contribution to Ny-Alesund (Moroni et al., 2018).

Kandler et al. (2020) collected dry dust deposition near sources in Northwestern Africa, Central Asia, on Svalbard, and at three locations of the African outflow region, and studied particle sizes and composition. Their results showed low temporal variation in estimated optical properties for each site but considerable differences among the African, Central Asian, and Arctic regions. An insignificant difference was found between the K-feldspar relative abundances, indicating comparable ice-nucleation abilities. The mixing state between calcium and iron compounds differed for near-source and transport regimes, potentially and partially due to size-sorting effects. Thus, in certain situations (high acid availability, limited time), atmospheric processing of the dust is expected to lead to less iron solubility for near-source dust (for Central Asian ones) than for transported ones (particularly those of Sahelian origin).

In the southern part, under certain meteorological conditions, dust from lower latitudes can be transported far toward polar regions. Such was the case when a massive dust storm formed over Australia on 22 January 2020. Two days later, dust moved southward, covering a large part of Antarctica's eastern coast. The RHMSS global version of the DREAM model with incorporated ice nucleation parameterization due to dust (Nickovic et al., 2016) predicted the formation of cold clouds over the Antarctic. This ice cloud phase was also documented by NASA satellite observations (Fig. 16). The simulation was part of a WMO SDS-WAS initiative to include dust impacts on high latitudes in its agenda to better understand the role of mineral dust as a climate factor at high latitudes.

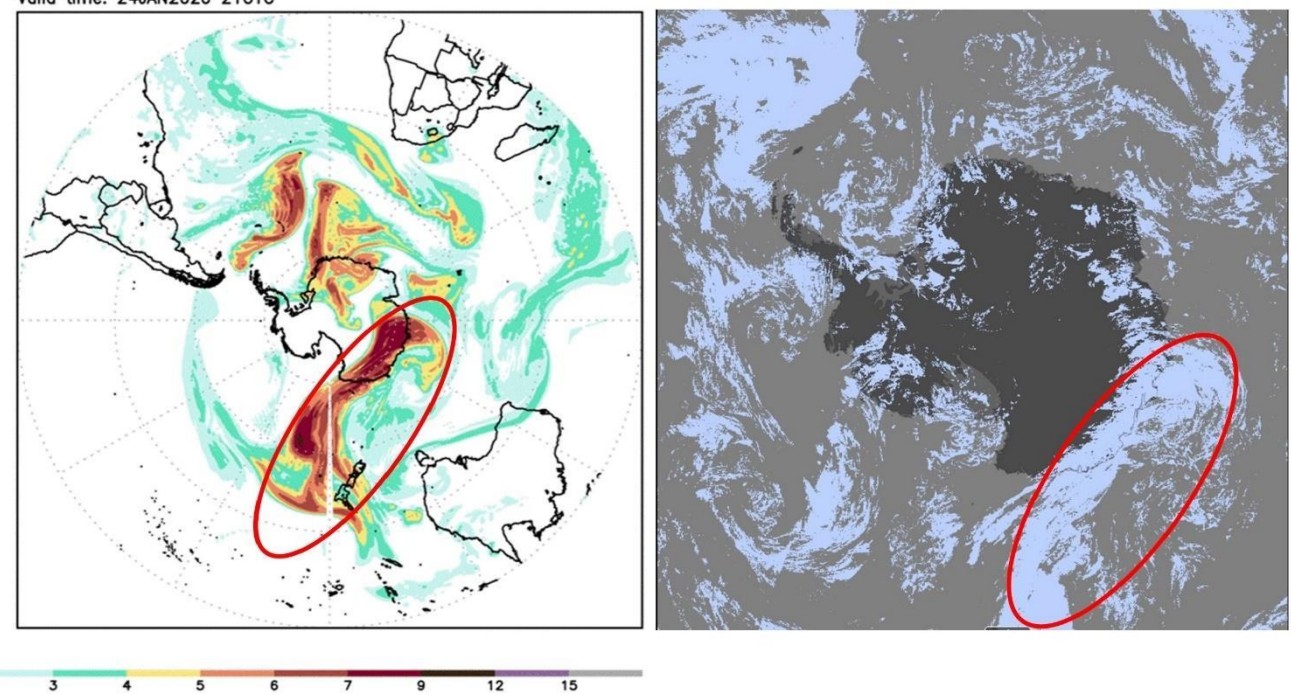

**Figure 16. Global NMMB DREAM model experiments over Australia and the South Pole. Model dust load 22 January 2020; B) Model log$_{10}$ (vertical load of ice nuclei number) (left). NASA MODIS ice cloud phase for 24 January 2020 (right).**

The McMurdo Dry Valleys (MDV) were previously assumed to be a significant regional source of dust (e.g., Bullard, 2016). New observations show otherwise. Instead, the McMurdo Ice Shelf's (sometimes called the McMurdo debris bands) debris-covered surface is the major dust source. In this study, more details are provided to underline the importance and estimates of the size of the areas. The MDV (4 800 km$^2$) was estimated to fit Category 3 best. Despite active local aeolian sediment transport with many annual occurrences, they are an insignificant source or exporter of dust regionally, thus having only a small but poorly known climatic or environmental significance. The MDV are changing quickly with increased ablation, meltwater, and permafrost incision, so their importance regarding dust generation may change in the near future. The McMurdo Ice shelf "debris bands" fit Category 2 best. Although it is only about 1500 km$^2$, the McMurdo Ice shelf is clearly the region's largest and most important dust source—active with a continuous supply of new sediment for export and exposed to frequent strong winds (with many events during the year), although few have been documented. The aeolian sediment impacts sea ice albedo (not directly measured) and marine sedimentation, contributing enough dissolved Fe to potentially support up to 15% of primary productivity in the SW Ross Sea (Winton et al., 2014).

Ice core studies from Antarctica ice sheets show that Antarctica receives long-range dust transport from Australia, South
America, South Africa, and New Zealand (e.g., Bullard, 2016). However, several studies around coastal areas have shown that
locally, Antarctic sourced dust accumulation rates are at least two orders of magnitude higher than that recorded from the polar
plateau or global dust models (Chewings et al., 2014; Winton et al., 2014).

## 4 Conclusions and outlook

This study aimed to identify new HLD sources with the focus on their potential climatic and environmental impacts. A
literature survey on impacts and model calculations on emission, transport and deposition were made to investigate the local,
regional, and global significance of the HLD sources. We identified 64 new HLD sources. We estimated that in the high
latitudes, the land area with higher (SI$\geq$0.5), very high (SI$\geq$0.7), and the highest potential (SI$\geq$0.9) for dust emission cover >1
000 km$^2$, >560 000 km$^2$, and >240 000 km$^2$, respectively. These estimations agree with the first HLD sources' estimate of
an area >500 000 km$^2$ by Bullard et al. (2016), which mainly included the sources with a very high potential for dust emission,
as classified in this study. Our study shows that active sources cover a significantly larger area, confirmed by over 60 new
HLD sources with evidence of their dust activity, which is not limited to dry areas. The potential HLD emission areas need
proof of observed and identified HLD emission sources.
Our modeling results on spatial distribution of the dust sources showed evidence supporting a northern High Latitude Dust
(HLD) belt, defined as the area north of 50°N, with a "transitional HLD-source area," extending at latitudes 50–58 °N in
Eurasia, 50–55°N in Canada, and a "cold HLD-source area," including areas north of 60°N in Eurasia and north of 58°N in
Canada, with currently "no dust source" area between the HLD and LLD dust belt, except for British Columbia. Using the
global atmospheric transport model SILAM, we estimated that 1.0% of the global dust emission originated from the high-
latitude regions. About 57% of the dust deposition on snow and the ice-covered Arctic regions came from HLD sources.
Our update provides crucial information on the extent of active HLD sources and their locations. Active HLD sources as
essential sources of aerosols that directly and indirectly affect climate and the environment in remote regions are often poorly
understood and predicted. HLD is likely a significant source of atmospheric iron deposition in the Southern Ocean encircling
Antarctica. More research is needed to quantify the deposition flux of HLD and nutrient (Fe, P, and trace metals such as Co)
content and solubility, which can then be fed to ocean biogeochemical models to quantify their impact on ocean
biogeochemistry. HLD is also an active ice-nucleating particle changing cloud properties, which has severe consequences
when deposited within the cryosphere. However, more studies are needed for HLD from different regions. For example,
Northern Asia HLD sources are assumed to be numerous but are difficult to access and gain information from. Such points to
the following main action items for monitoring dust in high latitudes:

· Firstly, the work on HLD sources needs a multidisciplinary combination of field, laboratory, and experimental work; remote sensing; and emission, transport and deposition modeling. An increase in observational and modeling studies improves HLD monitoring and predicting.

· Secondly, the activity of the currently identified active sources should be followed and reevaluated in the coming years and decades.

· Thirdly, research gaps and future research directions essentially include finding, identifying, and characterizing new dust sources. As soon as there is evidence of finding a new HLD source, it should be included in the list of dust sources and subject to further study.

· Fourthly, the role of different types of road dust in the Arctic could be separately assessed using a standard methodology.

Namely, in Arctic communities, road dust as a signature of non-exhaust traffic dust formed via the abrasion and wearing down of pavement, traction control materials, vehicle brakes, and tires is a common concern (e.g., Kupiainen et al., 2016; Nordic Council of Ministers, 2017). This paper excluded this type of road dust and only included significant anthropogenic road dust sources where unpaved roads are a substantial dust source. Unpaved areas of parking lots, storage areas, road shoulders, roadside lawn dust, and winter's effects could be considered, too. During winter's cold and wet conditions, dust accumulates in snow and ice and the humid road surface texture. As snow and ice melt and street surfaces dry up in spring, high amounts of dust become available for suspension. For example, in Finland, north of 60°N, a major anthropogenic dust source comes from sand and gravel uptake for building purposes from ridges formed during the Ice Age. These nonrenewable ridges cover 1.5 million ha. Since 1960, it has been estimated that approximately 40 million tons per year have been utilized (Fig. 211 of Wahlström et al., 1996). These open-sand areas are visible in aircraft photos and satellite images. In Finland, long-range transported low latitude dust contributes to the dust amounts, too (e.g., Meinander et al., 2021). Another health-significant anthropogenic springtime dust source is wintertime pavement traction sanding (Kuhns et al., 2010; Kupiainen, 2007; Stojiljkovic et al., 2019). These springtime dust events are annual but local throughout the country. In comparison, the Moscow metropolitan area (55°45'N, 37°37'E) is one of the most significant sources of dust at latitudes above 50° N, where dust's impact can extend over several hundred kilometers (Adzhiev et al., 2017). Moscow's road dust is mainly generated on paved roads, but roadside soils also contribute (Kasimov et al., 2020). Most often, unsealed soils are covered with lawns and are widespread in parks and recreational and industrial zones, characterized by heavy pollution, mixed upper horizon, and a high degree of soil cover heterogeneity.

This paper aimed to contribute beyond the state-of-the-art of HLD sources by focusing on collecting and providing information on the geographical distribution of dust-productive soils and potential dust sources. This is some of the most important

information that is currently lacking but is necessary to perform successful long-range transport and deposition modeling. The information on the geographical distribution of dust-productive soils needs evidence and verification on detected dust events and is insufficient alone. Therefore, the paper focused on identifying new dust sources, clarifying their climatic and environmental importance, and using emission, long-range transport and deposition modeling to study where the potential impact areas of the HLD sources are. Our results suggest that future HLD studies should include and update sources within the here defined high latitude dust belt, i.e, at 50–58°N in Eurasia and 50–55°N in Canada, and at >60°N in Eurasia and >58°N in Canada, as well as sources in the periphery of these regions, especially if sources are highly elevated (Wang et al. 2016).

Icelandic sources have shown that each source, even if nearby, may have different particle size distributions and optical properties. A detailed specification of the geographic distribution of potential dust productive soils, verified dust sources, and their physical (e.g., particle size distribution, optics, etc.) and mineralogical/chemical (e.g., mineral fractions, chemical composition, etc.) properties can contribute to the various topics: predicting dust forecasts (e.g., health protection warnings during extreme events); long-range emission, transport and deposition modeling; dust monitoring control; understanding extreme and rare events; Arctic protection; aviation control; health; tourist boards; assessing climate, environment, and air quality (e.g., Arctic Council Arctic Monitoring and Assessment Program AMAP, and Intergovernmental Panel on Climate Change IPCC reports); and implementing HLD in calculations on direct and indirect radiative forcing, including cloud formation, cryospheric effects, and modeling the impacts. The new observations in this study improved the representation of HLD sources for various approaches and applications related to the observed current, previous, and future environmental changes at high latitudes.

In summary, establishing continuous monitoring of HLD sources and their future changes is key to understanding the climatic and environmental effects at high latitudes, especially in the Arctic. Climate change causes permafrost thaw, decreases snow cover duration, and increases drought, glacial melt, and heatwave intensity and frequency – all leading to increasing the frequency of topsoil conditions favorable for dust emission (increasing soil's exposure to wind erosion) and the probability of dust storms. Although dust originates from natural soils, dust sources are also influenced by human activities, e.g., when deforestation and land management in cold regions leads to ecosystem collapse and desertification (Prospero et al., 2012; Arnalds, 2015). Dust storms from agricultural fields (as reported, e.g., in Poland) can reach distances over 300 km, drastically reducing visibility and resulting in hundreds of car accidents and fatalities (Hojan et al., 2019). Whether natural or anthropogenic, wildfires can result from new dust sources also (Miller et al., 2012). Hence, human actions can positively and negatively influence HLD and its effects. To understand and assess the temporal activity changes in HLD sources and the multiple impacts of high-latitude dust on the Earth systems over time, continuous monitoring and regular updates on location, particle properties, and activities of current and new HLD sources are needed.

**Competing interests.** The authors declare they have no conflict of interest.

**Special issue statement**. This article is part of the special issue "Arctic climate, air quality, and health impacts from short-lived climate forcers (SLCFs): contributions from the AMAP Expert Group (ACP/BG inter-journal SI)". It is not associated with a conference.

## Acknowledgements

This paper was developed as part of the Arctic Monitoring and Assessment Programme (AMAP), AMAP 2021 assessment: Arctic climate, air quality, and health impacts from short-lived climate forcers (SLCFs). We are grateful to the anonymous referees whose comments have been most valuable and have greatly improved our manuscript. Kaarle Kupiainen, Johanna Ikävalko, and Terhikki Manninen are gratefully acknowledged. The staff's help regarding the stations is highly appreciated.

## Financial support

This research has been supported by the Ministry for Foreign Affairs of Finland (IBA-project No. PC0TQ4BT-25). The study of dust composition in Moscow and Tiksi was supported by the Russian Science Foundation (No. 19-77-30004). Firn cores collection on southern Spitsbergen, Svalbard, has been co-funded by the Research Council of Norway, Arctic Field Grant 2018 (No. 282538), funds of the Leading National Research Centre (KNOW) received by the Centre for Polar Studies of the University of Silesia, and statutory activities No. 3841/E-41/S/2018 of the Ministry of Science and Higher Education of Poland. The Czech Science Foundation projects 20-06168Y, GA20-20240S, and the Ministry of Education, Youth and Sports of the Czech Republic projects No. LM2015078 and CZ.02.1.01/0.0/0.0/16_013/0001708 are acknowledged. The support of the EPOS-PL project (No. POIR.04.02.00-14-A003/16), co-financed by the European Union from the funds of the European Regional Development Fund (ERDF) to the laboratory facilities at IG PAS used in the study, is also acknowledged. European Union COST Action InDust is acknowledged. The preparation of this paper was partially funded by the Icelandic Research Fund (Rannis) Grant No. 207057-051. O. Meinander acknowledges funding from the Academy of Finland (ACCC Flagship funding grant No. 337552 and BBrCAC No. 341271), H2020 EU-Interact (No. 730938), International Arctic Science Committee (IASC Cross-Cutting grant), and the Ministry for Foreign Affairs of Finland (IBA-project No. PC0TQ4BT-20). D. Frolov is thankful to Lomonosov Moscow State University (state topic "Danger and risk of natural processes and phenomena" No. 121051300175-4). K. Kandler was funded by the Deutsche Forschungsgemeinschaft (DFG, German Research Foundation No. 264912134, 416816480, 417012665N). J. King acknowledges NSERC Discovery 2016-05417, CFI 36564, and the CMN RES00044975. B. Murray, A. Sanchez-Marroquin, and S. Barr thank the European Research Council (648661 MarineIce) and the Natural Environment Research Council (NE/T00648X/1; NE/R006687/1). O. Möhler and N.S. Umo acknowledge the

funding support from Helmholtz Association of German Research Centres through its 'Changing Earth — Sustaining our Future' Programme. M. Kulmala, N.S. Kasimov, and O. Popovicheva acknowledge funding from the Russian Ministry of Education and Science (075-15-2021-574). K. Ranjbar and N.T. O'Neill acknowledge the PAHA project (NSERC-CCAR program; RGPCC-433842-2012), the SACIA project (CSA-ESSDA program; 16UASACIA), and the NSERC DG grants of O'Neill (RGPIN-05002-2014). I. Semenkov, O. Popovicheva, and N. Kasimov acknowledge funding from the M.V. Lomonosov Moscow State University (the Interdisciplinary Scientific and Educational School «Future Planet and Global Environmental Change» and project No. 121051400083-1). Z. Shi and C. Baldo are funded by the UK Natural Environment Research Council (NE/L002493/1; NE/S00579X.

**Supplement**

The supplement related to this article is available online at:

**Data availability**

Data are mostly included in this article or available on request via personal communication.

**Author contribution**

The paper was initiated and lead by O. Meinander. P. Dagsson-Waldhauserová co-coordinated and edited. HLD SI and area calculations were by A. Vukovic and B. Cvetkovic. New HLD sources were identified as follows: Alaska, Canada: S. Barr, P. Dagsson-Waldhauserová, P., S. Gassó, J. King, B.J. Murray, J.B. McQuaid, N.T. O'Neill, K. Ranjbar. Antarctica: P. Dagsson-Waldhauserová, J. Kavan, K. Láska, O. Meinander, E. Shevnina. Denmark and Sweden: O. Meinander. Greenland: A. Baklanov, L.G. Benning, P. Dagsson-Waldhauserová, S. Gassó. Iceland: T. Thorsteinsson. Russia: P. Amosov, A. Baklanov, P. Enchilik, T. Koroleva, V. Krupskaya, O. Popovicheva, A. Sharapova, I. Semenkov, M. Timofeev. Svalbard: B. Barzycka, M. Kusiak, M. Laska, M. Lewandowski, B. Luks, A. Nawrot, T. Werner, K. Kandler, N. S. Umo, B.J. Murray, J.B. McQuaid, A. Sánchez-Marroquín, O. Möhler. South America, Argentina, and Patagonia: S. Gassó. DREAM model: B. Cvetkovic, S. Nickovic. SILAM model: A. Uppstu and M. Sofiev. N. Kasimov, E. Aseyeva, and O. Samonova contributed supplementary material on the central part of European Russia (potential dust source). Dust and clouds: B.J. Murray and A. Sánchez-Marroquín. Dust and ocean biogeochemistry: Z. Shi and C. Baldo. Dust and atmospheric chemistry: F. Thevenet, M.N. Romanias, J.Lasne, D. Urupina. Dust and cryosphere: O. Meinander. All authors contributed significantly to preparing the manuscript.

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
