# Peer review of "Newly identified climatically and environmentally significant high-latitude dust sources"

_Atmospheric Chemistry and Physics, 2021_

## Referee Comment (RC1)

**Review of Meinander et al. (2022): Newly identified climatically and environmentally significant high latitude dust sources.**

This manuscript describes in detail the sources of high latitude dust (HLD) from both hemispheres and the properties of this dust source in relation to its impacts throughout the Earth system. The manuscript also includes a small section on Icelandic dust modelling. The manuscript sections dealing with sources and impacts are well written and comprehensive. This alone provides an update on Bullard et al. seminal paper on HLD and will be a useful reference for a wide variety of topics. However, the modelling sections are short and appear out of context with the rest of the paper. The modelling results come across as underdeveloped, offering little-to-no new insights that I can see beyond the general literature or model description papers. The paper is really quite long, which is perhaps understandable given the Authors effort to document all HLD. Given that I can see no additional benefit from the inclusion of Icelandic modelling I suggest it is removed in entirety. This then leaves a much more concise paper on the sources and impacts. Once the modelling section is dealt with, I would recommend publication in ACP.

**Major comment:**

The two short sections on modelling Iceland dust currently do not sufficiently add enough to warrant their inclusion in what is already a long manuscript. Especially as the title indicates that this will be a paper on newly identified HLD sources and Iceland is a single location in the Northern Hemisphere which has been recognized for a while as a potentially important dust source.

The bulk of the manuscript does not read like it should include model descriptions. The majority of the DREAM section (first paragraph) is given over to model description, not results. Such text unfortunately does not aid the reader in understanding the sources or impacts of HLD and instead provides a distraction. Such text I imagine will covered, correctly, already in the Authors submitted paper: "Fully dynamic numerical prediction model for dispersion of Icelandic mineral dust". Similarly, the majority (first paragraph) of the SILAM model 'results' are instead model description.

Despite these two model sections being framed as results, I can see no new results presented here of significance, or that are not within the general literature already. E.g., Icelandic transport of aerosol or that a coarser resolution model will have difficulty capturing sub-gridscale level processes/emissions. Specifically:

(1) Figure 13 is for a single day. I feel that such a snapshot will offer the reader little insight about the general source strength or dispersion of HLD aerosol needed to make for a deeper understanding of HLD impacts. It is impossible to see the WORLDVIEW AOD in panel B and no scale is given. Furthermore, there a myriad of issues with correlating a single snapshot of total AOD with modelled dust aerosol loading – especially in a region known for high cloud clover and sea spray. There are co-authors who are experts in these issues and a thorough examination of bias is needed to provided confidence that this model is doing a 'good job' and a much longer comparison for relevance to the rest of the manuscript. This would require a significant undertaking.

(2) There is discussion of how well the SILAM model correlates to observations, yet these details are not provided anywhere, or any references given. I am therefore left not knowing how well the model does. The emission maps are better than for the DREAM model as they do provide some indication of seasonality – but again this is not a paper on Icelandic

dust sources but a paper on all HLD sources, especially new ones. A full emission map of all HLD sources would have been helpful.

Despite the HLD impacts section giving many great details on how HLD impacts clouds, chemistry, the cryosphere, and the marine biosphere, no modelling of these impacts are given using these models. This is what I would have expected the modelling results sections to be used for, but in their absence, I can see no need to have a section detailing Icelandic modelling.

Given the significant amount of work that would be needed I suggest that the modelling sections are simply removed. Particular points that the Authors feel are required to remain can be easily included in other sections. This creates a much more concise paper.

I also would like to state that modelling of HLD is a new and growing scientific avenue. And an exciting one. If the Authors are inclined, then what is provided here does provide the foundation of a new paper dedicated to this topic.

**General comments:**

High Latitude Dust locations: previously the definition of HLD has been chosen, somewhat arbitrarily, on a defined latitude (rounded) rather than a definition based on the properties of the dust and/or its emission environment. This process has been repeated here but looking at Figure 2 there appears to me to be a very clear demarcation of two dust belts. The lower latitude belt which ends around 58N in Eurasia and 55N in Canada and the higher latitude belt which begins around 60N in Siberia and 58N in Canada. With the exception of British Columbia, a clear grey 'no dust emission' region exists separating the HLD and LLD regions. Given this: Why are HLD not defined by these belts, which are based on the characteristics of the environment and thus influence the physicochemical properties of the emission?

Going further, I then find it difficult to see how #7, #8, and #48 in Figure 1 are truly HLD sources and not dust sources on the periphery of low latitude source regions, which thus would have more in common with low latitude dust than HLD. More argument is needed on why these sources do indeed share common characteristics with those of HLD even though not in the HLD dust belt.

Alternatively, the definition of Arctic dust seems more in line with the true HLD source region (rather than a sub-region as present). The 50-60N region is then at best a transition zone (containing #7, #8, and #48 ) which is the sub-region and dust here can maybe called HLD-'like'.

Abstract: Small clarifier of what SI means as this is likely to be unknown to most readers.

L71: Please define 'High' Arctic.

L75: Please clarify how volcanic origin aerosol links to dust aerosol. The resuspension of previously deposited ash?

L79: 'weather and air quality, marine life, and human health' I think would read better as 'weather, marine life, air quality and human health'. Also add relevant refs for weather/marine life/air quality.

L93: Similarly, some refs are needed.

L181-200: Define if the total area is w.r.t. the region or the globe.

Table 1" Change the 'S' column header to say surface area – no need for an acronym that needs looking up here. Add 'area' to the column headers: total (km2), land (km2), land (%). To make the table a complete standalone item, I suggest adding a column defining what SI above a given threshold mean in general terms.

Section 5: Reads well to me, although I am not an expert for many regions listed here. Maybe add the points from Figure 1 that reside in the given area.

Section 6: See main comment.

Section 7: A very nice summary but I would think a summary figure which relates the sources to the impacts would help to tie the section together and give the reader a reference point. Such a synthesis could also contain the level of certainty in some processes or regions for example as well as research priorities.

Section 7: Although introduced earlier, I could find no direct aerosol radiative forcing interaction of HLD discussion. In particular, given the high iron content (10%; L1019) of Icelandic dust mentioned earlier I think this could be interesting to add in terms of HLD impacts.

L734: Which case? Please clarify.

L885: Reference needed.

L995: Is mineral dust a source of nitrogen in itself? Linking to the atmospheric chemistry section will likely help make this statement stronger.

L995: None of these papers detail marine nutrient limitation patterns, suggest adding (Moore et al., 2013) or similar.

L1005: Iron solubility is introduced here but it needs explaining why this is important in terms of marine productivity (similarly, what soluble iron is and why it is important).

L1028: It is quite a large assumption to infer that because a volcanic eruption will alter marine biogeochemistry that HLD will too. The amount of aerosol released in an eruption is orders of magnitude higher than in a dust event. Can the authors provide evidence that the speciation and composition of these aerosol are also alike?

L1030: The Arctic is not Fe limited. It is N limited. The reference given here shows that under very specific conditions in an artificial aerosol addition experiment - low light and high nitrate already added (i.e., potentially representative of below the euphotic zone) - in a very specific area then iron could be **co-**limiting. This cannot be used to infer that the whole basin is iron limited at times. Please remove this section as it gives the impression that the iron within HLD significantly impacts Arctic biogeochemistry when there is not the evidence of this occurring in the literature to back up such a declaration.

L1040: I think a reference to glacial tilled dust here would be insightful. This has been shown to have a higher iron solubility e.g., (Schroth et al., 2009).

Discussion/Conclusions: Is it possible to add a summary table of how HLD compares to LLD in terms of important characteristics? There are many details and some figures already in the paper which detail parts of this difference. Can this information be synthesised in a concise readily referenceable manner?

Conclusions: What is the percentage of HLD emissions to total global dust emissions? And are there estimates of past/future changes in HLD emissions which can be added?

**Technical comments:**

L58**:** retrieval -> retreat; waves -> wave

L125: acronym SDS needs defining (appears in combination of longer acronym before)

L125: on resolution -> on a resolution

L185: Minimum values dust -> Minimum values within dust

L864: Australia of 22 January 2020 -> Australia on 22 January 2020.

Moore, C. M. M., Mills, M. M. M., Arrigo, K. R. R., Berman-Frank, I., Bopp, L., Boyd, P. W. W., Galbraith, E. D. D., Geider, R. J. J., Guieu, C., Jaccard, S. L. L., Jickells, T. D. D., La Roche, J., Lenton, T. M. M., Mahowald, N. M. M., Marañón, E., Marinov, I., Moore, J. K. K., Nakatsuka, T., Oschlies, A., Saito, M. A. A., Thingstad, T. F. F., Tsuda, A. and Ulloa, O.: Processes and patterns of oceanic nutrient limitation, Nat. Geosci., 6(9), 701–710, doi:10.1038/ngeo1765, 2013.

Schroth, A. W., Crusius, J., Sholkovitz, E. R. and Bostick, B. C.: Iron solubility driven by speciation in dust sources to the ocean, Nat. Geosci., 2(5), 337–340, doi:10.1038/ngeo501, 2009.

---

## Referee Comment (RC2)

Review of Meinander et al

This paper provides an overview of the state of knowledge of high latitude dust sources, identifies new sources, and provides new data on the physicochemical properties of these particles. The paper is certainly timely and a good fit for ACP; however, the paper needs major revisions to make the points clearer and more concise. A lot of information regarding the methodology used to determine the physicochemical properties of HLD is missing making it difficult to assess the quality of the work presented in this paper. My comments below are meant to be additive to the points made by the 1st reviewer.

Major Comments:

1. The paper needs to be revised to improve the clarity and readability of the work. The paper feels a bit jumbled without a clear goal of what the authors want to convey—e.g., some parts get into remote sensing, others get into detailed chemical knowledge, there is a section on impacts on clouds. I suggest making the changes suggested by Reviewer 1 regarding the modeling, shortening the sections 7.5-7.8 and providing the impacts of this dust in the introduction to motivate the work. The authors can then provide the new information learned about these impacts in the Implications at the end.

2. I also found it very difficult to tease apart what new knowledge the authors were presenting as opposed to previous findings. I was confused by the fact that the authors showed figures from previous papers yet put their own new figures in the SI. For instance, new figures on the properties of particles from high latitude dust emitted from Russia is in the SI while reprints of already published work are in the main manuscript.

3. A lot of key references and definitions are missing that would help the reader digest the new information presented.

4. The introduction needs major work. It has lots of redundancies, does not clearly articulate the goals of this work, and is missing key information needed to understand the impact of high latitude dust. Similarly, the methods section needs more information regarding the key measurements performed at each site discussed in the paper.

Specific Comments:

Introduction:

1. The Introduction needs a section on mechanisms that describe how high latitude dust is formed and how these processes differ from low latitude dust emission.

2. The introduction also needs a section on how climate change is affecting the intensity and seasonality of HLD and how, in turn, HLD is affecting climate. This last point can draw upon a very shortened version of sections 7.5-7.8.

Methods:

1. The method section needs an overview of which sites discussed have what measurement capabilities. This will help the reader interpret the information given regarding PM loadings as well as the physical and chemical information reported.

Results:

1. Figures 1-3 can be improved to more clearly show the regions described in this work. The figures have a lot of white space and are hard to read, especially figures 1 and 3.
2. Line 251 about Sr and Rb needs a citation.
3. Lines 271-272 reference new measurements but do not show the data
4. Line 352 and elsewhere in this section, what do the No.s refer to? The numbers in Figure 1?
5. Line 419: are those units correct for the dust storms of 7 mg/m3 (milligram not microgram??)
6. Section 5.7.1 presents what I believe to be new data, but all of the data is in the SI instead of in the actual manuscript and none of the data is described. The authors need to present their data here.
7. Section 5.7.3 are these measurements of sediment or aerosols? If aerosols, the KCl could also be from biomass burning rather than sylvite and more evidence is needed to prove the assignment of this particle class.
8. Line 566: how were these minerals characterized? Again, a lot of methodology is missing in this paper that needs to be provided in order to assess the quality of the work presented.
9. Lines 728-729 make a good point about the heterogeneous nature of HLD that should be emphasized in other parts of the manuscript.
10. Lines 885-886 require a reference for this statement.
11. Line 995: cite (Jickells and Moore, 2015)
12. Line 997, cite (Mahowald, 2011)

Minor Comments

1. In general, the paper needs to be carefully reviewed before resubmission. There are a lot of awkwardly worded sentences that are hard to read. I pointed out a few examples here but there were too many to point them all out in this review.
2. Line 58: "retrieval" should be "retreat"
3. Line 69: bio productivity should be biological productivity
4. Several definitions are missing in the paper such as SI, UNCOR, UNEP, etc
5. Line 128 describes "the best productive surfaces". This needs to be rephrased.
6. Lines 143-144, define the terms given "gleysols, retisols"
7. Line 148, remove an "in"
8. Line 186-187: "minimum values dust productive surface areas" needs to be rephrased.
9. Line 188: "dependable" should be "dependent" and "comprehend" should be "contain"
10. Line 222: change "glacial sediment carrying major rivers" to "major rivers carrying glacial sediment"
11. Line 301: change "roundly" to "round"

12. Line 305: change "heat supply" to "supply heat"
13. Line 375: change "south easter" to "southeastern"

REFERENCES

Jickells, T., Moore, C.M. (2015). The Importance of Atmospheric Deposition for Ocean Productivity. Annu. Rev. Ecol. Evol. Syst. 46, 481–501. https://doi.org/10.1146/annurev-ecolsys-112414-054118

Mahowald, N. (2011). Aerosol indirect effect on biogeochemical cycles and climate. Science (80-. ). 334, 794–796.

---

## Author Comment (AC1)

**Reply to reviews**

RC = Referee Comment

AR = Author Reply

**General remarks to the editor and both referees**

Please see the revised manuscript with tracked changes. The revised manuscript introduces new results, text, and discussion, as well as two authors. We include now: Hanna K Lappalainen and Markku Kulmala. Please see the revised author list. The acknowledgements have been updated. The new references are listed at the end of this reply.

**Referee 1**

**RC1.1:** "Review of Meinander et al. (2022): Newly identified climatically and environmentally significant high latitude dust sources. This manuscript describes in detail the sources of high latitude dust (HLD) from both hemispheres and the properties of this dust source in relation to its impacts throughout the Earth system. The manuscript also includes a small section on Icelandic dust modelling. The manuscript sections dealing with sources and impacts are well written and comprehensive. This alone provides an update on Bullard et al. seminal paper on HLD and will be a useful reference for a wide variety of topics. However, the modelling sections are short and appear out of context with the rest of the paper. The modelling results come across as underdeveloped, offering little-to-no new insights that I can see beyond the general literature or model description papers. The paper is really quite long, which is perhaps understandable given the Authors effort to document all HLD. Given that I can see no additional benefit from the inclusion of Icelandic modelling I suggest it is removed in entirety. This then leaves a much more concise paper on the sources and impacts. Once the modelling section is dealt with, I would recommend publication in ACP."

**AR1.1:** We thank Referee1 for the positive and constructive feedback. We agree to follow Referee1 comments to deal with the modeling section. We have removed the modeling section ("6 Modeling results on high latitude dust", p. 25-32).

**Major comment:**

**RC1.2:** "Major comment: The two short sections on modelling Iceland dust currently do not sufficiently add enough to warrant their inclusion in what is already a long manuscript. Especially as the title indicates that this will be a paper on newly identified HLD sources and Iceland is a single location in the Northern Hemisphere which has been recognized for a while as a potentially important dust source. The bulk of the manuscript does not read like it should include model descriptions. The majority of the DREAM section (first paragraph) is given over to model description, not results. Such text unfortunately does not aid the reader in understanding the sources or impacts of HLD and instead provides a distraction. Such text I

imagine will covered, correctly, already in the Authors submitted paper: "Fully dynamic numerical prediction model for dispersion of Icelandic mineral dust". Similarly, the majority (first paragraph) of the SILAM model 'results' are instead model description. Despite these two model sections being framed as results, I can see no new results presented here of significance, or that are not within the general literature already. E.g., Icelandic transport of aerosol or that a coarser resolution model will have difficulty capturing sub-gridscale level processes/emissions."

**AR1.2:** We thank Referee1 for the critical comments that are helpful to improve the manuscript.

We agree with Referee1 that the two short sections on modeling in their current format do not add enough to warrant their inclusion and we agree to remove these sections (6.1 DREAM model results, p. 25-30; and 6.2 SILAM model results, p. 30-32) from the revised manuscript.

Regarding Iceland, we agree with Referee1 that Iceland has been recognized for a while as a potentially important dust source. We included, however, in our collection 13 new sources identified in Iceland (Table S2), as compared to previously documented sources.

The main reasoning for inclusion of the two computations for Iceland in the paper, was the lack of observations and complexity of the AOD interpretation in polar and subpolar regions. In addition, we think that in absence or high uncertainty of direct measurements, the importance of the HLD modeling rises and models validated over better-observed regions may become an important or primary source of information.

**Specifically**

**RC1.3**: "Specifically (1) Figure 13 is for a single day. I feel that such a snapshot will offer the reader little insight about the general source strength or dispersion of HLD aerosol needed to make for a deeper understanding of HLD impacts. It is impossible to see the WORLDVIEW AOD in panel B and no scale is given. Furthermore, there a myriad of issues with correlating a single snapshot of total AOD with modelled dust aerosol loading – especially in a region known for high cloud clover and sea spray. There are co-authors who are experts in these issues and a thorough examination of bias is needed to provided confidence that this model is doing a 'good job' and a much longer comparison for relevance to the rest of the manuscript. This would require a significant undertaking."

**AR1.3:** Referee1 comment here deals with the modeling section, which, following the Referee1 comment, has been totally removed in its current format.

Regarding Figure 13, which has been removed, we agree that correlating a single day and a single snapshot of total AOD with a modelled dust aerosol loading in a region known for high cloud cover and sea spray is problematic. Instead, we think that the Supplementary DREAM animation of http://www.seevccc.rs/HLDpaper/NMMB_DREAM_circumpolar_dustload_animation.gif could offer the reader a better insight, and suggest to keep it and refer to it in the revised manuscript.

**RC1.4:** "Specifically (2) There is discussion of how well the SILAM model correlates to observations, yet these details are not provided anywhere, or any references given. I am therefore left not knowing how well the model does. The emission maps are better than for the DREAM model as they do provide some indication of seasonality – but again this is not a paper on Icelandic dust sources but a paper on all HLD sources, especially new ones. A full emission map of all HLD sources would have been helpful."

**AR1.4:** We thank Referee1 for the critical comments on the SILAM model section, which have been most helpful to improve the manuscript.

We agree with Referee1 that a full emission map of all HLD sources would have been helpful and therefore SILAM has been rerun to produce the whole-Arctic emission and deposition maps. Therefore, we suggest to include in the revised manuscript the following new text and results:

Abstract

"Using the global atmospheric transport model SILAM, we estimated that 1.0 % of the global dust emission originated from the high latitude regions and about 57 % of the dust deposition on snow and ice covered Arctic regions was from HLD dust."

Materials and methods

"To assess the global impact of arctic dust, estimates of the emission and deposition of both global and arctic dust have been computed separately using the SILAM model. The computations have been performed using ECMWF ERA5 meteorological reanalysis data for the year 2017 at a resolution of 0.5 x 0.5 degrees. The dust emission model has been validated against AERONET aerosol optical density (AOD) data and provides unbiased results for the main dust emission areas. For arctic areas, where dust is not contributing to the AOD as dominantly, the simulated AOD from all aerosols is unbiased with respect to the measurements. While the relatively coarse resolution of the simulation is not able to capture the smaller point-like sources of dust, it is still expected to provide a good approximation of the overall patterns and magnitudes of the dust emission and deposition."

Results

*"The SILAM model was used to estimate the total emission of arctic dust, as well as its deposition onto snow-covered land surface, frozen sea surface and total sea surface (frozen and non-frozen). The computations were performed both for arctic dust and total global dust, and the results are presented in Table X, with results for overall dust (diameter less than 30 μm) and fine dust (diameter less than 2.5 μm) presented separately. For comparison, the same values are presented also for anthropogenic black carbon, based on the Copernicus Atmosphere Monitoring Service (CAMS) global emission inventory version 4.2, and black carbon originating from wildfires from the SILAM IS4FIRES fire emission model. The IS4FIRES model is based on fires observed by the MODIS instrument onboard the Terra and Aqua satellites.

Based on the model, the total emission of arctic dust equals about 1.0 % of the global total dust emission. The deposition of arctic dust onto snow and ice covered surface equals globally about 19 % of the total dust deposition onto these areas, and about 57 % of the deposition onto the

areas located specifically in the arctic region. For fine dust, the corresponding figures are 7 % and 22 %. Compared to the deposition of black carbon (anthropogenic sources and wildfires combined) onto snow and ice, the deposition of fine arctic dust is about 70 % higher globally and about 580 % higher in the arctic regions. While these figures provide a general quantification of the deposited amounts, detailed calculations of the thermal and optical properties of dust and black carbon deposited on snow would be required for a more detailed comparison of the net impacts on the climate of the deposited substances."

[Figure]

[Figure]

[Figure]

**Figure x.** Global dust emission, deposition and and deposition on snow and ice separately, modeled using the global atmospheric transport model SILAM.

**Table X.** Emission and deposition of global dust and Arctic dust and as compared to anthropogenic BC and wildfire BC modeled using the global atmospheric transport model SILAM.

| Emission (megatonnes) | Global dust < 30 um | Global dust < 2.5 um | Arctic dust < 30 um | Arctic dust < 2.5 um | Anthropogenic black carbon (CAMS global emissions v4.2) | Wildfire black carbon (5% of fire PM emissions) |
|---|---|---|---|---|---|---|
| total emission | 3000 | 160 | 30 | 1,6 | 4,6 | 1,9 |
| deposition on snow | 32 | 5,2 | 4 | 0,21 | 0,18 | 0,029 |
| deposition on sea ice | 5,5 | 0,59 | 3 | 0,17 | 0,009 | 0,008 |
| deposition on arctic snow | 7,6 | 1,1 | 4 | 0,19 | 0,027 | 0,013 |
| deposition on arctic sea ice | 4,7 | 0,52 | 3 | 0,17 | 0,0055 | 0,0074 |
| deposition on sea surface | 500 | 86 | 15 | 1,0 | 1,7 | 0,9 |
| deposition on arctic sea surface | 21 | 2,4 | 12 | 0,68 | 0,035 | 0,063 |

**RC1.5**: "Despite the HLD impacts section giving many great details on how HLD impacts clouds, chemistry, the cryosphere, and the marine biosphere, no modelling of these impacts are given using these models. This is what I would have expected the modelling results sections to be used for, but in their absence, I can see no need to have a section detailing Icelandic modelling. Given the significant amount of work that would be needed I suggest that the modelling sections are simply removed. Particular points that the Authors feel are required to remain can be easily included in other sections. This creates a much more concise paper. I also would like to state that modelling of HLD is a new and growing scientific avenue. And an exciting one. If the Authors are inclined, then what is provided here does provide the foundation of a new paper dedicated to this topic."

**AR1.5:** We thank Referee1 for the constructive and useful feedback.

We agree about the necessity to clarify and condense the modelling part, generalize the message, structure it, and formulate properly to reflect the actual value of the computations. We are thankful to Referee1 for comment on modeling saying that "*Particular points that the Authors feel are required to remain can be easily included in other sections.*"

We have now given motivation and clarified in the Introduction, why long-range transport modeling can be useful. Our paper aims at identification of dust sources and focusing on their climatic and environmental impacts. To investigate if dust sources have local, regional or global significance, new emission and deposition calculations have now been included in our paper. We think that characterizing the sources and modeling the impacts are important future steps to be taken but including these is beyond the scope and possibilities of the current paper.

The following modifications have been made to particular points of long-range transport modeling to include these in other sections, as suggested by Referee 1:

- the original modeling section has been removed

- SILAM has been rerun to produce the whole-Arctic emission and deposition maps, as presented in AR1.4

- particular points from modeling sections have been included in other sections as follows:
  Figure 11: DREAM Figure 11a removed
  Figure 12: DREAM Figure 12a removed
  Figure 11 b & 12 b make a new DREAM Figure included in the Iceland section
  Figure 13: DREAM Figure 13a and 13b removed
  Figure 13c: DREAM Figure in Greenland section and its animation included as "Supplementary DREAM animation of http://www.seevccc.rs/HLDpaper/NMMB_DREAM_circumpolar_dustload_animation.gif", in Greeenland section
  Figure 14: SILAM Figure 14 included in the Iceland section

- connection to the rest of the paper and the added value of the model computations have been clarified

- the model descriptions of DREAM and SILAM have been briefly included in the Materials and methods as follows:

(SILAM was given in AR1.4)

"DREAM is a fully dynamic numerical prediction model for atmospheric dust dispersion originating from soil sources. The dust component of this modelling system (Pejanovic et al., 2011; Nickovic et al., 2016) is online driven by the atmospheric model NMME (Janjic et al., 2001). Dust concentration in the model is described with eight particle bins with radii ranging from 0.18–9 μm. DREAM-ICELAND is the model version arranged to predict dust transport emitted from the largest European dust sources in Iceland (Cvetkovic et al., 2021, submitted). The size distribution of particles in the model is specified according to in-situ measurements in the Icelandic hot spots. The model horizontal resolution of ~3.5 km is sufficiently fine to resolve rather heterogeneous and small-scale character of the Icelandic dust sources (Fig. XX). DREAM-ICELAND, as the first operational numerical HLD model in the international community, is used to daily predict the Icelandic dust since April 2018 (Fig. XX).

**General comments**

**RC1.6:** "General comments: High Latitude Dust locations: previously the definition of HLD has been chosen, somewhat arbitrarily, on a defined latitude (rounded) rather than a definition based on the properties of the dust and/or its emission environment. This process has been repeated here but looking at Figure 2 there appears to me to be a very clear demarcation of two dust belts. The lower latitude belt which ends around 58N in Eurasia and 55N in Canada and the higher latitude belt which begins around 60N in Siberia and 58N in Canada. With the

exception of British Columbia, a clear grey 'no dust emission' region exists separating the HLD and LLD regions. Given this: Why are HLD not defined by these belts, which are based on the characteristics of the environment and thus influence the physicochemical properties of the emission?

**AR1.6:** We are grateful to Referee1 for this comment, which, first of all, points out the difficulties in defining high latitude dust. This question is also related to the similar problem of defining the Arctic, where no single correct definition exists either. For the Arctic, we used the definition of AMAP (north of 60 °N). For HLD, we used the definition by Bullard et al. (2016).

In our paper, we focused on presenting the source intensity values (areas with potential sources) and identifying new sources in wide latitudinal regions ($\geq$ 50 °N and $\geq$ 40 °S). We then compared our results on the dust emission land area against Bullard et al. (2016). We found that e land area with higher (SI$\geq$0.5), very high (SI$\geq$0.7) and the highest potential (SI$\geq$0.9) for dust emission cover >1 670 000 km$^2$, >560 000 km$^2$, and >240 000 km$^2$, respectively, and that this agrees with the first HLD sources estimate of an area >500 000 km$^2$ by Bullard et al. (2016).

As suggested by Referee1, we gratefully agree that it is worth considering, if, based on our results (Fig. 2), we could take one step further in defining HLD.

Referring to Referee1 comment, we agree that we could consider to say that our results (Fig. 2) suggest that two northern high latitude dust belts could be identified, where the first HLD belt would extend at 50-58 °N in Eurasia and 50-55 °N in Canada, and the second dust belt would extend at >60 °N in Eurasia and >58 °N in Canada, with a 'no dust' belt between HLD and LLD dust belt (with the exception of British Columbia).

We think that at this point it is, however, too ambitious to define more than one HLD belt in the HLD region. We think that the northern mid-latitude belt could be included as "cold HLD-source area", but the southern HLD part we would suggest to call "transitional HLD-source area", because it actually represents transition between two major climate temperature belts and dust source belts. Hence, after careful consideration, we would like to suggest the include the following sentence in the revised manuscript:

"Our spatial dust source distribution analysis modeling results (Fig. 2), showed evidence in support of a northern High Latitude Dust (HLD) belt, defined as the area north of 50°N, where we distinguish the following HLD-source areas: (a) 'transitional HLD-source area' which extends at latitudes 50-58 °N in Eurasia and 50-55 °N in Canada, and (b) 'cold HLD-source area' which includes areas north of 60 °N in Eurasia and north of 58 °N in Canada; with currently 'no dust source' area between HLD and LLD dust belt (with the exception of British Columbia)."

**RC1.7:** "Going further, I then find it difficult to see how #7, #8, and #48 in Figure 1 are truly HLD sources and not dust sources on the periphery of low latitude source regions, which thus would have more in common with low latitude dust than HLD. More argument is needed on why these sources do indeed share common characteristics with those of HLD even though not in the HLD dust belt. "

**AR1.7:** We agree with Referee1 that sources #7 and #8 could be identified as dust sources on the periphery of low latitude source regions. But they could be identified as dust sources in the periphery of HLD, too.

As for the source #7, it is the Altai mountains. Some parts of these territories are covered by permafrost. Winter lasts for 5–6 months there. In lower mountains (less than 1000 m a.s.l.), a stable snow cover persists from October and in higher mountains (more than 1500 m a.s.l.) – from September. The mean daily air temperature during winter within the areas of lower, middle and higher mountains is –21°C, –29°C and less than –30°C, respectively.

The source #8 occurs in Central Kazakhstan. From late December to early March, there is a stable snow cover with a thickness from 5 cm to 30 cm within plains and up to 50 cm within hollows. Periods of snow cover establishment and thaw correspond to transitions of the mean daily temperature of air through 0°C, which on average are the 7th of November and 23rd of March plus/minus 10–12 days. From early January to the late February, the mean daily temperature of air can be as low as –20°C.

Soil Atlas of the Northern Circumpolar Region (https://esdac.jrc.ec.europa.eu/content/soil-atlas-northern-circumpolar-region) covers all land surfaces in Eurasia and North America above the latitude of 50 ºN. So, soil scientists suggest these territories are located in high latitudes.

**RC1.8:** "Alternatively, the definition of Arctic dust seems more in line with the true HLD source region (rather than a sub-region as present). The 50-60N region is then at best a transition zone (containing #7, #8, and #48 ) which is the sub-region and dust here can maybe called HLD-'like'."

**AR1.8**: We agree that many HLD sources are in the Arctic region, when Arctic is defined as north from 60ºN, as is according to AMAP (SLCF Expert Group). Hence, like Referee1 says, Arctic would not be a subregion of HLD, but more in line with the true HLD region. As suggested by Referee1 in RC1.6, based on Fig.2, we agree, and have suggested the following (repeated from AR1.6):

"Our spatial dust source distribution analysis modeling results (Fig. 2), showed evidence in support of a northern High Latitude Dust (HLD) belt, defined as the area north of 50°N, where we distinguish the following HLD-source areas: (a) 'transitional HLD-source area' which extends at latitudes 50-58 °N in Eurasia and 50-55 °N in Canada, and (b) 'cold HLD-source area' which includes areas north of 60 °N in Eurasia and north of 58 °N in Canada; with currently 'no dust source' area between HLD and LLD dust belt (with the exception of British Columbia)."

**RC1.9:** "Abstract: Small clarifier of what SI means as this is likely to be unknown to most readers."

**AR1.9:** Thank you, now clarified as:

"Here, we identify, describe, and quantify the Source Intensity (SI) values, which show the potential of soil surfaces for dust emission scaled to values 0 to 1 with respect to globally best

productive sources, using the Global Sand and Dust Storms Source Base Map (G-SDS-SBM), for sixty-four HLD sources included in our collection in the Northern (Alaska, Canada, Denmark, Greenland, Iceland, Svalbard, Sweden, and Russia) and Southern (Antarctica and Patagonia) high latitudes."

**RC1.10**: "L71: Please define 'High' Arctic. L75: Please clarify how volcanic origin aerosol links to dust aerosol. The resuspension of previously deposited ash?"

**AR1.10:** High Arctic (>80°N), has been added to the text. Volcanic origin in this context means rather suspension from volcanic deserts or volcanic glacial outwash plains around glaciers. It has been shown that resuspension of volcanic ash occurs only few months after eruption. The main mechanism is volcanic dust suspension. We added 'desert' into the text to have this clearer - volcanic desert origin.

RC1.11: "L79: 'weather and air quality, marine life, and human health' I think would read better as 'weather, marine life, air quality and human health'. Also add relevant refs for weather/marine life/air quality."

**AR1.11:** Thank you, now changed, and new references added for weather/marine life/air quality. The new references are presented at the end of Author reply.

**RC1.12:** "L93: Similarly, some refs are needed."

**AR1.12**: Thank you. New reference has been added for "HLD is a short-lived climate forcer, air pollutant and nutrient source…" [L93]. The new references are presented at the end of Author reply and in the tracked changes manuscript.

**RC1.13**: "L181-200: Define if the total area is w.r.t. the region or the globe."

**AR1.13:** We have included the sentence which explains "total area" before this part of the text: "Further analysis includes assessment of areal coverage of sources, with different thresholds for SI values, in absolute values (km2) and percentage they occupy with respect to the total land surface area in defined HLD regions."

**RC1.14:** "Table 1" Change the 'S' column header to say surface area – no need for an acronym that needs looking up here. Add 'area' to the column headers: total (km2), land (km2), land (%). To make the table a complete standalone item, I suggest adding a column defining what SI above a given threshold mean in general terms."

**AR1.14:** Changed as suggested. Instead of adding a column defining the meaning SI above a threshold, we added a short explanation in the Table header.

**RC1.15:** "Section 5: Reads well to me, although I am not an expert for many regions listed here. Maybe add the points from Figure 1 that reside in the given area."

**AR1.15:** Thank you for the positive comment. As suggested, we have added the points from Figure 1 into text in the first paragraph of section 5 (Observations and characteristics of the identified regional dust sources).

**RC1.16:** "Section 6: See main comment."

**AR1.16:** Section 6 has been removed.

**RC1.17:** "Section 7: A very nice summary but I would think a summary figure which relates the sources to the impacts would help to tie the section together and give the reader a reference point. Such a synthesis could also contain the level of certainty in some processes or regions for example as well as research priorities."

**AR1.17:** Thank you. We have re-organized sections 7.5-7.8 to be in their correct place as part of literature survey results and not previously known information (referring to Referee 2 comments) The co-authors who contributed to these Sections were added in "Authors contribution". We suggest and have now included the following illustrative summary figure, which relates sources to impacts.

[Figure]

**RC1.18**: "Section 7: Although introduced earlier, I could find no direct aerosol radiative forcing interaction of HLD discussion. In particular, given the high iron content (10%; L1019) of Icelandic dust mentioned earlier I think this could be interesting to add in terms of HLD impacts."

**AR1.18**: Thank you, this is true. We have now added sentences on radiative forcing in Discussion and in Introduction, where also motivation to impacts sections is now clarified.

**RC1.19:** "L734: Which case? Please clarify."

**AR1.19:** 'in this case' is misleading and has now been removed to clarify the sentence.

**RC1.20**: "L885: Reference needed."

**AR1.20**: Thank you, now added. The new references are presented at the end of Author reply and in the tracked changes manuscript.

**RC1.21:** "L995: Is mineral dust a source of nitrogen in itself? Linking to the atmospheric chemistry section will likely help make this statement stronger."

**AR1.21:** Mineral dust particles are a source of essential nutrients such as phosphorus (P) and iron (Fe) to the ocean ecosystems. They are not a significant source of atmospheric nitrogen which is dominated by anthropogenic emissions (Jickells and Moore, 2015; Jickells et al., 2016).

We have revised the text as follows:

Mineral dust particles are a source of essential nutrients such as phosphorus (P) and iron (Fe) to the ocean ecosystems (e.g., Jickells et al., 2005; Mahowald et al., 2005; Stockdale et al., 2016).

**RC1.22**: "L995: None of these papers detail marine nutrient limitation patterns, suggest adding (Moore et al., 2013) or similar."

**AR1.22:** We agree. References have been updated in the text as follows:

The extent of these impacts primarily depends on the dust deposition fluxes and its chemical properties, and the nutrients (co)limitations patterns in the ocean waters (e.g., Boyd et al., 2007; Boyd et al., 2010; Kanakidou et al., 2018; Mahowald et al., 2010; Mills et al., 2004; Moore et al., 2013; Shi et al., 2012; Stockdale et al., 2016).

**RC1.23:** "L1005: Iron solubility is introduced here but it needs explaining why this is important in terms of marine productivity (similarly, what soluble iron is and why it is important)."

**AR1.23:** The aerosol fractional Fe solubility (%) is defined as the ratio of dissolved Fe (in the filtrate which has passed through 0.2 or 0.45 µm pore size filters) to the total Fe contained in the bulk aerosol (e.g., Meskhidze et al., 2019; Shi et al., 2012). It is important to know this because not all iron is bio-accessible. A majority of the refractory iron is not available for biology. There is a lot of debate about what is bio-accessible but current technology does not allow an accurate assessment of bio-accessibility. Fractional Fe solubility is therefore used to indicate the fraction of Fe which is likely to be bio-accessible for marine ecosystems (Meskhidze et al., 2019).

This has been added to the text as follows:

The aerosol fractional Fe solubility (%) is defined as the ratio of dissolved Fe (in the filtrate which has passed through 0.2 or 0.45 µm pore size filters) to the total Fe contained in the bulk aerosol (e.g., Meskhidze et al., 2019; Shi et al., 2012). This is typically used to indicate the fraction of Fe which is likely to be bio-accessible for marine ecosystems (Meskhidze et al., 2019).

**RC1.24:** "L1028: It is quite a large assumption to infer that because a volcanic eruption will alter marine biogeochemistry that HLD will too. The amount of aerosol released in an eruption is orders of magnitude higher than in a dust event. Can the authors provide evidence that the speciation and composition of these aerosol are also alike?"

**AR1.24:** We agree. There are insufficient observations to compare the aerosol speciation, but we have shown that, as volcanic ash, Icelandic dust is also primarily composed of glass. In this sense, their composition is similar, but we recognize that more observations of volcanic ash speciation will be needed. It is noted that Achterberg et al. (2013) measured an initial fractional Fe solubility of 0.04 %-0.14 % for Icelandic ash which is below or towards the lower end of range of values estimated for Icelandic dust (0.08%-0.6%, Baldo et al., 2020). More onboard and incubation measurements are needed to better understand the impact of Icelandic dust on surface ocean

We have revised the text as follows:

However, during the 2010 eruption of the Icelandic volcano Eyjafjallajökull, Achterberg et al. (2013) observed elevated dissolved Fe concentration and nitrate depletion in the Iceland Basin, followed by an early spring bloom. They measured an initial fractional Fe solubility of 0.04 %-0.14 % for Icelandic ash which is below or towards the lower end of range of values estimated for Icelandic dust (0.08%-0.6%, Baldo et al., 2020). High deposition flux (Arnalds et al., 2016) and higher Fe solubility of Icelandic dust (Baldo et al., 2020) suggests that they may impact Fe biogeochemistry and primary productivity in the surface ocean but more research is needed to confirm this.

**RC1.25:** "L1030: The Arctic is not Fe limited. It is N limited. The reference given here shows that under very specific conditions in an artificial aerosol addition experiment - low light and high nitrate already added (i.e., potentially representative of below the euphotic zone) - in a very specific area then iron could be co-limiting. This cannot be used to infer that the whole basin is iron limited at times. Please remove this section as it gives the impression that the iron within HLD significantly impacts Arctic biogeochemistry when there is not the evidence of this occurring in the literature to back upsuch a declaration."

**AR1.25:** Thank you. We agree. Lines 1030-1035 have been deleted.

**RC1.26**: "L1040: I think a reference to glacial tilled dust here would be insightful. This has been shown to have a higher iron solubility e.g., (Schroth et al., 2009)."

**AR1.26**: We agree. As suggested, we compared the fractional Fe solubility from mineral dust sources in Antarctica measured by Winton et al. (2016) with the Fe solubility observed in dust originating from glacial sediments on the Gulf of Alaska coastline (Schroth et al., 2017), but also with Fe solubility of Icelandic dust which are also influenced by glacial processes.

The fractional Fe solubility in mineral dust sources in Antarctica was 0.7% (Winton et al. 2016) which is comparable to the upper limit of Fe solubilities observed in Icelandic dust (Baldo et al., 2020). However, dust sourced in the Gulf of Alaska coastline showed higher Fe solubility (1.4%, Schroth et al., 2017). These is likely due to the different mineralogy and Fe speciation in the samples. The different methods used to determine the fractional Fe solubility in these studies may also contribute to this difference (Perron et al., 2020).

We have revised the text as follows:

Winton et al. (2016) reported a background fractional Fe solubility from Antarctic dust sources of 0.7% which is similar to the upper limit of Fe solubilities observed in Icelandic dust (Baldo et al., 2020). However, mineral dust originating from glacial sediments from the Gulf of Alaska coastline showed higher Fe solubilities (1.4%, Schroth et al., 2017). This is likely due to the different mineralogy and Fe speciation in the samples. The different methods used to determine the fractional Fe solubility in these studies may also contribute to this difference (Perron et al., 2020).

**RC1.27**: "Discussion/Conclusions: Is it possible to add a summary table of how HLD compares to LLD in terms of important characteristics? There are many details and some figures already in the paper which detail parts of this difference. Can this information be synthesised in a concise readily referenceable manner?"

**AR1.27:** Thank you, we agree that this is a good suggestion. However, it would take considerable effort to produce and we therefore think it is beyond what can be achieved in this paper.

In this paper, the focus is on identifying new sources (Fig. 1) and defining the source areas (Fig. 2), and then pointing out climatic and environmental impacts of the HLD sources, and further using long-range transport modeling to study where the potential impact areas of the

HLD sources are located. As the next step, the properties of the dust particles in the identified source areas should be investigated, also adding into this context what are the properties of the particles that are lifted up in the atmosphere, long-range trans51ported and deposited.

**RC1.28:** "Conclusions: What is the percentage of HLD emissions to total global dust emissions? And are there estimates of past/future changes in HLD emissions which can be added?"

**AR1.28:** To define percentage of HLD emission to global dust emission, assessments require data on observed dust emissions of HLD which are very limited (for example satellite observations mostly do not cover highest latitudes, like MODIS AOD) and could be underestimated, or global dust cycle model climate runs. Dust-atmospheric models require input information on dust sources, which are relatively unknown for this region, and this paper has a purpose to explore such sources. Modeling of atmospheric transport of HLD dust just starts to develop.

Such assessments are currently, at best, at the level of hemisphere or large emissive regions (North Africa, Middle East, etc.), as to the knowledge of authors.

For example, recently published:

https://acp.copernicus.org/articles/21/8169/2021/acp-21-8169-2021.pdf

Regarding future emissions assessments, it would require coupled atmospheric-dust models which consider change of land conditions under future GHG emission scenarios (change of vegetation, retrieval of glaciers, etc.), and simulations on relatively high resolution because of the relatively small-scale sources in HLD. Today, numerical models which comprehend all required conditions to represent well HLD climate (past and future) are still not developed.

New sentence to Conclusions:

"Contribution of HLD emissions to global dust cycle remains unknown, as to the knowledge of the authors. Such assessments require observations, which are limited in this area, and atmospheric-dust model simulations, which are still under development for these regions and require high resolution input information on HLD sources and high resolution runs because of the relatively small-scale features of the sources. Under the impact of climate change, we believe that emissions of HLD will be submitted to changes, but future assessments also remain unknown because of the current modeling limitations and lack of information on sources change under future GHG emission scenarios."

**Technical comments**

**RC1.29:** "Technical comments: L58: retrieval -> retreat; waves -> wave"

**AR1.29**: Now changed.

**RC1.30**: "L125: acronym SDS needs defining (appears in combination of longer acronym before)"

**AR1.30**: Now defined.

**RC1.31**: "L125: on resolution -> on a resolution L185: Minimum values dust -> Minimum values within dust "

**AR1.31:** Now changed.

**RC1.32**: "L864: Australia of 22 January 2020 -> Australia on 22 January 2020. "

**AR1.32:** Now changed.

**RC1.33**: "Moore, C. M. M., Mills, M. M. M., Arrigo, K. R. R., Berman-Frank, I., Bopp, L., Boyd, P. W. W., Galbraith, E. D. D., Geider, R. J. J., Guieu, C., Jaccard, S. L. L., Jickells, T. D. D., La Roche, J., Lenton, T. M. M., Mahowald, N. M. M., Marañón, E., Marinov, I., Moore, J. K. K., Nakatsuka, T., Oschlies, A., Saito, M. A. A., Thingstad, T. F. F., Tsuda, A. and Ulloa, O.: Processes and patterns of oceanic nutrient limitation, Nat. Geosci., 6(9), 701–710, doi:10.1038/ngeo1765, 2013. "

**AR1.33:** Now added.

**RC1.34:** "Schroth, A. W., Crusius, J., Sholkovitz, E. R. and Bostick, B. C.: Iron solubility driven by speciation in dust sources to the ocean, Nat. Geosci., 2(5), 337–340, doi:10.1038/ngeo501, 2009."

**AR1.34:** Now added.

**Referee 2**

**RC2.1**: "Review of Meinander et al. This paper provides an overview of the state of knowledge of high latitude dust sources, identifies new sources, and provides new data on the physicochemical properties of these particles. The paper is certainly timely and a good fit for ACP; however, the paper needs major revisions to make the points clearer and more concise. A lot of information regarding the methodology used to determine the physicochemical properties of HLD is missing making it difficult to assess the quality of the work presented in this paper. My comments below are meant to be additive to the points made by the 1st reviewer."

**AR2.1:** We thank Referee2 for the positive as well as critical and constructive feedback. We have revised and clarified the manuscript according to Referee2 comments, in addition to revision based on comments given by Referee1. The revised version is presented using track-changes.

**Major Comments**

**RC2.2:** "Major Comments: 1. The paper needs to be revised to improve the clarity and readability of the work. The paper feels a bit jumbled without a clear goal of what the authors want to convey—e.g., some parts get into remote sensing, others get into detailed chemical knowledge, there is a section on impacts on clouds. I suggest making the changes suggested by Reviewer 1 regarding the modeling, shortening the sections 7.5-7.8 and providing the impacts of this dust in the introduction to motivate the work. The authors can then provide the new information learned about these impacts in the Implications at the end."

**AR2.2:** Thank you. As suggested and supported by Referee 2, we have made the changes suggested by Referee 1 regarding the modeling.

As constructively criticized by Referee 2, we have now revised the manuscript to make the points clearer and more concise, and the methodology used in the various sections is now presented in Materials and methods.

We have also clarified in the revised manuscript that our paper aims at identification of dust sources and focusing on their climatic and environmental impacts. To investigate if dust sources have local, regional or global significance, new emission and deposition calculations have now been included in our paper. This comment by Referee 2 we find most helpful to improve and clarify the manuscript further, and it concerns also sections 7.5-7.8, which Referee 2 suggests to be shortened. Namely, the story behind section 7.5-7.8 is that co-authors were originally invited by the first author to provide in a google docs their names and which sections they would like to contribute to, including a possibility to suggest additional sections, too. As a result of this interactive process by all the co-authors, the contributions to 7.5-7.8 were created, including the sections of 7.5 Impacts of HLD on clouds and climate feedbacks, 7.6 on atmospheric chemistry, 7.7 on marine environment and 7.8 on cryosphere and cryosphere-atmosphere feedbacks. Sections 7.5-7.8 do not provide a literature review, but a literature survey, which information is now included in the revised manuscript in the Materials and methods section. The co-authors who contributed to these Sections were added in "Authors contribution". In addition, as suggested by Referee 2, we have provided information on the impacts of dust in the introduction to motivate the work, i.e., also motivating the literature survey presented in sections 7.5-7.8. as part of the manuscript.

**RC2.3**: "2. I also found it very difficult to tease apart what new knowledge the authors were presenting as opposed to previous findings. I was confused by the fact that the authors showed figures from previous papers yet put their own new figures in the SI. For instance, new figures on the properties of particles from high latitude dust emitted from Russia is in the SI while reprints of already published work are in the main manuscript."

**AR2.3:** Thank you. Yes, it is true that Supplementary includes figures with new data. We placed the figures on properties of particles from high latitude dust emitted from Russia in the Supplementary because these data refer not to the active, but a potential dust source and represent partitioning of elements among soil particle size fractions. The reason to show these data on topsoil horizons in the central part of the East European Plain and place this information into the Supplementary we described in section 7.2 "Comparison of various regions". To avoid confusion, we edited and shortened this paragraph to make it clearer to the readers, as shown in the track-changes version of the manuscript.

Regarding the reprint of the already published figures (Figure 7 redrawn from Varga et al., and Figure 8, which was extracted from the Supplementary Material of Sanchez-Marroquin, 2020), it was necessary to show these Figures in the main manuscript because we wanted to raise the importance of detecting differences in Icelandic dust sources inside Iceland and as compared to Saharan dust. Figure 7 is redrawn to show the variability in the grain size distributions of samples from Icelandic source areas. Figure 8 shows chemical composition of Icelandic dust particles (a) and Saharan dust particles collected in Barbados (b).

**RC2.4**: "3. A lot of key references and definitions are missing that would help the reader digest the new information presented."

**AC2.4**: Thank you, we have now added references as listed, e.g., at the end of the author reply, and improved definitions are given to terms and abbreviations used in the manuscript, including SI and LLD, for example.

**RC2.5**: "4. The introduction needs major work. It has lots of redundancies, does not clearly articulate the goals of this work, and is missing key information needed to understand the impact of high latitude dust. Similarly, the methods section needs more information regarding the key measurements performed at each site discussed in the paper."

**AC2.5**: Thank you for this comment on Introduction and Materials and methods, which has helped us to improve the readability of the manuscript. We have now clarified Introduction. We have added sentences on impacts of HLD. The methods section has been revised to include information regarding the key methods, including the key measurements performed at each site. The following new text in Introduction has been added:

"The fundamental processes controlling aeolian dust emissions in high latitudes are essentially the same as in temperate regions, but there are additional processes specific to or enhanced in cold regions. Low temperatures, humidity, strong winds, permafrost and niveo-aeolian processes, which can affect the efficiency of dust emission and distribution of sediments, were listed in Bullard et al. (2016). The IPCC special report (IPCC 2019) recognizes dark dust aerosols as short-lived climate forcer (SLCF) and light-absorbing aerosols connected to cryospheric changes. Dust aerosols are not just one type of aerosols but consist of a variety of different dust particle types with various particle sizes and shapes distributions, as well as chemical, physical and optical properties. Therefore, impacts on climate and environment can differ from those of LLD, too. For example, Icelandic dust is of volcanic origin, often dark, and consists of higher proportions of heavy metals than crustal dust."

New text in the Material and methods has been added as follows:

2.4 Literature survey
Environmental and climatic impacts of HLD were investigated with the help of literature surveys. The co-authors were invited to provide in a shared internet document their names and to which impact they would like to contribute to, including a possibility to suggest additional impact sections. As a result, the contributions to sections 7.5-7.8 were created, including the sections of 7.5 Impacts of HLD on clouds and climate feedbacks, 7.6 on atmospheric chemistry, 7.7 on marine environment and 7.8 on cryosphere and cryosphere-atmosphere feedbacks. These impact sections do not provide a literature review, but literature surveys, where a brief summary, not reviewing, of the content of the selected cited sources is provided. Each impact section presents an independent literature survey and has its own co-author list, as indicated in the author contribution section.

**Specific Comments**

**Introduction**

**RC2.6:** "Specific Comments: Introduction: 1. The Introduction needs a section on mechanisms that describe how high latitude dust is formed and how these processes differ from low latitude dust emission."

**AR2.6**: Thank you, we have now revised the introduction to include motivation of the paper, mechanisms that describe how high latitude dust is formed and how these processes differ from low latitude dust emission (AC2.5), and impacts of HLD (revised Introduction can be found in the tracked changes version of the manuscript).

**RC2.7**: "2. The introduction also needs a section on how climate change is affecting the intensity and seasonality of HLD and how, in turn, HLD is affecting climate. This last point can draw upon a very shortened version of sections 7.5-7.8."

**AR2.7**: As suggested by Referee 2, we agree, and have added in the introduction the following: how climate change is affecting the intensity and seasonality of HLD and how, in turn, HLD is affecting climate. HLD affecting climate in the sections 7.5-7.8. is part of the contents of the paper based on literature survey, as now explicitly described in the Materials and methods section.

**Methods**

**RC2.8**: "Methods: 1. The method section needs an overview of which sites discussed have what measurement capabilities. This will help the reader interpret the information given regarding PM loadings as well as the physical and chemical information reported."

**AR2.8**: Thank you, we have revised the Materials and methods section accordingly.

**Results**

**RC2.9:** "Results: 1. Figures 1-3 can be improved to more clearly show the regions described in this work. The figures have a lot of white space and are hard to read, especially figures 1 and 3. "

**AR2.9**: Thank you. To improve the readability of the Figures, we changed Figure 2 and 3 into landscape mode and provided to the publisher high quality Figures, which can be zoomed. Note that, because of the high resolution of the data, and scattered patterns of the sources in some regions, it is likely that some values cannot be visible even if the Figures are made for the specific regions. For this reason, we are referencing the web-portal https://maps.unccd.int/sds/ where values of G-SDS-SBM can be seen.

As it is challenging to present clearly all sources at one place (or figure) clearly, we provided the Figures just as general overview of the regional abundance of identified sources and spatial distribution of potential dust sources.

Figures 2 and 3 are scaled so that surfaces of both figures are of the same size, in order to show the difference of the areal coverage of potential sources in these two belts "on the first look". To reduce the size of the manuscript, we have provided specific values from Figure 2 and 3 for the locations from Figure 1 in the Supplementary material.

**RC2.10**: "2. Line 251 about Sr and Rb needs a citation."

**AR2.10:** The reference was added.

**RC2.11**: 3. Lines 271-272 reference new measurements but do not show the data

**AR2.11**: Thank you, yes this is correct, no data are shown here but the reference to Meinander et al. 2018 is given and data were shown there We have now rephrased this.

**RC2.12**: "4. Line 352 and elsewhere in this section, what do the No.s refer to? The numbers in Figure 1? "

**AR2.12**: Thank you. Yes, it has been better clarified.

Manuscript changes: L352 and L354 (2x) - Change (No. 51) into (Location 51 in figure 1), (Location 15 in figure 1), (Location 1 in figure 1).

The number are the identified 64 dust sources as shown in Figure 1 and additional information, including latitude and longitude and SI values, can be found in Supplement Table S1, S2 , S 3 and S4, for example. We have now clarified this information in the revised manuscript.

**RC2.13:** 25. Line 419: are those units correct for the dust storms of 7 mg/m3 (milligram not microgram??) "

**AR2.13:** Indeed, it is over 7000 ug m-3, meaning 7 milligram per m3. We changed the unit to micrograms, so it is not confusing.

Manuscript changes: L419 – Change 7 mgm-3 into 7000 $\mu$gm-3.

**RC2.14:** 6. Section 5.7.1 presents what I believe to be new data, but all of the data is in the SI instead of in the actual manuscript and none of the data is described. The authors need to present their data here.

**AR2.14:** Thank you. We are grateful for this comment as we first had these data in the actual manuscript. The Tables are large and hard to be presented in the main body of manuscript, but we agree and will present the data in the actual manuscript.

**RC2.15: 7**. Section 5.7.3 are these measurements of sediment or aerosols? If aerosols, the KCl could also be from biomass burning rather than sylvite and more evidence is needed to prove the assignment of this particle class.

**AR2.15**: Yes, aerosol measurements as stated in the first word of the paragraph. Thank you for this comment. We have reworked this part based. It is focused on the findings related to dust possible sources rather than to identify sources for all the elements.

Manuscript changes: L473 Analysis of wind and aerosol pollutants roses combined with long-range transport analysis helped to identify the sources for dust at Tiksi, demonstrating impacts either from lower latitudes or/and local emissions from the adjacent urban Tiksi area. In warm periods, $Na^+$, $Cl^-$, $K^+$, and $Mg^{2+}$ are found to be the major ions in the sea-salt aerosols which are ubiquitous in the marine boundary layer and significantly impact the dust concentrations in coastal region. However, $Cl^-$ and $K^+$ could also originate from biomass burning during the warm period.

**RC2.16:** 8. Line 566: how were these minerals characterized? Again, a lot of methodology is missing in this paper that needs to be provided in order to assess the quality of the work presented.

**AR2.16:** Details of methodology that lead to the conclusion of "the presence of magnetite and iron sulfide" are given in Lewandowski et al. (2020). Magnetic susceptibility upon heating results revealed the presence of magnetite (primary or/and secondary). SEM data indicated iron sulfides. More detailed description was included in the text." (magnetic susceptibility and SEM data, Lewandowski et al., 2020).

**RC2.17:** 9. Lines 728-729 make a good point about the heterogeneous nature of HLD that should be emphasized in other parts of the manuscript.

**AR2.17:** Thank you, we agree and have now emphasized this in other parts of the manuscript, too. [Lines 728-729 say: "… until the source characteristics and particle properties have been characterized more in detail. For example, Icelandic sources have shown that each source, even located closely, may have different particle size distributions and optical properties."]

**RC2.18: 10.** Lines 885-886 require a reference for this statement.

**AR2.18**: Thank you, the reference is:

Winton, V. H. L., Dunbar, G. B., Bertler, N. A. N., Millet, M. A., Delmonte, B., Atkins, C. B., Chewings, J. M., and Andersson, P.: The contribution of aeolian sand and dust to iron fertilization of phytoplankton blooms in southwestern Ross Sea, Antarctica, Global Biogeochemical Cycles, 28, 423-436, doi: 10.1002/2013gb004574, 2014.

[Lines 885-886 say: "The aeolian sediment has an impact on sea ice albedo (not directly measured), marine sedimentation and contributes enough dissolved Fe to support potentially up to 15% of primary productivity in the SW Ross Sea."]

**RC2.19:** 11. Line 995: cite (Jickells and Moore, 2015)

**AR2.19**: References (RC2.19) has been updated in the text as follows:

Mineral dust particles are a source of essential nutrients such as phosphorus (P) and iron (Fe) to the ocean ecosystems (e.g., Jickells et al., 2005; Mahowald et al., 2005; Stockdale et al., 2016). Dust deposition onto the ocean's surface has the potential to stimulate primary productivity and consequently enhance carbon uptake, which indirectly affects the climate (e.g., Jickells and Moore, 2015; Mahowald, 2011).

**RC2.20**: 12. Line 997, cite (Mahowald, 2011)

**AR2.20**: References (RC2.20) have been updated in the text as follows:

Mineral dust particles are a source of essential nutrients such as phosphorus (P) and iron (Fe) to the ocean ecosystems (e.g., Jickells et al., 2005; Mahowald et al., 2005; Stockdale et al., 2016). Dust deposition onto the ocean's surface has the potential to stimulate primary productivity and consequently enhance carbon uptake, which indirectly affects the climate (e.g., Jickells and Moore, 2015; Mahowald, 2011).

**Minor comments**

**RC2.21**: Minor Comments 1. In general, the paper needs to be carefully reviewed before resubmission. There are a lot of awkwardly worded sentences that are hard to read. I pointed out a few examples here but there were too many to point them all out in this review.

**AR2:21**: Thank you. We will carefully review the manuscript before resubmission.

**RC2.22**: 2. Line 58: "retrieval" should be "retreat"

**AR2:22:** Changed.

**RC2.23**: "3. Line 69: bio productivity should be biological productivity"

**AR2:23**: Changed.

**RC2.24**: 4. Several definitions are missing in the paper such as SI, UNCOR, UNEP, etc

**AR2:24:** Thank you. Now defined.

**RC2.25**: 5. Line 128 describes "the best productive surfaces". This needs to be rephrased.

**AR2:25:** Now rephrased.

**RC2.26:** "6. Lines 143-144, define the terms given "gleysols, retisols" "

**AR2:26:** The terms are defined according to (FAO, 2015, which refers to FAO: World reference base for soil resources 2014 International soil classification system, FAO, Rome., 2015) and (IUSS Working Group WRB, 2015). Since the officially recommended reference is IUSS 2015, we have included in the revised manuscript the following:

The reference has been added in the text as:
 (IUSS Working Group WRB, 2015)

The new reference added in the References is:
IUSS Working Group WRB. 2015. World Reference Base for Soil Resources 2014, update 2015 International soil classification system for naming soils and creating legends for soil maps. World Soil Resources Reports No. 106. FAO, Rome

**RC2.27**: 7. Line 148, remove an "in"

**AR2:27:** Removed.

**RC2.28**: 8. Line 186-187: "minimum values dust productive surface areas" needs to be rephrased.

**AR2:28:** Rephrased as "Surface of dust productive areas of minimum seasonal SI values"

**RC2.29**: 9. Line 188: "dependable" should be "dependent" and "comprehend" should be "contain"

**AR2:29**: Changed.

**RC2.30:** 10. Line 222: change "glacial sediment carrying major rivers" to "major rivers carrying glacial sediment"

**AR2:30:** Changed.

**RC2.31:** 11. Line 301: change "roundly" to "round"

**AR2:31:** Changed.

**RC2.32**: 12. Line 305: change "heat supply" to "supply heat"

**AR2:32**: Now changed.

**RC2.33:** 13. Line 375: change "south easter" to "southeastern"

**AR2:33:** Now changed.

**RC2.34:** REFERENCES Jickells, T., Moore, C.M. (2015). The Importance of Atmospheric Deposition for Ocean Productivity. Annu. Rev. Ecol. Evol. Syst. 46, 481–501. https://doi.org/10.1146/annurev-ecolsys-112414-054118

**AR2:34:** Now added.

**RC2.35**: Mahowald, N. (2011). Aerosol indirect effect on biogeochemical cycles and climate. Science (80-. ). 334, 794–796.

**AR2:35**: Now added.

**New references**

[revised manuscript text omitted]

Wheaton, E. E. and Chakravarti, A. K.: Dust storms in the Canadian Prairies, Int. J. Climatol., 10(8), 829–837, doi:https://doi.org/10.1002/joc.3370100805, 1990.. Alos, there is work from Cheryl McKenna that should be included like Neuman, C. M.: Observations of winter aeolian transport and niveo-aeolian deposition at crater lake, pangnirtung pass, N.W.T., Canada, Permafr. Periglac. Process., 1(3–4), 235–247, doi:10.1002/ppp.3430010304, 1990.

Hugenholtz, C. H. and Wolfe, S. A.: Rates and environmental controls of aeolian dust accumulation, Athabasca River Valley, Canadian Rocky Mountains, Geomorphology, 121(3), 274–282, doi:https://doi.org/10.1016/j.geomorph.2010.04.024, 2010.

---

## Author Response (AR2)

**Reply to reviews (revised manuscript)**

RC = Referee Comment

AR = Author Reply

**General remarks to the editor and both referees**

Please see the revised manuscript with tracked changes and with one new reference. One of the anonymous referees has indicated in the review report that she/he is willing to become acknowledged by her/his name. We are grateful to both anonymous referees whose comments have been most valuable and have greatly improved our manuscript. This new sentence we suggest being added in the acknowledgements of our manuscript, and the acknowledgements to be then completed by the editorial office with the correct name(s) if the manuscript gets accepted.

**Referee 1**

**RC1.1**

"**Anonymous during peer-review: Yes** No

**Anonymous in acknowledgements of published article:** Yes **No**

**Recommendation to the editor**

| | |
|---|---|
| **1) Scientific significance**
Does the manuscript represent a substantial contribution to scientific progress within the scope of this journal (substantial new concepts, ideas, methods, or data)? | Outstanding Ex |
| **2) Scientific quality**
Are the scientific approach and applied methods valid? Are the results discussed in an appropriate and balanced way (consideration of related work, including appropriate references)? | Outstanding Ex |
| **3) Presentation quality**
Are the scientific results and conclusions presented in a clear, concise, and well structured way (number and quality of figures/tables, appropriate use of English language)? | Outstanding Ex |

For final publication, the manuscript should be

accepted as is

**accepted subject to technical corrections**

accepted subject to minor revisions

reconsidered after major revisions

rejected

**Were a revised manuscript to be sent for another round of reviews:**

**I would be willing to review the revised manuscript.**

I would not be willing to review the revised manuscript."

**Suggestions for revision or reasons for rejection (will be published if the paper is accepted for final pu**

**RC1.2:** *"The Authors have addressed all my comments satisfactorily. The paper is clearer and well structured* the new SILAM modelling emission/deposition patterns, and related data analysis, is insightful and I feel was defi effort. I recommend publication in ACP."

**RC1.3:** *"Note that some new Figures and Tables have the holder 'X' instead of the new number, which will ne* publication. The first column of new table "Emission and deposition of global dust..." is defined as emission, but e data is presented in the rows; therefore, maybe alter header to be "flux" or similar."

**AR1.1:** We thank Referee1 for the positive feedback. We are grateful to Referee1 whose comments have been most valuable and have greatly improved our manuscript. We are happy to acknowledge Referee1 by name in the acknowledgements of our manuscript if it gets accepted.

**AR1.2:** Thank you, we are happy to hear this.

**AR1.3:** We are grateful to Referee1 for checking the new numbers of the new Figures and Tables and have updated these. The header of the new table "Emission and deposition of global dust..." in the column of "emission" (below, surrounded with red color) has been altered to be Flux, as suggested by Referee1, since the rows represent emission and deposition data.

[Figure]

| EMISSION (MEGATONNES) | GLOBAL DUST < 30 µm | GLOBAL DUST < 2.5 µm | ARCTIC DUST < 30 µm | ARCTIC DUST < 2.5 µm | ANTHROPOGENIC BLACK CARBON (CAMS global emissions v4.2) | WILDFIRE BLACK CARBON (5% of fire PM emissions) |
|---|---|---|---|---|---|---|
| Total emission | 3000 | 160 | 30 | 1,6 | 4,6 | 1,9 |
| Deposition on snow | 32 | 5,2 | 4 | 0,21 | 0,18 | 0,029 |
| Deposition on sea ice | 5,5 | 0,59 | 3 | 0,17 | 0,009 | 0,008 |
| Deposition on Arctic snow | 7,6 | 1,1 | 4 | 0,19 | 0,027 | 0,013 |
| Deposition on Arctic Sea ice | 4,7 | 0,52 | 3 | 0,17 | 0,0055 | 0,0074 |
| Deposition on sea surface | 500 | 86 | 15 | 1,0 | 1,7 | 0,9 |
| Deposition on Arctic Sea surface | 21 | 2,4 | 12 | 0,68 | 0,035 | 0,063 |

New revised:

[Figure]

| FLUX (MEGATONNES) | GLOBAL DUST < 30 μm | GLOBAL DUST < 2.5 μm | ARCTIC DUST < 30 μm | ARCTIC DUST < 2.5 μm | ANTHROPOGENIC BLACK CARBON (CAMS global emissions v4.2) | WILDFIRE BLACK CARBON (5% of fire PM emissions) |
|---|---|---|---|---|---|---|
| Total emission | 3000 | 160 | 30 | 1,6 | 4,6 | 1,9 |
| Deposition on snow | 32 | 5,2 | 4 | 0,21 | 0,18 | 0,029 |
| Deposition on sea ice | 5,5 | 0,59 | 3 | 0,17 | 0,009 | 0,008 |
| Deposition on Arctic snow | 7,6 | 1,1 | 4 | 0,19 | 0,027 | 0,013 |
| Deposition on Arctic Sea ice | 4,7 | 0,52 | 3 | 0,17 | 0,0055 | 0,0074 |
| Deposition on sea surface | 500 | 86 | 15 | 1,0 | 1,7 | 0,9 |
| Deposition on Arctic Sea surface | 21 | 2,4 | 12 | 0,68 | 0,035 | 0,063 |

**Referee 2**

**RC2.1:**

Anonymous during peer-review: Yes No

Anonymous in acknowledgements of published article: Yes No

**Recommendation to the editor**

| | |
|---|---|
| **1) Scientific significance**
Does the manuscript represent a substantial contribution to scientific progress within the scope of this journal (substantial new concepts, ideas, methods, or data)? | Outstanding **Excellent** Good Fair Low |
| **2) Scientific quality**
Are the scientific approach and applied methods valid? Are | Outstanding **Excellent** Good Fair Low |

the results discussed in an appropriate and balanced way (consideration of related work, including appropriate references)?

| 3) Presentation quality
Are the scientific results and conclusions presented in a clear, concise, and well structured way (number and quality of figures/tables, appropriate use of English language)? | Outstanding Excellent **Good** Fair Low |
|---|---|

For final publication, the manuscript should be

accepted as is

accepted subject to technical corrections

**accepted subject to minor revisions**

reconsidered after major revisions

rejected

Were a revised manuscript to be sent for another round of reviews:

**I would be willing to review the revised manuscript.**

I would not be willing to review the revised manuscript.

**Suggestions for revision or reasons for rejection (will be published if the paper is accepted for final publication)**

**RC2.2:** This revised manuscript is much improved. I have just a few minor comments.

For the introduction, there are a few redundancies that should be removed on lines 134-143 and lines 151-153

**RC2.3:** Lines 221-231 belong in the introduction of the paper

**RC2.4:** Dust sources #7 and #8 are outside of the authors self-defined latitudinal cut off for HLD.

**RC2.5:** For Figure 9, it would be helpful to include the EDX spectra with the SEM images.

**AR2.1:** We are thankful to Referee2 for the positive feedback. We thank Referee2 whose review has been most helpful and has greatly improved our manuscript. We are happy to acknowledge the valuable work of anonymous Referee2 in the acknowledgements.

**AR2.2:** Thank you, we are pleased to hear this. The redundances of the introduction on lines 134-143 and lines 151-153 have now been removed.

**AR2.3:** We agree and have moved the lines 221-231 in the introduction.

**AR2.4:** We thank Referee2 for pointing out that sources no. 7 and 8 are outside our self-defined latitudinal cut off for HLD. It is true that our collection contains two sources that are outside the latitudinal cut off for HLD, when we have followed the definition of Bullard et al. (2016) saying that HLD refers to ≥ 50°N and ≥ 40°S.

We are most grateful for Referee2 for bringing this up once more and making us to clarify the manuscript further. Namely, as an outcome of our own results, we say in the revised manuscript (lines 1001-1004):

*"The results (Fig. 2) suggest two northern high-latitude dust belts. The first HLD belt would extend at 50–58°N in Eurasia and 50–55°N in Canada, and the second dust belt at >60°N in Eurasia and >58°N in Canada, with a "no dust" belt between the HLD and LLD dust belts (except for British Columbia)."*

Hence, our results suggest a potential need for updating the definition for HLD. Therefore, we have now removed from the abstract the latitudinal cut off for HLD latitudes according to Bullard et al. (2016) as follows:

**Abstract.** Dust particles from high latitudes have a potentially large local, regional, and global significance to climate and the environment as short-lived climate forcers, air pollutants, and nutrient sources. Identifying the locations of local dust sources and their emission, transport, and deposition processes is important for understanding the multiple impacts of High Latitude Dust (HLD) on the Earth's systems. Here, we identify, describe, and quantify the Source Intensity (SI) values, which show the potential of soil surfaces for dust emission scaled to values 0 to 1 concerning globally best productive sources, using the Global Sand and Dust Storms Source Base Map (G-SDS-SBM). This includes sixty-four HLD sources in our collection for the Northern (Alaska, Canada, Denmark, Greenland, Iceland, Svalbard, Sweden, and Russia) and Southern (Antarctica and Patagonia) high latitudes. Activity from most of these HLD dust sources shows seasonal character. It is estimated that high-latitude land areas with higher (SI≥0.5), very high (SI≥0.7), and the highest potential (SI≥0.9) for dust emission cover >1 670 000 km$^2$, >560 000 km$^2$, and >240 000 km$^2$, respectively. In the Arctic HLD region (≥ 60°N), land area with SI≥0.5 is 5.5% (1 035 059 km$^2$), area with SI≥0.7 is 2.3% (440 804 km$^2$), and with SI≥0.9 is 1.1% (208 701 km$^2$). Minimum SI values in the north HLD region are about three orders of magnitude smaller, indicating that the dust sources of this region greatly depend on weather conditions. Our spatial dust source distribution analysis modeling results showed evidence supporting a northern High Latitude Dust (HLD) belt, defined as the area north of 50°N, with a 'transitional HLD-source area' extending at latitudes 50–58°N in Eurasia and 50–55°N in Canada, and a 'cold HLD-source area' including areas north of 60°N in Eurasia and north of 58°N in Canada, with currently 'no dust source' area between the HLD and LLD dust belt, except for British Columbia. Using the global atmospheric transport model SILAM, we estimated that 1.0% of the global dust emission originated from the high-latitude regions. About 57% of the dust deposition in snow- and ice-covered Arctic regions was from HLD sources. In the south HLD region, soil surface conditions are favorable for dust emission during the whole year. Climate change can decrease snow cover duration, retrieval of glaciers, and increase drought, heatwave intensity, and frequency, leading to the increasing frequency of topsoil conditions favorable for dust emission, which increases the probability of dust storms. Our study provides a step forward to improve the representation of HLD in models and to monitor, quantify, and assess the environmental and climate significance of HLD going forward.

Similarly, we have checked the consistency of using HLD definition throughout the manuscript, keeping in mind that our results suggest a need to update the definition for HLD. For example, in "3 Results and discussion":

**3 Results and discussion**

**3.1 Locations of the HLD sources**

Sixty-four HLD sources at northern and southern high latitudes (Fig. 1) were identified. In the north HLD region are 49 locations (47 locations ≥50°N and two >47°N) in A
Sweden, and Russia, of 35 are in the Arctic HLD subregion (≥60°N). In the south HLD region (≥40°S), 15 sources were identified in Antarctica and Patagonia, South A

and in "3.6.1 Source intensity values", where we also found a typo in the latitude for sources no. 8 and no. 48; now corrected, as follows:

Forty-nine locations were in the north HLD region (47 according to HLD definition by Bullard et al. (2016), (except for two: no. 8 and

When checking and discussing these latitudinal definitions among the co-authors, we found that we would need carefully consider the sources no. 7, 8 and 48. In the appendix, we have the coordinates for these sources as follows:

| 7 | 51.3 | 88.5 | 0.0 | 0.0 | 0.0 | 0.0 | 0.2 | 0.0 | 0.4 | 0.0 | 0.4 |
| 8 | 47.3 | 66.7 | 0.5 | 0.0 | 0.5 | 0.0 | 0.6 | 0.3 | 0.7 | 0.4 | 1.0 |
| 48 | 47.6 | -111.25 | 0.5 | 0.1 | 0.8 | 0.1 | 0.8 | 0.7 | 1.0 | 0.7 | 1.0 |

Hence, no. 7 is inside the defined HLD according to Bullard et al. (2016), but no. 8 and no. 48 are not. In the revised manuscript (lines 600-610), we described sources no. 7 and no. 8 as follows:

*"Some Russian sources included in our collection (e.g., no. 7 and 8 of Fig. 1) could be identified as dust sources on the periphery of HLD and low-latitude source regions. Source no. 7 of Fig. 1 is the Altai Mountains. Some parts of these territories are covered by permafrost, where winter lasts for 5–6 months. From October, in lower mountains (less than 1000 m a.s.l.), and from September, in higher mountains (more than 1500 m a.s.l.), a stable snow cover persists. The mean daily air temperature during winter within the lower, middle, and higher mountains is –21°C, –29°C, and below –30°C, respectively. Source no. 8 is in Central Kazakhstan. From late December to early March, a stable snow cover from 5 cm to 30 cm occurs within plains and up to 50 cm within hollows. Periods of snow cover and thaw correspond to transitions of the mean daily temperature of air through 0°C, which, on average, are the 7 November and 23 March plus/minus 10–12 days. From early January to late February, the air's mean daily temperature can be as low as –20°C. Soil Atlas of the Northern Circumpolar Region (https://esdac.jrc.ec.europa.eu/content/soilatlas-northern-circumpolar-region) covers all land surfaces in Eurasia and North America above the latitude of 50°N. Thus, these territories are considered high-latitude."*

Keeping in mind that our results suggest a need to update the definition for HLD, we would like to suggest that including #8 and #48 in our collection is reasonable, when referring to the new self-defined high latitude dust belt of our manuscript and suggesting to add one **new sentence** and **one new reference in the Conclusions** to clarify further by saying that:

*"Our results suggest that future HLD studies should include and update sources within the here defined high latitude dust belt, i.e., at 50–58°N in Eurasia and 50–55°N in Canada, and at >60°N in Eurasia and >58°N in Canada, as well as sources in the periphery of these regions, especially if sources are highly elevated (Wang et al. 2016)."*

*New reference:*
*Wang, Q., Fan, X. & Wang, M. Evidence of high-elevation amplification versus Arctic amplification, Sci Rep, 6, 19219, https://doi.org/10.1038/srep19219, 2016.*

**AR2.5:** Thank you. We agree that it could be helpful to include EDX spectra with the SEM images. Legend for Figure 9 (Figure 13 in the revised version) is created based on the EDX spectra statistical analyses. We performed such analyses just after sampling in 2015-2016. Unfortunately, however, we did not withdraw all EDX spectra from the software data set stored on the EDX machine. Now this instrument does not operate anymore, and unfortunately, we cannot fully address this comment.